# ON THE STABILITY OF EXPRESSIVE POSITIONAL ENCODINGS FOR GRAPHS

**Yinan Huang**[*,1]**, William Lu**[*,2]**, Joshua Robinson**[3]**, Yu Yang**[4]**, Muhan Zhang**[5]**,
Stefanie Jegelka**[6]**, Pan Li**[1]
[1]Georgia Institute of Technology, [2]Purdue University, [3]Stanford University, [4]Tongji University,
[5]Peking University, [6] MIT CSAIL
{yhuang903, panli}@gatech.edu, lu909@purdue.edu,
joshrob@cs.stanford.edu, yangyu0879@tongji.edu.cn, muhan@pku.edu.cn, stefje@mit.edu

## ABSTRACT

Designing effective positional encodings for graphs is key to building powerful graph transformers and enhancing message-passing graph neural networks. Although widespread, using Laplacian eigenvectors as positional encodings faces two fundamental challenges: (1) *Non-uniqueness*: there are many different eigendecompositions of the same Laplacian, and (2) *Instability*: small perturbations to the Laplacian could result in completely different eigenspaces, leading to unpredictable changes in positional encoding. Despite many attempts to address nonuniqueness, most methods overlook stability, leading to poor generalization on unseen graph structures. We identify the cause of instability to be a "hard partition" of eigenspaces. Hence, we introduce Stable and Expressive Positional Encodings (SPE), an architecture for processing eigenvectors that uses eigenvalues to "softly partition" eigenspaces. SPE is the first architecture that is (1) provably stable, and (2) universally expressive for basis invariant functions whilst respecting all symmetries of eigenvectors. Besides guaranteed stability, we prove that SPE is at least as expressive as existing methods, and highly capable of counting graph structures. Finally, we evaluate the effectiveness of our method on molecular property prediction, and out-of-distribution generalization tasks, finding improved generalization compared to existing positional encoding methods. Our code is available at `https://github.com/Graph-COM/SPE`.

## 1 INTRODUCTION

Deep learning models for graph-structured data such as Graph Neural Networks (GNNs) and Graph Transformers have been arguably one of the most popular machine learning models on graphs, and have achieved remarkable results for numerous applications in drug discovery, computational chemistry, and social network analysis, etc. (Kipf & Welling, 2017; Bronstein et al., 2017; Duvenaud et al., 2015; Stokes et al., 2020; Zhang & Chen, 2018; Ying et al., 2021; Rampášek et al., 2022b). However, there is a common concern about these models: the limited *expressive power*. For example, it is known that message-passing GNNs are at most expressive as the Weisfeiler-Leman test (Xu et al., 2019; Morris et al., 2019) in distinguishing non-isomorphic graphs, and in general cannot even approximate common functions such as the number of certain subgraph patterns (Chen et al., 2020; Arvind et al., 2020; Tahmasebi et al., 2020; Huang et al., 2023). These limitations could significantly restrict model performance, e.g., since graph substructures can be closely related to the target function in chemistry, biology and social network analysis (Girvan & Newman, 2002; Granovetter, 1983; Koyutürk et al., 2004; Jiang et al., 2010; Bouritsas et al., 2022).

To alleviate expressivity limitations, there has been considerable interest in designing effective positional encodings for graphs (You et al., 2019; Dwivedi & Bresson, 2021; Wang et al., 2022a). Generalized from the positional encodings of 1-D sequences for Transformers (Vaswani et al., 2017), the idea is to endow nodes with information about their relative position within the graph and thus make them more distinguishable. Many promising graph positional encodings use the eigenvalue decomposition of the graph Laplacian (Dwivedi et al., 2023; Kreuzer et al., 2021). The eigenvalue

---

[*]equal contribution

decomposition is a strong candidate because the Laplacian fully describes the adjacency structure of a graph, and there is a deep understanding of how these eigenvectors and eigenvalues inherit this information (Chung, 1997). However, eigenvectors have special structures that must be taken into consideration when designing architectures that process eigenvectors.

Firstly, eigenvectors are *not* unique: if $v$ is an eigenvector, then so is $-v$. Furthermore, when there are multiplicities of eigenvalues then there are many more symmetries, since any orthogonal change of basis of the corresponding eigenvectors yields the same Laplacian. Because of this basis ambiguity, neural networks that process eigenvectors should be *basis invariant*: applying basis transformations to input eigenvectors should not change the output of the neural network. This avoids the pathological scenario where different eigendecompositions of the same Laplacian produce different model predictions. Several prior works have explored sign and basis symmetries of eigenvectors. For example, Dwivedi & Bresson (2021); Kreuzer et al. (2021) randomly flip the sign of eigenvectors during training so that the resulting model is robust to sign transformation. Lim et al. (2023) instead design new neural architectures that are invariant to sign flipping (SignNet) or basis transformation (BasisNet). Although these basis invariant methods have the right symmetries, they do not yet account for the fact that two Laplacians that are similar but distinct may produce completely different eigenspaces.

This brings us to another important consideration, that of *stability*. Small perturbations to the input Laplacian should only induce a limited change of final positional encodings. This "small change of Laplacians, small change of positional encodings" actually generalizes the previous concept of basis invariance and proposes a stronger requirement on the networks. But this stability (or continuity) requirement is a great challenge for graphs, because small perturbations can produce completely different eigenvectors if some eigenvalues are close (Wang et al. (2022a), Lemma 3.4). Since the neural networks process eigenvectors, not the Laplacian matrix itself, they run the risk of being highly discontinuous with respect to the input matrix, leading to an inability to generalize to new graph structures and a lack of robustness to any noise in the input graph's adjacency. In contrast, stable models enjoy many benefits such as adversarial robustness (Cisse et al., 2017; Tsuzuku et al., 2018) and provable generalization (Sokolić et al., 2017).

Unfortunately, existing positional encoding methods are not stable. Methods that only focus on sign invariance (Dwivedi & Bresson, 2021; Kreuzer et al., 2021; Lim et al., 2023), for instance, are not guaranteed to satisfy "same Laplacian, same positional encodings" if multiplicity of eigenvalues exists. Basis invariant methods such as BasisNet are unstable because they apply different neural networks to different eigensubspaces. In a high-level view, they perform a *hard partitioning* of eigenspaces and treat each chunk separately (see Appendix C for a detailed discussion). The discontinuous nature of partitioning makes them highly sensitive to perturbations of the Laplacian. The hard partition also requires fixed eigendecomposition thus unsuitable for graph-level tasks. On the other hand, Wang et al. (2022a) proposes a provably stable positional encoding. But, to achieve stability, it completely ignores the distinctness of each eigensubspaces and processes the merged eigenspaces homogeneously. Consequently, it loses expressive power and has, e.g., a subpar performance on molecular graph regression tasks (Rampášek et al., 2022a).

**Main contributions.** In this work, we present Stable and Expressive Positional Encodings (SPE). The key insight is to perform a **soft and learnable** "partition" of eigensubspaces in a **eigenvalue dependent** way, hereby achieving both stability (from the soft partition) and expressivity (from dependency on both eigenvalues and eigenvectors). Specifically:

- SPE is provably stable. We show that the network sensitivity w.r.t. the input Laplacian is determined by the gap between the $d$-th and $(d + 1)$-th smallest eigenvalues if using the first $d$ eigenvectors and eigenvalues. This implies our method is stable regardless of how the used $d$ eigenvectors and eigenvalues change.

- SPE can universally approximate basis invariant functions and is as least expressive as existing methods in distinguishing graphs. We also prove its capability in counting graph substructures.

- We empirically illustrate that introducing more stability helps generalize better but weakens the expressive power. Besides, on the molecule graph prediction datasets ZINC and Alchemy, our method significantly outperforms other positional encoding methods. On DrugOOD (Ji et al., 2023), a ligand-based affinity prediction task with domain shifts, our method demonstrates a clear and constant improvement over other unstable positional encodings. All these validate the effectiveness of our stable and expressive method.

## 2 PRELIMINARIES

**Notation.** We always use $n$ for the number of nodes in a graph, $d \leq n$ for the number of eigenvectors and eigenvalues chosen, and $p$ for the dimension of the final positional encoding for each node. We use $\|\cdot\|$ to denote the $L2$ norm of vectors and matrices, and $\|\cdot\|_{\mathsf{F}}$ for the Frobenius norm of matrices.

**Graphs and Laplacian Encodings.** Denote an undirected graph with $n$ nodes by $\mathcal{G} = (\boldsymbol{A}, \boldsymbol{X})$, where $\boldsymbol{A} \in \mathbb{R}^{n \times n}$ is the adjacency matrix and $\boldsymbol{X} \in \mathbb{R}^{n \times p}$ is the node feature matrix. Let $\boldsymbol{D} = \mathrm{diag}([\sum_{j=1}^{n} A_{i,j}]_{i=1}^{n})$ be the diagonal degree matrix. The normalized Laplacian matrix of $\mathcal{G}$ is a positive semi-definite matrix defined by $\boldsymbol{L} = \boldsymbol{I} - \boldsymbol{D}^{-1/2}\boldsymbol{A}\boldsymbol{D}^{-1/2}$. Its eigenvalue decomposition $\boldsymbol{L} = \boldsymbol{V}\mathrm{diag}(\boldsymbol{\lambda})\boldsymbol{V}^{\top}$ returns eigenvectors $\boldsymbol{V}$ and eigenvalues $\boldsymbol{\lambda}$, which we denote by $\mathrm{EVD}(\boldsymbol{L}) = (\boldsymbol{V}, \boldsymbol{\lambda})$. In practice we may only use the smallest $d \leq n$ eigenvalues and eigenvectors, so abusing notation slightly, we also denote the smallest $d$ eigenvalues by $\boldsymbol{\lambda} \in \mathbb{R}^{d}$ and the corresponding $d$ eigenvectors by $\boldsymbol{V} \in \mathbb{R}^{n \times d}$. A Laplacian positional encoding is a function that produces node embeddings $\boldsymbol{Z} \in \mathbb{R}^{n \times p}$ given $(\boldsymbol{V}, \boldsymbol{\lambda}) \in \mathbb{R}^{n \times d} \times \mathbb{R}^{d}$ as input.

**Basis invariance.** Given eigenvalues $\boldsymbol{\lambda} \in \mathbb{R}^{d}$, if eigenvalue $\lambda_i$ has multiplicity $d_i$, then the corresponding eigenvectors $\boldsymbol{V}_i \in \mathbb{R}^{n \times d_i}$ form a $d_i$-dimensional eigenspace. A vital symmetry of eigenvectors is the infinitely many choices of basis eigenvectors describing the same underlying eigenspace. Concretely, if $\boldsymbol{V}_i$ is a basis for the eigenspace of $\lambda_i$, then $\boldsymbol{V}_i\boldsymbol{Q}_i$ is, too, for any orthogonal matrix $\boldsymbol{Q}_i \in O(d_i)$. The symmetries of each eigenspace can be collected together to describe the overall symmetries of $\boldsymbol{V}$ in terms of the direct sum group $O(\boldsymbol{\lambda}) := \oplus_i O(d_i) = \{\oplus_i \boldsymbol{Q}_i \in \mathbb{R}^{\sum_i d_i \times \sum_i d_i} : \boldsymbol{Q}_i \in O(d_i)\}$, i.e., block diagonal matrices with $i$th block belonging to $O(d_i)$. Namely, for any $\boldsymbol{Q} \in O(\boldsymbol{\lambda})$, both $(\boldsymbol{V}, \boldsymbol{\lambda})$ and $(\boldsymbol{V}\boldsymbol{Q}, \boldsymbol{\lambda})$ are eigendecompositions of the same underlying matrix. When designing a model $f$ that takes eigenvectors as input, we want $f$ to be *basis invariant*: $f(\boldsymbol{V}\boldsymbol{Q}, \boldsymbol{\lambda}) = f(\boldsymbol{V}, \boldsymbol{\lambda})$ for any $(\boldsymbol{V}, \boldsymbol{\lambda}) \in \mathbb{R}^{n \times d} \times \mathbb{R}^{d}$, and any $\boldsymbol{Q} \in O(\boldsymbol{\lambda})$.

**Permutation equivariance.** Let $\Pi(n) = \{\boldsymbol{P} \in \{0,1\}^{n \times n} : \boldsymbol{P}\boldsymbol{P}^{\top} = \boldsymbol{I}\}$ be the permutation matrices of $n$ elements. A function $f : \mathbb{R}^{n} \to \mathbb{R}^{n}$ is called permutation equivariant, if for any $\boldsymbol{x} \in \mathbb{R}^{n}$ and any permutation $\boldsymbol{P} \in \Pi(n)$, it satisfies $f(\boldsymbol{P}\boldsymbol{x}) = \boldsymbol{P}f(\boldsymbol{x})$. Similarly, $f : \mathbb{R}^{n \times n} \to \mathbb{R}^{n}$ is said to be permutation equivariant if satisfying $f(\boldsymbol{P}\boldsymbol{X}\boldsymbol{P}^{\top}) = \boldsymbol{P}f(\boldsymbol{X})$.

## 3 A PROVABLY STABLE AND EXPRESSIVE PE

In this section we introduce our model *Stable and Expressive Positional Encodings* (SPE). SPE is both stable and a maximally expressive basis invariant architecture for processing eigenvector data, such as Laplacian eigenvectors. We begin with formally defining the stability of a positional encoding. Then we describe our SPE model, and analyze its stability. In the final two subsections we show that higher stability leads to improved out-of-distribution generalization, and show that SPE is a universally expressive basis invariant architecture.

### 3.1 STABLE POSITIONAL ENCODINGS

Stability intuitively means that a small input perturbation yields a small change in the output. For eigenvector-based positional encodings, the perturbation is to the Laplacian matrix, and should result in a small change of node-level positional embeddings.

**Definition 3.1** (PE Stability). *A PE method* $\mathrm{PE} : \mathbb{R}^{n \times d} \times \mathbb{R}^{d} \to \mathbb{R}^{n \times p}$ *is called stable, if there exist constants* $c, C > 0$, *such that for any Laplacian* $\boldsymbol{L}, \boldsymbol{L}'$,

$$\|\mathrm{PE}(\mathrm{EVD}(\boldsymbol{L})) - \boldsymbol{P}_{*}\mathrm{PE}(\mathrm{EVD}(\boldsymbol{L}'))\|_{\mathsf{F}} \leq C \cdot \left\|\boldsymbol{L} - \boldsymbol{P}_{*}\boldsymbol{L}'\boldsymbol{P}_{*}^{\top}\right\|_{\mathsf{F}}^{c}, \tag{1}$$

*where* $\boldsymbol{P}_{*} = \arg\min_{\boldsymbol{P} \in \Pi(n)} \left\|\boldsymbol{L} - \boldsymbol{P}\boldsymbol{L}'\boldsymbol{P}^{\top}\right\|_{\mathsf{F}}$ *is the permutation matrix matching two Laplacians.*

It is worth noting that here we adopt a slightly generalized definition of typical stability via Lipschitz continuity ($c = 1$). This definition via Hölder continuity describes a more comprehensive stability behavior of PE methods, while retaining the essential idea of stability.

**Remark 3.1** (Stability implies permutation equivariance). *Note that a PE method is permutation equivariant if it is stable: simply let* $\boldsymbol{L} = \boldsymbol{P}\boldsymbol{L}'\boldsymbol{P}^{\top}$ *for some* $\boldsymbol{P} \in \Pi(n)$ *and we obtain the desired permutation equivariance* $\mathrm{PE}(\mathrm{EVD}(\boldsymbol{P}\boldsymbol{L}\boldsymbol{P}^{\top})) = \boldsymbol{P} \cdot \mathrm{PE}(\mathrm{EVD}(\boldsymbol{L}))$.

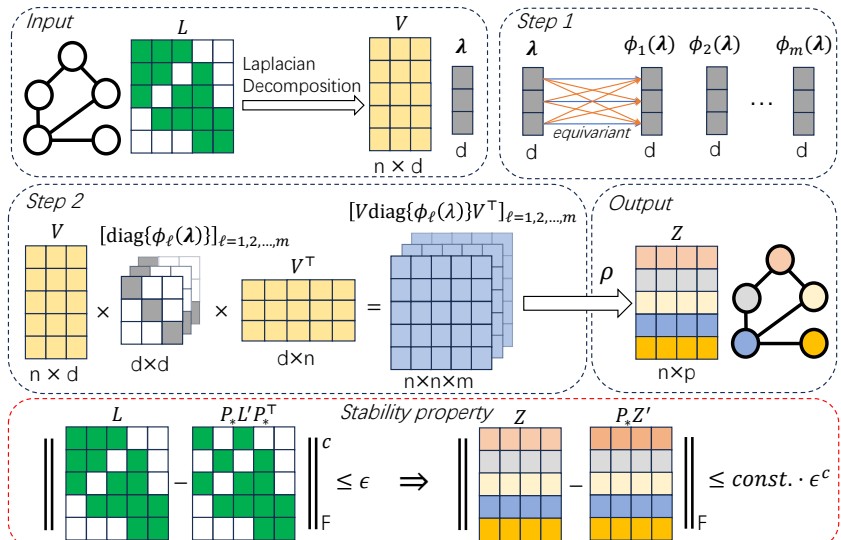

Figure 1: Illustration of the SPE architecture (first two rows) and its stability property (last row). The input graph is first decomposed into eigenvectors $\boldsymbol{V}$ and eigenvalues $\boldsymbol{\lambda}$. In step 1, a permutation equivariant $\phi_\ell$ act on $\boldsymbol{\lambda}$ to produce another vector $\phi_\ell(\boldsymbol{\lambda})$. In step 2, we compute $\boldsymbol{V}\mathrm{diag}\{\phi_\ell(\boldsymbol{\lambda})\}\boldsymbol{V}^\top$ for each $\phi_\ell$ and concatenate the results into a tensor. This tensor is input into a permutation equivariant network $\rho$ to produce final node positional encodings.

Stability is hard to achieve due to the instability of eigenvalue decomposition—a small perturbation of the Laplacian can produce completely different eigenvectors (Wang et al. (2022a), Lemma 3.4). Since positional encoding models process the eigenvectors (and eigenvalues), they naturally inherit this instability with respect to the input matrix. Indeed, as mentioned above, many existing positional encodings are not stable. The main issue is that they partition the eigenvectors by eigenvalue, which leads to instabilities. See Appendix C for a detailed discussion.

## 3.2 SPE: A POSITIONAL ENCODING WITH GUARANTEED STABILITY

To achieve stability, the key insight is to avoid a hard partition of eigensubspaces. Simultaneously, we should fully utilize the information in the eigenvalues for strong expressive power. Therefore, we propose to do a "soft partitioning" of eigenspaces by leveraging eigenvalues. Instead of treating each eigensubspace independently, we apply a weighted sum of eigenvectors in an **eigenvalue dependent** way. If done carefully, this can ensure that as two distinct eigenvalues converge—these are exactly the degenerate points creating instability—the way their respective eigenvectors are processed becomes more similar. This means that if two eigenvectors are "swapped", as happens at degenerate points, the model output does not change much. The resulting method is (illustrated in Figure 1):

$$\textbf{SPE}: \quad \mathrm{SPE}(\boldsymbol{V}, \boldsymbol{\lambda}) = \rho\big(\boldsymbol{V}\mathrm{diag}(\phi_1(\boldsymbol{\lambda}))\boldsymbol{V}^\top, \boldsymbol{V}\mathrm{diag}(\phi_2(\boldsymbol{\lambda}))\boldsymbol{V}^\top, ..., \boldsymbol{V}\mathrm{diag}(\phi_m(\boldsymbol{\lambda}))\boldsymbol{V}^\top\big), \quad (2)$$

where the input is the $d$ smallest eigenvalues $\boldsymbol{\lambda} \in \mathbb{R}^d$ and corresponding eigenvectors $\boldsymbol{V} \in \mathbb{R}^{n \times d}$, $m$ is a hyper-parameter, and $\phi_\ell : \mathbb{R}^d \to \mathbb{R}^d$ and $\rho : \mathbb{R}^{n \times n \times m} \to \mathbb{R}^{n \times p}$ are always **permutation equivariant** neural networks. Here, permutation equivariance means $\phi_\ell(\boldsymbol{P}\boldsymbol{\lambda}) = \boldsymbol{P}\phi_\ell(\boldsymbol{\lambda})$ for $\boldsymbol{P} \in \Pi(d)$ and $\rho(\boldsymbol{P}\boldsymbol{A}\boldsymbol{P}^\top) = \boldsymbol{P}\rho(\boldsymbol{A})$ for any $\boldsymbol{P} \in \Pi(n)$ and input $\boldsymbol{A}$. There are many choices of permutation equivariant networks that can be used, such as element-wise MLPs or Deep Sets (Zaheer et al., 2017) for $\phi_\ell$, and graph neural networks for $\rho$. The permutation equivariance of $\phi_\ell$ and $\rho$ ensures that SPE is *basis invariant*.

Note that in Eq. (2), the term $\boldsymbol{V}\mathrm{diag}(\phi_\ell(\boldsymbol{\lambda}))\boldsymbol{V}^\top$ looks like a spectral graph convolution operator. But they are methodologically different: SPE uses $\boldsymbol{V}\mathrm{diag}(\phi_\ell(\boldsymbol{\lambda}))\boldsymbol{V}^\top$ to construct positional encodings, which are not used as a convolution operation to process node attributes (say as $\boldsymbol{V}\mathrm{diag}(\phi_\ell(\boldsymbol{\lambda}))\boldsymbol{V}^\top\boldsymbol{X}$). Also, $\phi_\ell$'s are general permutation equivariant functions that may express the

interactions between different eigenvalues instead of elementwise polynomials on each eigenvalue separately which are commonly adopted in spectral graph convolution.

It is also worthy noticing that term $\boldsymbol{V}\mathrm{diag}(\phi_\ell(\boldsymbol{\lambda}))\boldsymbol{V}^\top$ will reduce to hard partitions of eigenvectors in the $\ell$-th eigensubspace if we let $[\phi_\ell(\boldsymbol{\lambda})]_j = \mathbb{1}(\lambda_j$ is the $l$-th smallest eigenvalue). To obtain stability, what we need is to constrain $\phi_\ell$ to continuous functions to perform a continuous "soft partition".

**Assumption 3.1.** *The key assumptions for SPE are as follows:*

- *$\phi_\ell$ and $\rho$ are permutation equivariant (see definitions after SPE Eq. (2)).*

- *$\phi_\ell$ is $K_\ell$-Lipshitz continuous: for any $\boldsymbol{\lambda}, \boldsymbol{\lambda}' \in \mathbb{R}^d$, $\|\phi_\ell(\boldsymbol{\lambda}) - \phi_\ell(\boldsymbol{\lambda}')\|_\mathsf{F} \leq K_\ell \|\boldsymbol{\lambda} - \boldsymbol{\lambda}'\|$.*

- *$\rho$ is $J$-Lipschitz continuous: for any $[\boldsymbol{A}_1, \boldsymbol{A}_2, ..., \boldsymbol{A}_m] \in \mathbb{R}^{n \times n \times m}$ and $[\boldsymbol{A}_1', \boldsymbol{A}_2', ..., \boldsymbol{A}_m'] \in \mathbb{R}^{n \times n \times m}$, $\|\rho(\boldsymbol{A}_1, \boldsymbol{A}_2, ..., \boldsymbol{A}_m) - \rho(\boldsymbol{A}_1', \boldsymbol{A}_2', ..., \boldsymbol{A}_m')\|_\mathsf{F} \leq J \sum_{l=1}^m \|\boldsymbol{A}_\ell - \boldsymbol{A}_\ell'\|_\mathsf{F}$.*

These two continuity assumptions generally hold by assuming the underlying networks have norm-bounded weights and continuous activation functions, such as ReLU. As a result, Assumption 3.1 is mild for most neural networks.

Now we are ready to present our main theorem, which states that continuity of $\phi_\ell$ and $\rho$ leads to the desired stability.

**Theorem 3.1** (Stability of SPE). *Under Assumption 3.1, **SPE is stable** with respect to the input Laplacian: for Laplacians $\boldsymbol{L}, \boldsymbol{L}'$,*

$$
\begin{aligned}
\|\mathrm{SPE}(\mathrm{EVD}(\boldsymbol{L})) - \boldsymbol{P}_*\mathrm{SPE}(\mathrm{EVD}(\boldsymbol{L}'))\|_\mathsf{F} \leq &(\alpha_1 + \alpha_2)d^{5/4}\sqrt{\|\boldsymbol{L} - \boldsymbol{P}_*\boldsymbol{L}\boldsymbol{P}_*^\top\|_\mathsf{F}} \\
&+ \left(\alpha_2\frac{d}{\gamma} + \alpha_3\right)\|\boldsymbol{L} - \boldsymbol{P}_*\boldsymbol{L}\boldsymbol{P}_*^\top\|_\mathsf{F},
\end{aligned}
\tag{3}
$$

*where the constants are $\alpha_1 = 2J\sum_{l=1}^m K_\ell$, $\alpha_2 = 4\sqrt{2}J\sum_{l=1}^m M_\ell$ and $\alpha_3 = J\sum_{l=1}^m K_\ell$. Here $M_\ell = \sup_{\boldsymbol{\lambda} \in [0,2]^d}\|\phi_\ell(\boldsymbol{\lambda})\|$ and again $\boldsymbol{P}_* = \arg\min_{\boldsymbol{P} \in \Pi(n)}\|\boldsymbol{L} - \boldsymbol{P}_*\boldsymbol{L}\boldsymbol{P}_*^\top\|_\mathsf{F}$. The eigengap $\gamma = \lambda_{d+1} - \lambda_d$ is the difference between the $(d+1)$-th and $d$-th smallest eigenvalues, and $\gamma = +\infty$ if $d = n$.*

Note that the stability of SPE is determined by both the Lipschitz constants $J, K_\ell$ and the eigengap $\gamma = \lambda_d - \lambda_{d+1}$. The dependence on $\gamma$ comes from the fact that we only choose to use $d$ eigenvectors/eigenvalues. It is inevitable as long as $d < n$, and it disappears ($\gamma = +\infty$) if we let $d = n$. This phenomenon is also observed in PEG (Wang et al. (2022a), Theorem 3.6).

## 3.3 FROM STABILITY TO OUT-OF-DISTRIBUTION GENERALIZATION

An important implication of stability is that one can characterize the domain generalization gap by the model's Lipschitz constant (Courty et al., 2017; Shen et al., 2018). Although our method satisfies Hölder continuity instead of strict Lipschitz continuity, we claim that interestingly, a **similar bound can still be obtained** for domain generalization.

We consider graph regression with domain shift: the training graphs are sampled from source domain $\boldsymbol{L} \sim \mathbb{P}_\mathcal{S}$, while the test graphs are sampled from target domain $\boldsymbol{L} \sim \mathbb{P}_\mathcal{T}$. With ground-truth function $f(\boldsymbol{L}) \in \mathbb{R}$ and a prediction model $h(\boldsymbol{L}) \in \mathbb{R}$, we are interested in the gap between in-distribution error $\varepsilon_s(h) = \mathbb{E}_{\boldsymbol{L} \sim \mathbb{P}_\mathcal{S}}|h(\boldsymbol{L}) - f(\boldsymbol{L})|$ and out-of-distribution error $\varepsilon_t(h) = \mathbb{E}_{\boldsymbol{L} \sim \mathbb{P}_\mathcal{T}}|h(\boldsymbol{L}) - f(\boldsymbol{L})|$. The following theorem states that for a base GNN equipped with SPE, we can upper bound the generalization gap in terms of the Hölder constant of SPE, the Lipschitz constant of the base GNN and the 1-Wasserstein distance between source and target distributions.

**Proposition 3.1.** *Assume Assumption 3.1 hold, and assume a base GNN model $\mathrm{GNN}(\boldsymbol{L}, \boldsymbol{X}) \in \mathbb{R}$ that is $C$-Lipschitz continuous, i.e.,*

$$
|\mathrm{GNN}(\boldsymbol{L}, \boldsymbol{X}) - \mathrm{GNN}(\boldsymbol{L}', \boldsymbol{X}')| \leq C \min_{\boldsymbol{P} \in \Pi(n)}\left(\|\boldsymbol{L} - \boldsymbol{P}\boldsymbol{L}'\boldsymbol{P}^\top\|_\mathsf{F} + \|\boldsymbol{X} - \boldsymbol{P}\boldsymbol{X}'\|_\mathsf{F}\right),
\tag{4}
$$

*for any Laplacians $\boldsymbol{L}, \boldsymbol{L}'$ and node features $\boldsymbol{X}, \boldsymbol{X}'$. Now let GNN take positional encodings as node features $\boldsymbol{X} = \mathrm{SPE}(\mathrm{EVD}(\boldsymbol{L}))$ and let the resulting prediction model be $h(\boldsymbol{L}) = \mathrm{GNN}(\boldsymbol{L}, \mathrm{SPE}(\mathrm{EVD}(\boldsymbol{L})))$. Then the domain generalization gap $\varepsilon_t(h) - \varepsilon_t(s)$ satisfies*

$$
\varepsilon_t(h) - \varepsilon_s(h) \leq 2C(1 + \alpha_2\tfrac{d}{\gamma} + \alpha_3)W(\mathbb{P}_\mathcal{S}, \mathbb{P}_\mathcal{T}) + 2Cd^{5/4}(\alpha_1 + \alpha_2)\sqrt{W(\mathbb{P}_\mathcal{S}, \mathbb{P}_\mathcal{T})},
\tag{5}
$$

*where $W(\mathbb{P}_{\mathcal{S}}, \mathbb{P}_{\mathcal{T}})$ is the 1-Wasserstein distance*[1].

### 3.4 SPE IS A UNIVERSAL BASIS INVARIANT ARCHITECTURE

SPE is a basis invariant architecture, but is it universally powerful? The next result shows that SPE is universal, meaning that *any* continuous basis invariant function can be expressed in the form of SPE (Eq. 2). To state the result, recall that $\text{SPE}(\boldsymbol{V}, \boldsymbol{\lambda}) = \rho(\boldsymbol{V} \text{diag}(\phi(\boldsymbol{\lambda}))\boldsymbol{V}^\top)$, where for brevity, we express the multiple $\phi_\ell$ channels by $\phi = (\phi_1, \ldots, \phi_m)$.

**Proposition 3.2** (Basis Universality). *SPE can universally approximate any continuous basis invariant function. That is, for any continuous $f$ for which $f(\boldsymbol{V}) = f(\boldsymbol{V}\boldsymbol{Q})$ for any eigenvalue $\boldsymbol{\lambda}$ and any $\boldsymbol{Q} \in O(\boldsymbol{\lambda})$, there exist continuous $\rho$ and $\phi$ such that $f(\boldsymbol{V}) = \rho(\boldsymbol{V} \text{diag}(\phi(\boldsymbol{\lambda}))\boldsymbol{V}^\top)$.*

Only one prior architecture, BasisNet (Lim et al., 2023), is known to have this property. However, unlike SPE, BasisNet does not have the critical stability property. Section 5 shows that this has significant empirical implications, with SPE considerably outperforming BasisNet across all evaluations. Furthermore, unlike prior analyses, we show that SPE can provably make effective use of eigenvalues: it can distinguish two input matrices with different eigenvalues using 2-layer MLP models for $\rho$ and $\phi$. In contrast, the original form of BasisNet does not use eigenvalues, though it is easy to incorporate them.

**Proposition 3.3.** *Suppose that $(\boldsymbol{V}, \boldsymbol{\lambda})$ and $(\boldsymbol{V}', \boldsymbol{\lambda}')$ are such that $\boldsymbol{V}\boldsymbol{Q} = \boldsymbol{V}'$ for some orthogonal matrix $\boldsymbol{Q} \in O(d)$ and $\boldsymbol{\lambda} \neq \boldsymbol{\lambda}'$. Then there exist 2-layer MLPs for each $\phi_\ell$ and a 2-layer MLP $\rho$, each with ReLU activations, such that $\text{SPE}(\boldsymbol{V}, \boldsymbol{\lambda}) \neq \text{SPE}(\boldsymbol{V}', \boldsymbol{\lambda}')$.*

Finally, as a concrete example of the expressivity of SPE for graph representation learning, we show that SPE is able to count graph substructures under stability guarantee.

**Proposition 3.4** (SPE can count cycles). *Assume Assumption 3.1 hold and let $\rho$ be 2-IGNs (Maron et al., 2019b). Then SPE can determine the number of 3, 4, 5 cycles of a graph.*

## 4 RELATED WORKS

**Expressive GNNs.** Since message-passing graph neural networks have been shown to be at most as powerful as the Weisfeiler-Leman test (Xu et al., 2019; Morris et al., 2019), there are many attempts to improve the expressivity of GNNs. We can classify them into three types: (1) high-order GNNs (Morris et al., 2020; Maron et al., 2019a;b); (2) subgraph GNNs (You et al., 2021; Zhang & Li, 2021; Zhao et al., 2022; Bevilacqua et al., 2022); (3) node feature augmentation (Li et al., 2020; Bouritsas et al., 2022; Barceló et al., 2021). In some senses, positional encoding can also be seen as an approach of node feature augmentation, which will be discussed below.

**Positional Encoding for GNNs.** Positional encodings aim to provide additional global positional information for nodes in graphs to make them more distinguishable and add global structural information. It thus serves as a node feature augmentation to boost the expressive power of general graph neural networks (message-passing GNNs, spectral GNNs or graph transformers). Existing positional encoding methods can be categorized into: (1) Laplacian-eigenvector-based (Dwivedi & Bresson, 2021; Kreuzer et al., 2021; Maskey et al., 2022; Dwivedi et al., 2022; Wang et al., 2022b; Lim et al., 2023; Kim et al., 2022); (2) graph-distance-based (Ying et al., 2021; You et al., 2019; Li et al., 2020); and (3) random node features (Eliasof et al., 2023). A comprehensive discussion can be found in (Rampášek et al., 2022a). Most of these methods do not consider basis invariance and stability. Notably, Wang et al. (2022a) also studies the stability of Laplacian encodings. However, their method ignores eigenvalues and thus implements a stricter symmetry that is invariant to rotations of the entire eigenspace. As a result, the "over-stability" restricts its expressive power. Bo et al. (2023) propose similar operations as $\boldsymbol{V}\text{diag}(\phi(\boldsymbol{\lambda}))\boldsymbol{V}^\top$. However they focus on a specific architecture design ($\phi$ is transformer) for spectral convolution instead of positional encodings, and do not provide any stability analysis.

**Stability and Generalization of GNNs.** The stability of neural networks is desirable as it implies better generalization (Sokolić et al., 2017; Neyshabur et al., 2017; 2018; Bartlett et al., 2017) and

---

[1]For graphs, $W(p_s, p_t) := \inf_{\pi \in \Pi(\mathbb{P}_{\mathcal{S}}, \mathbb{P}_{\mathcal{T}})} \int \min_{\boldsymbol{P} \in \Pi(n)} \left\| \boldsymbol{L} - \boldsymbol{P}\boldsymbol{L}'\boldsymbol{P}^\top \right\|_{\mathsf{F}} \pi(\boldsymbol{L}, \boldsymbol{L}')d\boldsymbol{L}d\boldsymbol{L}'$. Here $\Pi(\mathbb{P}_{\mathcal{S}}, \mathbb{P}_{\mathcal{T}})$ is the set of product distributions whose marginal distribution is $\mathbb{P}_{\mathcal{S}}$ and $\mathbb{P}_{\mathcal{T}}$ respectively.

Table 1: Test MAE results (4 random seeds) on ZINC and Alchemy.

| Dataset | PE method | #PEs | #param | Test MAE |
|---------|-----------|------|--------|----------|
| ZINC | No PE | N/A | 575k | $0.1772_{\pm 0.0040}$ |
| | PEG | 8 | 512k | $0.1444_{\pm 0.0076}$ |
| | PEG | Full | 512k | $0.1878_{\pm 0.0127}$ |
| | SignNet | 8 | 631k | $0.1034_{\pm 0.0056}$ |
| | SignNet | Full | 662k | $0.0853_{\pm 0.0026}$ |
| | BasisNet | 8 | 442k | $0.1554_{\pm 0.0068}$ |
| | BasisNet | Full | 513k | $0.1555_{\pm 0.0124}$ |
| | SPE | 8 | 635k | $0.0736_{\pm 0.0007}$ |
| | SPE | Full | 650k | $\mathbf{0.0693}_{\pm \mathbf{0.0040}}$ |
| Alchemy | No PE | N/A | 1387k | $0.112_{\pm 0.001}$ |
| | PEG | 8 | 1388k | $0.114_{\pm 0.001}$ |
| | SignNet | Full | 1668k | $0.113_{\pm 0.002}$ |
| | BasisNet | Full | 1469k | $0.110_{\pm 0.001}$ |
| | SPE | Full | 1785k | $\mathbf{0.108}_{\pm \mathbf{0.001}}$ |

transferability under domain shifts (Courty et al., 2017; Shen et al., 2018). In the context of GNNs, many works theoretically study the stability of various GNN models (Gama et al., 2020; Kenlay et al., 2020; 2021; Yehudai et al., 2020; Arghal et al., 2022; Xu et al., 2021; Chuang & Jegelka, 2022). Finally, some works try to characterize the generalization error of GNNs using VC dimension (Morris et al., 2023) or Rademacher complexity (Garg et al., 2020).

## 5 EXPERIMENTS

In this section, we use numerical experiments to verify our theory and the empirical effectiveness of our SPE. Section 5.1 tests SPE's strength as a graph positional encoder, and Section 5.2 tests the robustness of SPE to domain shifts, a key promise of stability. Section 5.3 further explores the empirical implications of stability in positional encodings. Our key finding is that there is a *trade-off* between generalization and expressive power, with less stable positional encodings fitting the training data better than their stable counterparts, but leading to worse test performance. Finally, Section 5.4 tests SPE on challenging graph substructure counting tasks that message passing graph neural networks cannot solve, and SPE significantly outperforms prior positional encoding methods.

**Datasets.** We primarily use three datasets: ZINC (Dwivedi et al., 2023), Alchemy (Chen et al., 2019) and DrugOOD (Ji et al., 2023). ZINC and Alchemy are graph regression tasks for molecular property prediction. DrugOOD is an OOD benchmark for AI drug discovery, for which we choose ligand-based affinity prediction as our classfication task (to determine if a drug is active). It considers three types of domains where distribution shifts arise: (1) Assay: which assay the data point belongs to; (2) Scaffold: the core structure of molecules; and (3) Size: molecule size. For each domain, the full dataset is divided into five partitions: the training set, the in-distribution (ID) validation/test sets, the out-of-distribution validation/test sets. These OOD partitions are expected to be distributed on the domains differently from ID partitions.

**Implementation.** We implement SPE by: $\phi_l$ either being a DeepSet (Zaheer et al., 2017), element-wise MLPs or piece-wise cubic splines (see Appendix B.1 for detailed definition); and $\rho$ being GIN (Xu et al., 2019). Note that the input of $\rho$ is $n \times n \times m$ tensors, hence we first split it into $n$ many $n \times m$ tensors, and then independently give each $n \times m$ tensors as node features to an identical GIN. Finally, we sum over the first $n$ axes to output a permutation equivariant $n \times p$ tensor.

**Baselines.** We compare SPE to other positional encoding methods including (1) No positional encodings, (2) SignNet and BasisNet (Lim et al., 2023), and (3) PEG (Wang et al., 2022a). In all cases we adopt GIN as the base GNN model. For a fair comparison, all models will have comparable budgets on the number of parameters. We also conducted an ablation study to test the effectiveness of our key component $\phi_\ell$, whose results are included in Appendix B.

Table 2: AUROC results (5 random seeds) on DrugOOD.

| Domain | PE Method | ID-Val (AUC) | ID-Test (AUC) | OOD-Val (AUC) | OOD-Test (AUC) |
|---|---|---|---|---|---|
| Assay | No PE | $92.92_{\pm 0.14}$ | $92.89_{\pm 0.14}$ | $71.02_{\pm 0.79}$ | $71.68_{\pm 1.10}$ |
| | PEG | $92.51_{\pm 0.17}$ | $92.57_{\pm 0.22}$ | $70.86_{\pm 0.44}$ | $71.98_{\pm 0.65}$ |
| | SignNet | $92.26_{\pm 0.21}$ | $92.43_{\pm 0.27}$ | $70.16_{\pm 0.56}$ | $\mathbf{72.27_{\pm 0.97}}$ |
| | BasisNet | $88.96_{\pm 1.35}$ | $89.42_{\pm 1.18}$ | $71.19_{\pm 0.72}$ | $71.66_{\pm 0.05}$ |
| | SPE | $92.84_{\pm 0.20}$ | $92.94_{\pm 0.15}$ | $71.26_{\pm 0.62}$ | $\mathbf{72.53_{\pm 0.66}}$ |
| Scaffold | No PE | $96.56_{\pm 0.10}$ | $87.95_{\pm 0.20}$ | $79.07_{\pm 0.97}$ | $68.00_{\pm 0.60}$ |
| | PEG | $95.65_{\pm 0.29}$ | $86.20_{\pm 0.14}$ | $79.17_{\pm 0.29}$ | $\mathbf{69.15_{\pm 0.75}}$ |
| | SignNet | $95.48_{\pm 0.34}$ | $86.73_{\pm 0.56}$ | $77.81_{\pm 0.70}$ | $66.43_{\pm 1.06}$ |
| | BasisNet | $85.80_{\pm 3.75}$ | $78.44_{\pm 2.45}$ | $73.36_{\pm 1.44}$ | $66.32_{\pm 5.68}$ |
| | SPE | $96.32_{\pm 0.28}$ | $88.12_{\pm 0.41}$ | $80.03_{\pm 0.58}$ | $\mathbf{69.64_{\pm 0.49}}$ |
| Size | No PE | $93.78_{\pm 0.12}$ | $93.60_{\pm 0.27}$ | $82.76_{\pm 0.04}$ | $\mathbf{66.04_{\pm 0.70}}$ |
| | PEG | $92.46_{\pm 0.35}$ | $92.67_{\pm 0.23}$ | $82.12_{\pm 0.49}$ | $\mathbf{66.01_{\pm 0.10}}$ |
| | SignNet | $93.30_{\pm 0.43}$ | $93.20_{\pm 0.39}$ | $80.67_{\pm 0.50}$ | $64.03_{\pm 0.70}$ |
| | BasisNet | $86.04_{\pm 4.01}$ | $85.51_{\pm 4.04}$ | $75.97_{\pm 1.71}$ | $60.79_{\pm 3.19}$ |
| | SPE | $92.46_{\pm 0.35}$ | $92.67_{\pm 0.23}$ | $82.12_{\pm 0.49}$ | $\mathbf{66.02_{\pm 1.00}}$ |

## 5.1 SMALL MOLECULE PROPERTY PREDICTION

We use SPE to learn graph positional encodings on ZINC and Alchemy. We let $\phi_l$ be Deepsets using only the top 8 eigenvectors (PE-8), and be element-wise MLPs when using all eigenvectors (PE-full). As before, we take $\rho$ to be a GIN.

**Results.** The test mean absolute errorx (MAEs) are shown in Table 4. On ZINC, SPE performs much better than other baselines, both when using just 8 eigenvectors (0.0736) and all eigenvectors (0.0693) . On Alchemy, we always use all eigenvectors since the graph size only ranges from 8 to 12. For Alchemy we observe no significant improvement of any PE methods over base model w/o positional encodings. But SPE still achieves the least MAE among all these models.

## 5.2 OUT-OF-DISTRIBUTION GENERALIZATION: BINDING AFFINITY PREDICTION

We study the relation between stability and out-of-distribution generalization using the DrugOOD dataset (Ji et al., 2023). We take $\phi_l$ to be element-wise MLPs and $\rho$ be GIN as usual.

**Results.** The results are shown in Table 2. All models have comparable Area Under ROC (AUC) on the ID-Test set. However, there is a big difference in OOD-Test performance on Scaffold and Size domains, with the unstable methods (SignNet and BasisNet) performing much worse than stable methods (No PE, PEG, SPE). This emphasizes the importance of stability in domain generalization. Note that this phenomenon is less obvious in the Assay domain, which is because the Assay domain represents concept (labels) shift instead of covariant (graph features) shift.

## 5.3 TRADE-OFFS BETWEEN STABILITY, GENERALIZATION AND EXPRESSIVITY

We hypothesize that stability has different effects on expressive power and generalization. Intuitively, very high stability means that outputs change very little as inputs change. Consequently, we expect highly stable models to have lower expressive power, but to generalize more reliably to new data. To test this behavior in practice we evaluate SPE on ZINC using 8 eigenvectors. We control the stability by tuning the complexity of underlying neural networks in the following two ways:

1. Directly control the Lipschitz constant of each MLP in SPE (in both $\phi_\ell$ and $\rho$) by normalizing weight matrices.

2. Let $\phi_\ell$ be a piecewise cubic spline. Increase the number of spline pieces from 1 to 6, with fewer splines corresponding to higher stability.

See Appendix B for full details. In both cases we use eight $\phi_\ell$ functions. We compute the summary statistics over different random seeds. As a measure of expressivity, we report the average training loss over the last 10 epochs on ZINC. As a measure of stability, we report the generalization gap (the difference between the test loss and the training loss) at the best validation epoch over ZINC.

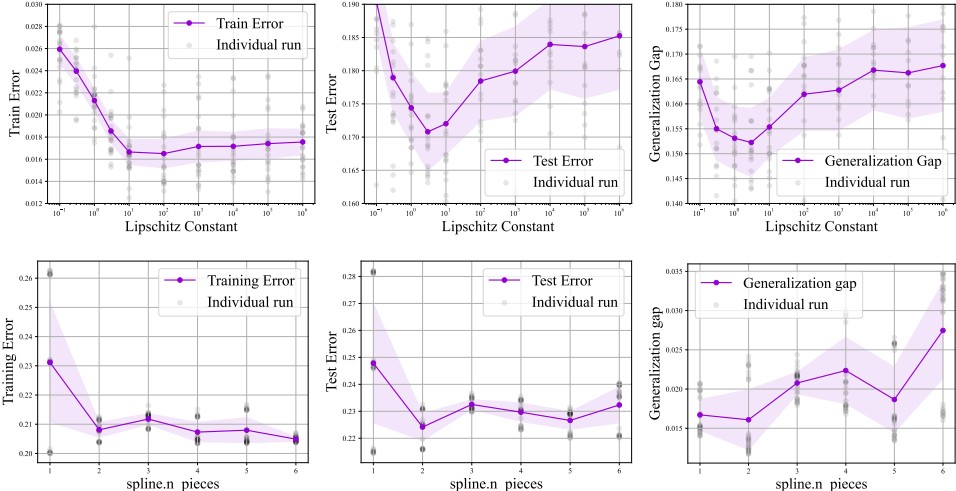

Figure 2: Training error, test error and generalization gap v.s. model complexity (stability). In the first row, we directly change the Lipschitz constant of individual MLPs; in the second row, we choose $\phi_\ell$ to be piecewise spline functions and change the number of pieces.

**Results.** In Figure 2, we show the trend of training error, test error and generalization gap as Lipschiz constant of individual MLPs (first row) or the number of spline pieces (second row) changes. We can see that as model complexity increases (stability decreases), the training error gets reduced (more expressive power) while the generalization gap grows. This justifies the important practical role of model stability for the trade-off between expressive power and generalization.

## 5.4 COUNTING GRAPH SUBSTRUCTURES

To empirically study the expressive power of SPE, we follow prior works that generate random graphs (Zhao et al., 2022; Huang et al., 2023). The dataset contains Erdős-Renyi random graphs and other random regular graphs (see Appendix M.2.1 in Chen et al. (2020)) and is randomly split into train/valid/test splitting with ratio 3:2:5. and label nodes according to the number of substructures they are part of. We aggregate the node labels to obtain the number of substructures in the overall graph and view this as a graph regression task. We let $\phi_l$ be element-wise MLPs and $\rho$ be GIN.

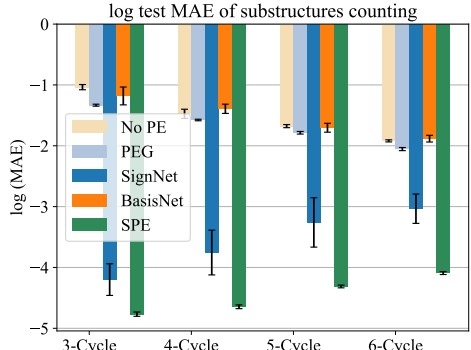

Figure 3: $\log_e$(Test MAE) over 3 random seeds on cycle counting task. Lower is better.

**Results.** Figure 3 shows that SPE significantly outperforms SignNet in counting 3,4,5 and 6-cycles. We emphasize that linear differences in log-MAE correspond to exponentially large differences in MAE. This result shows that SPE still achieves very high expressive power, whilst enjoying improved robustness to domain-shifts thanks to its stability (see Section 5.2).

## 6 CONCLUSION

We present SPE, a learnable Laplacian positional encoding that is both provably stable and expressive. Extensive experiments show the effectiveness of SPE on molecular property prediction benchmarks, the high expressivity in learning graph substructures, and the robustness as well as generalization ability under domain shifts. In the future, this technique can be extended to link prediction or other tasks involving large graphs where stability is also crucial and desired. Finally, our analysis provides a general technique for graph eigenspace stability, not just limited to domains of positional encodings and graph learning.

## ACKNOWLEDGMENTS

The authors would like to thank Derek Lim for a constructive discussion. Yinan Wang and Pan Li are partially supported by the NSF awards PHY-2117997, IIS-2239565.

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

## A    DEFERRED PROOFS

**Basic conventions.** Let $[n] = \mathbb{Z} \cap [1, n]$ and $[\![a, b]\!] = \mathbb{Z} \cap [a, b]$ denote integer intervals. Let $\Pi(n)$ be the symmetric group on $n$ elements. Unless otherwise stated, eigenvalues are counted *with multiplicity*; for example, the first and second smallest eigenvalues of $\boldsymbol{I} \in \mathbb{R}^{2 \times 2}$ are both 1. Let $\|\cdot\|$ and $\|\cdot\|_{\mathsf{F}}$ be L2-norm and Frobenius norm of matrices.

**Matrix indexing.** For any matrix $\boldsymbol{A} \in \mathbb{R}^{n \times d}$:

- Let $[\boldsymbol{A}]_{i,j} \in \mathbb{R}$ be the entry at row $i$ and column $j$

- Let $[\boldsymbol{A}]_{\langle i \rangle} \in \mathbb{R}^d$ be row $i$ represented as a column vector

- Let $[\boldsymbol{A}]_j \in \mathbb{R}^n$ be column $j$

- For any set $\mathcal{J} = \{j_1, \cdots, j_{d'}\} \subseteq [d]$, let $[\boldsymbol{A}]_{\mathcal{J}} \in \mathbb{R}^{n \times d'}$ be columns $j_1, \cdots, j_{d'}$ arranged in a matrix

- If $n = d$, let $\mathrm{diag}(\boldsymbol{A}) \in \mathbb{R}^n$ be the diagonal represented as a column vector

**Special classes of matrices.** Define the following sets:

- All $n \times n$ diagonal matrices: $\mathrm{D}(n) = \left\{ \boldsymbol{D} \in \mathbb{R}^{n \times n} : \forall i \neq j, [\boldsymbol{D}]_{i,j} = 0 \right\}$

- The orthogonal group in dimension $n$, i.e. all $n \times n$ orthogonal matrices: $\mathrm{O}(n) = \left\{ \boldsymbol{Q} \in \mathbb{R}^{n \times n} : \boldsymbol{Q}^{-1} = \boldsymbol{Q}^{\top} \right\}$

- All $n \times n$ permutation matrices: $\Pi(n) = \left\{ \boldsymbol{P} \in \{0, 1\}^{n \times n} : \boldsymbol{P}^{-1} = \boldsymbol{P}^{\top} \right\}$

- All $n \times n$ symmetric matrices: $\mathrm{S}(n) = \left\{ \boldsymbol{A} \in \mathbb{R}^{n \times n} : \boldsymbol{A} = \boldsymbol{A}^{\top} \right\}$

**Spectral graph theory.** Many properties of an undirected graph are encoded in its (normalized) *Laplacian matrix*, which is always symmetric and positive semidefinite. In this paper, we only consider connected graphs. The Laplacian matrix of a connected graph always has a zero eigenvalue with multiplicity 1 corresponding to an eigenvector of all ones, and all other eigenvalues positive. The *Laplacian eigenmap* technique uses the eigenvectors corresponding to the $d$ smallest positive eigenvalues as vertex positional encodings. We assume the $d$th and $(d+1)$th smallest positive eigenvalues of the Laplacian matrices under consideration are distinct. This motivates definitions for the following sets of matrices:

- All $n \times n$ Laplacian matrices satisfying the properties below:

$$\begin{aligned} \mathrm{L}_d(n) = \{ \boldsymbol{L} \in \mathrm{S}(n) : \boldsymbol{L} \succeq \boldsymbol{0} \;\wedge\; \mathrm{rank}(\boldsymbol{L}) = n - 1 \;\wedge\; \boldsymbol{L}\boldsymbol{1} = \boldsymbol{0} \\ \wedge\; d\text{-th and } (d+1)\text{-th smallest positive eigenvalues are distinct} \} \end{aligned} \tag{6}$$

- All $n \times n$ diagonal matrices with positive diagonal entries: $\mathrm{D}_+(n) = \{ \boldsymbol{D} \in \mathrm{D}(n) : \boldsymbol{D} \succ \boldsymbol{0} \}$

- All $n \times d$ matrices with orthonormal columns: $\mathrm{O}(n, d) = \left\{ \boldsymbol{Q} \in \mathbb{R}^{n \times d} : [\boldsymbol{Q}]_i \cdot [\boldsymbol{Q}]_j = \mathbb{1}[i = j] \right\}$

**Eigenvalues and eigenvectors.** For the first two functions below, assume the given matrix has at least $d$ positive eigenvalues.

- Let $\Lambda_d : \bigcup_{n \geq d} \mathrm{S}(n) \to \mathrm{D}_+(d)$ return a diagonal matrix containing the $d$ smallest positive eigenvalues of the given matrix, sorted in increasing order.

- Let $\mathfrak{X}_d : \bigcup_{n \geq d} \mathrm{S}(n) \to \mathrm{O}(n, d)$ return a matrix whose columns contain an unspecified set of orthonormal eigenvectors corresponding to the $d$ smallest positive eigenvalues (sorted in increasing order) of the given matrix.

- Let $\mathfrak{X}_{\langle d \rangle} : \bigcup_{n \geq d} \mathrm{S}(n) \to \mathrm{O}(n, d)$ return a matrix whose columns contain an unspecified set of orthonormal eigenvectors corresponding to the $d$ smallest eigenvalues (sorted in increasing order) of the given matrix.

- Let $\Lambda^d : \bigcup_{n \geq d} \mathrm{S}(n) \to \mathrm{D}(d)$ return a diagonal matrix containing the $d$ greatest eigenvalues of the given matrix, sorted in increasing order.

- Let $\mathcal{X}^d : \bigcup_{n \geq d} \mathrm{S}(n) \to \mathrm{O}(n, d)$ return a matrix whose columns contain an unspecified set of orthonormal eigenvectors corresponding to the $d$ greatest eigenvalues (sorted in increasing order) of the given matrix.

**Batch submatrix multiplication.** Let $\boldsymbol{A} \in \mathbb{R}^{n \times d}$ be a matrix and let $\{\mathcal{J}_k\}_{k=1}^p$ be a partition of $[d]$. For each $k \in [p]$, let $d_k = |\mathcal{J}_k|$ and let $\boldsymbol{B}_k \in \mathbb{R}^{d_k \times d_k}$ be a matrix. For notational convenience, let $\mathcal{B} = \{(\boldsymbol{B}_k, \mathcal{J}_k)\}_{k=1}^p$. Define a binary *star operator* such that

$$\boldsymbol{A} \star \mathcal{B} \in \mathbb{R}^{n \times d} \text{ is the matrix where } \forall k \in [p], [\boldsymbol{A} \star \mathcal{B}]_{\mathcal{J}_k} = [\boldsymbol{A}]_{\mathcal{J}_k} \boldsymbol{B}_k . \tag{7}$$

We primarily use batch submatrix multiplication in the context of orthogonal invariance, where an orthogonal matrix is applied to each eigenspace. For any (eigenvalue) matrix $\boldsymbol{\Lambda} \in \mathrm{D}(d)$, define

$$\mathrm{O}(\boldsymbol{\Lambda}) = \left\{ \{(\boldsymbol{Q}_k, \mathcal{J}_k)\}_{k=1}^p : \{\boldsymbol{Q}_k\}_{k=1}^p \in \prod_{k=1}^p \mathrm{O}(d_k) \right\}, \tag{8}$$

where $\mathcal{J}_k = \left\{ j \in [d] : [\mathrm{diag}(\boldsymbol{\Lambda})]_j = \sigma_k \right\}$ for each $k \in [p]$ and $\{\sigma_k\}_{k=1}^p$ are the distinct values in $\mathrm{diag}(\boldsymbol{\Lambda})$. In this context, $\mathrm{O}(\boldsymbol{\Lambda})$ is the domain of the right operand of $\star$.

## A.1 PROOF OF THEOREM 3.1

**Theorem 3.1** (Stability of SPE). *Under Assumption 3.1, **SPE is stable** with respect to the input Laplacian: for Laplacians $\boldsymbol{L}, \boldsymbol{L}'$,*

$$\|\mathrm{SPE}(\mathrm{EVD}(\boldsymbol{L})) - \boldsymbol{P}_* \mathrm{SPE}(\mathrm{EVD}(\boldsymbol{L}'))\|_\mathsf{F} \leq (\alpha_1 + \alpha_2) d^{5/4} \sqrt{\|\boldsymbol{L} - \boldsymbol{P}_* \boldsymbol{L} \boldsymbol{P}_*^\top\|_\mathsf{F}}$$
$$+ \left(\alpha_2 \frac{d}{\gamma} + \alpha_3\right) \|\boldsymbol{L} - \boldsymbol{P}_* \boldsymbol{L} \boldsymbol{P}_*^\top\|_\mathsf{F}, \tag{3}$$

*where the constants are $\alpha_1 = 2J \sum_{l=1}^m K_\ell$, $\alpha_2 = 4\sqrt{2} J \sum_{l=1}^m M_\ell$ and $\alpha_3 = J \sum_{l=1}^m K_\ell$. Here $M_\ell = \sup_{\boldsymbol{\lambda} \in [0,2]^d} \|\phi_\ell(\boldsymbol{\lambda})\|$ and again $\boldsymbol{P}_* = \arg\min_{\boldsymbol{P} \in \Pi(n)} \|\boldsymbol{L} - \boldsymbol{P}_* \boldsymbol{L} \boldsymbol{P}_*^\top\|_\mathsf{F}$. The eigengap $\gamma = \lambda_{d+1} - \lambda_d$ is the difference between the $(d+1)$-th and $d$-th smallest eigenvalues, and $\gamma = +\infty$ if $d = n$.*

*Proof.* Fix Laplacians $\boldsymbol{L}, \boldsymbol{L}' \in \mathrm{L}_d(n)$. We will show that for any permutation matrix $\boldsymbol{P} \in \Pi(n)$,

$$\|\mathrm{SPE}(\mathrm{EVD}(\boldsymbol{L})) - \boldsymbol{P} \mathrm{SPE}(\mathrm{EVD}(\boldsymbol{L}'))\|_\mathsf{F} \leq (\alpha_1 + \alpha_2) d^{\frac{5}{4}} \|\boldsymbol{L} - \boldsymbol{P} \boldsymbol{L}' \boldsymbol{P}^\top\|_\mathsf{F}^{\frac{1}{2}}$$
$$+ \left(\alpha_2 \frac{d}{\gamma} + \alpha_3\right) \|\boldsymbol{L} - \boldsymbol{P} \boldsymbol{L}' \boldsymbol{P}^\top\|_\mathsf{F}. \tag{9}$$

Fix $\boldsymbol{P} \in \Pi(n)$. For notational convenience, we denote $\mathrm{diag}\{\rho(\boldsymbol{\lambda})\}$ by $\rho(\boldsymbol{\Lambda})$ with $\boldsymbol{\Lambda} = \mathrm{diag}\{\boldsymbol{\lambda}\}$, and let $\boldsymbol{X} = \mathcal{X}_d(\boldsymbol{L})$, $\boldsymbol{\Lambda} = \Lambda_d(\boldsymbol{L})$, $\boldsymbol{X}' = \mathcal{X}_d(\boldsymbol{L}')$, and $\boldsymbol{\Lambda}' = \Lambda_d(\boldsymbol{L}')$. Then

$$\|\mathrm{SPE}(\mathrm{EVD}(\boldsymbol{L})) - \boldsymbol{P} \mathrm{SPE}(\mathrm{EVD}(\boldsymbol{L}'))\|_\mathsf{F} \tag{10}$$

$$\stackrel{(a)}{=} \|\boldsymbol{\rho}(\boldsymbol{X}\phi_1(\boldsymbol{\Lambda})\boldsymbol{X}^\top, \cdots, \boldsymbol{X}\phi_m(\boldsymbol{\Lambda})\boldsymbol{X}^\top) - \boldsymbol{P}\boldsymbol{\rho}(\boldsymbol{X}'\phi_1(\boldsymbol{\Lambda}')\boldsymbol{X}'^\top, \cdots, \boldsymbol{X}'\phi_m(\boldsymbol{\Lambda}')\boldsymbol{X}'^\top)\|_\mathsf{F} \tag{11}$$

$$\stackrel{(b)}{=} \|\boldsymbol{\rho}(\boldsymbol{X}\phi_1(\boldsymbol{\Lambda})\boldsymbol{X}^\top, \cdots, \boldsymbol{X}\phi_m(\boldsymbol{\Lambda})\boldsymbol{X}^\top) - \boldsymbol{\rho}(\boldsymbol{P}\boldsymbol{X}'\phi_1(\boldsymbol{\Lambda}')\boldsymbol{X}'^\top\boldsymbol{P}^\top, \cdots, \boldsymbol{P}\boldsymbol{X}'\phi_m(\boldsymbol{\Lambda}')\boldsymbol{X}'^\top\boldsymbol{P}^\top)\|_\mathsf{F} \tag{12}$$

$$\stackrel{(c)}{\leq} J \sum_{\ell=1}^m \|\boldsymbol{X}\phi_\ell(\boldsymbol{\Lambda})\boldsymbol{X}^\top - \boldsymbol{P}\boldsymbol{X}'\phi_\ell(\boldsymbol{\Lambda}')\boldsymbol{X}'^\top\boldsymbol{P}^\top\|_\mathsf{F} \tag{13}$$

$$\stackrel{(d)}{\leq} J \sum_{\ell=1}^m \|\boldsymbol{X}\phi_\ell(\boldsymbol{\Lambda})\boldsymbol{X}^\top - \boldsymbol{P}\boldsymbol{X}'\phi_\ell(\boldsymbol{\Lambda})\boldsymbol{X}'^\top\boldsymbol{P}^\top\|_\mathsf{F}$$
$$+ \|\boldsymbol{P}\boldsymbol{X}'\phi_\ell(\boldsymbol{\Lambda})\boldsymbol{X}'^\top\boldsymbol{P}^\top - \boldsymbol{P}\boldsymbol{X}'\phi_\ell(\boldsymbol{\Lambda}')\boldsymbol{X}'^\top\boldsymbol{P}^\top\|_\mathsf{F} \tag{14}$$

$$\stackrel{(e)}{=} J \sum_{\ell=1}^m \left\{ \underbrace{\|\boldsymbol{X}\phi_\ell(\boldsymbol{\Lambda})\boldsymbol{X}^\top - \boldsymbol{P}\boldsymbol{X}'\phi_\ell(\boldsymbol{\Lambda})\boldsymbol{X}'^\top\boldsymbol{P}^\top\|_\mathsf{F}}_{①} + \underbrace{\|\boldsymbol{X}'\phi_\ell(\boldsymbol{\Lambda})\boldsymbol{X}'^\top - \boldsymbol{X}'\phi_\ell(\boldsymbol{\Lambda}')\boldsymbol{X}'^\top\|_\mathsf{F}}_{②} \right\},$$
$$\tag{15}$$

where (a) holds by definition of SPE, (b) holds by permutation equivariance of $\boldsymbol{\rho}$, (c) holds by Lipschitz continuity of $\boldsymbol{\rho}$, (d) holds by the triangle inequality, and (e) holds by permutation invariance of Frobenius norm.

Next, we upper-bound ①. Let $\delta = \min\left\{\gamma, d^{-\frac{1}{4}}\left\|\boldsymbol{L} - \boldsymbol{P}\boldsymbol{L}'\boldsymbol{P}^\top\right\|_\mathsf{F}^{\frac{1}{2}}\right\}$. The $\delta = 0$ case is trivial, because

$$\delta = 0 \iff \boldsymbol{L} = \boldsymbol{P}\boldsymbol{L}'\boldsymbol{P}^\top \implies \mathrm{SPE}(\mathrm{EVD}(\boldsymbol{L})) = \mathrm{SPE}(\mathrm{EVD}(\boldsymbol{P}\boldsymbol{L}'\boldsymbol{P}^\top))$$
$$\overset{(a)}{\iff} \mathrm{SPE}(\mathrm{EVD}(\boldsymbol{L})) = \boldsymbol{P}\,\mathrm{SPE}(\mathrm{EVD}(\boldsymbol{L}'))\,, \tag{16}$$

where (a) holds due to permutation equivariance of SPE. Thus, assume $\delta > 0$ for the remainder of this proof. Let

$$\{j_k\}_{k=1}^{p+1} = \{1\} \cup \{j \in [d+1] : \lambda_j - \lambda_{j-1} \ge \delta\}, \quad 1 = j_1 < \cdots < j_{p+1} \overset{(a)}{=} d+1 \tag{17}$$

be the *keypoint* indices at which eigengaps are greater than or equal to $\delta$, where (a) holds because

$$\lambda_{d+1} - \lambda_d = \gamma \ge \min\left\{\gamma, d^{-\frac{1}{4}}\left\|\boldsymbol{L} - \boldsymbol{P}\boldsymbol{L}'\boldsymbol{P}^\top\right\|_\mathsf{F}^{\frac{1}{2}}\right\} = \delta\,. \tag{18}$$

For each $k \in [p]$, let $\mathcal{J}_k = \{j \in [d] : j_k \le j < j_{k+1}\}$ be a *chunk* of contiguous indices at which eigengaps are less than $\delta$, and let $d_k = |\mathcal{J}_k|$ be the size of the chunk. Define a matrix $\tilde{\boldsymbol{\Lambda}} \in \mathrm{D}(d)$ as

$$\forall k \in [p], \ \forall j \in \mathcal{J}_k, \ \left[\mathrm{diag}\left(\tilde{\boldsymbol{\Lambda}}\right)\right]_j = \lambda_{j_k}\,. \tag{19}$$

It follows that

$$① \overset{(a)}{\le} \left\|\boldsymbol{X}\phi_\ell(\boldsymbol{\Lambda})\boldsymbol{X}^\top - \boldsymbol{X}\phi_\ell\left(\tilde{\boldsymbol{\Lambda}}\right)\boldsymbol{X}^\top\right\|_\mathsf{F} + \left\|\boldsymbol{X}\phi_\ell\left(\tilde{\boldsymbol{\Lambda}}\right)\boldsymbol{X}^\top - \boldsymbol{P}\boldsymbol{X}'\phi_\ell\left(\tilde{\boldsymbol{\Lambda}}\right)\boldsymbol{X}'^\top\boldsymbol{P}^\top\right\|_\mathsf{F}$$
$$+ \left\|\boldsymbol{P}\boldsymbol{X}'\phi_\ell\left(\tilde{\boldsymbol{\Lambda}}\right)\boldsymbol{X}'^\top\boldsymbol{P}^\top - \boldsymbol{P}\boldsymbol{X}'\phi_\ell(\boldsymbol{\Lambda})\boldsymbol{X}'^\top\boldsymbol{P}^\top\right\|_\mathsf{F} \tag{20}$$

$$\overset{(b)}{=} \left\|\boldsymbol{X}\phi_\ell(\boldsymbol{\Lambda})\boldsymbol{X}^\top - \boldsymbol{X}\phi_\ell\left(\tilde{\boldsymbol{\Lambda}}\right)\boldsymbol{X}^\top\right\|_\mathsf{F} + \left\|\boldsymbol{X}\phi_\ell\left(\tilde{\boldsymbol{\Lambda}}\right)\boldsymbol{X}^\top - \boldsymbol{P}\boldsymbol{X}'\phi_\ell\left(\tilde{\boldsymbol{\Lambda}}\right)\boldsymbol{X}'^\top\boldsymbol{P}^\top\right\|_\mathsf{F}$$
$$+ \left\|\boldsymbol{X}'\phi_\ell\left(\tilde{\boldsymbol{\Lambda}}\right)\boldsymbol{X}'^\top - \boldsymbol{X}'\phi_\ell(\boldsymbol{\Lambda})\boldsymbol{X}'^\top\right\|_\mathsf{F} \tag{21}$$

$$\overset{(c)}{\le} \|\boldsymbol{X}\|^2\left\|\phi_\ell(\boldsymbol{\Lambda}) - \phi_\ell\left(\tilde{\boldsymbol{\Lambda}}\right)\right\|_\mathsf{F} + \left\|\boldsymbol{X}\phi_\ell\left(\tilde{\boldsymbol{\Lambda}}\right)\boldsymbol{X}^\top - \boldsymbol{P}\boldsymbol{X}'\phi_\ell\left(\tilde{\boldsymbol{\Lambda}}\right)\boldsymbol{X}'^\top\boldsymbol{P}^\top\right\|_\mathsf{F}$$
$$+ \|\boldsymbol{X}'\|^2\left\|\phi_\ell\left(\tilde{\boldsymbol{\Lambda}}\right) - \phi_\ell(\boldsymbol{\Lambda})\right\|_\mathsf{F} \tag{22}$$

$$\overset{(d)}{=} 2\left\|\phi_\ell(\boldsymbol{\Lambda}) - \phi_\ell\left(\tilde{\boldsymbol{\Lambda}}\right)\right\|_\mathsf{F} + \left\|\boldsymbol{X}\phi_\ell\left(\tilde{\boldsymbol{\Lambda}}\right)\boldsymbol{X}^\top - \boldsymbol{P}\boldsymbol{X}'\phi_\ell\left(\tilde{\boldsymbol{\Lambda}}\right)\boldsymbol{X}'^\top\boldsymbol{P}^\top\right\|_\mathsf{F} \tag{23}$$

$$\overset{(e)}{\le} 2K_\ell\left\|\boldsymbol{\Lambda} - \tilde{\boldsymbol{\Lambda}}\right\|_\mathsf{F} + \left\|\boldsymbol{X}\phi_\ell\left(\tilde{\boldsymbol{\Lambda}}\right)\boldsymbol{X}^\top - \boldsymbol{P}\boldsymbol{X}'\phi_\ell\left(\tilde{\boldsymbol{\Lambda}}\right)\boldsymbol{X}'^\top\boldsymbol{P}^\top\right\|_\mathsf{F} \tag{24}$$

$$\overset{(f)}{=} 2K_\ell\left\|\boldsymbol{\Lambda} - \tilde{\boldsymbol{\Lambda}}\right\|_\mathsf{F}$$
$$+ \left\|\sum_{k=1}^p [\boldsymbol{X}]_{\mathcal{J}_k}\left[\phi_\ell\left(\tilde{\boldsymbol{\Lambda}}\right)\right]_{\mathcal{J}_k,\mathcal{J}_k}[\boldsymbol{X}]_{\mathcal{J}_k}^\top - \boldsymbol{P}\left(\sum_{k=1}^p [\boldsymbol{X}']_{\mathcal{J}_k}\left[\phi_\ell\left(\tilde{\boldsymbol{\Lambda}}\right)\right]_{\mathcal{J}_k,\mathcal{J}_k}[\boldsymbol{X}']_{\mathcal{J}_k}^\top\right)\boldsymbol{P}^\top\right\|_\mathsf{F} \tag{25}$$

$$\overset{(g)}{\le} 2K_\ell\underbrace{\left\|\boldsymbol{\Lambda} - \tilde{\boldsymbol{\Lambda}}\right\|_\mathsf{F}}_{③} + \sum_{k=1}^p \underbrace{\left\|[\boldsymbol{X}]_{\mathcal{J}_k}\left[\phi_\ell\left(\tilde{\boldsymbol{\Lambda}}\right)\right]_{\mathcal{J}_k,\mathcal{J}_k}[\boldsymbol{X}]_{\mathcal{J}_k}^\top - \boldsymbol{P}[\boldsymbol{X}']_{\mathcal{J}_k}\left[\phi_\ell\left(\tilde{\boldsymbol{\Lambda}}\right)\right]_{\mathcal{J}_k,\mathcal{J}_k}[\boldsymbol{X}']_{\mathcal{J}_k}^\top\boldsymbol{P}^\top\right\|_\mathsf{F}}_{④}\,, \tag{26}$$

where (a) holds by the triangle inequality, (b) holds by permutation invariance of Frobenius norm, (c) holds by lemma A.1, (d) holds because $\boldsymbol{X}$ and $\boldsymbol{X}'$ have orthonormal columns, (e) holds by Lipschitz

continuity of $\phi_\ell$, (f) holds by block matrix algebra, and (g) holds by the triangle inequality. Next, we upper-bound ③:

$$\text{③} \overset{(a)}{=} \sqrt{\sum_{k=1}^{p} \sum_{j \in \mathcal{J}_k} (\lambda_j - \lambda_{j_k})^2} \tag{27}$$

$$\overset{(b)}{=} \sqrt{\sum_{k=1}^{p} \sum_{j \in \mathcal{J}_k} \left( \sum_{j'=j_k+1}^{j} (\lambda_{j'} - \lambda_{j'-1}) \right)^2} \tag{28}$$

$$\overset{(c)}{\le} \sqrt{\sum_{k=1}^{p} \sum_{j \in \mathcal{J}_k} \left( \sum_{j'=j_k+1}^{j} \delta \right)^2} \tag{29}$$

$$= \delta \sqrt{\sum_{k=1}^{p} \sum_{j \in \mathcal{J}_k} (j - j_k)^2} \tag{30}$$

$$\overset{(d)}{=} \delta \sqrt{\sum_{k=1}^{p} \sum_{j=1}^{d_k-1} j^2} \tag{31}$$

$$\le \delta \sqrt{\sum_{k=1}^{p} d_k^3}, \tag{32}$$

where (a) holds by definition of $\tilde{\mathbf{\Lambda}}$, (b) holds because the innermost sum in (b) telescopes, (c) holds because $\mathcal{J}_k$ is a chunk of contiguous indices at which eigengaps are less than $\delta$, and (d) holds because $\mathcal{J}_k$ is a contiguous integer interval.

Next, we upper-bound ④. By definition of $\tilde{\mathbf{\Lambda}}$, the entries $\left[ \mathrm{diag}\left( \tilde{\mathbf{\Lambda}} \right) \right]_j$ are equal for all $j \in \mathcal{J}_k$. By permutation equivariance of $\phi_\ell$, the entries $\left[ \mathrm{diag}\left( \phi_\ell\left( \tilde{\mathbf{\Lambda}} \right) \right) \right]_j$ are equal for all $j \in \mathcal{J}_k$. Thus, $\left[ \phi_\ell\left( \tilde{\mathbf{\Lambda}} \right) \right]_{\mathcal{J}_k, \mathcal{J}_k} = \mu_{\ell,k} \mathbf{I}$ for some $\mu_{\ell,k} \in \mathbb{R}$. As $\phi_\ell$ is Lipschitz continuous and defined on a bounded domain $[0, 2]^d$, it must be bounded by constant $M_\ell = \sup_{\boldsymbol{\lambda} \in [0,2]^d} \phi_\ell(\boldsymbol{\lambda})$. Then by boundedness of $\phi_\ell$,

$$|\mu_{\ell,k}| = \frac{1}{\sqrt{d_k}} \|\mu_{\ell,k} \mathbf{I}\|_{\mathsf{F}} = \frac{1}{\sqrt{d_k}} \left\| \left[ \phi_\ell\left( \tilde{\mathbf{\Lambda}} \right) \right]_{\mathcal{J}_k, \mathcal{J}_k} \right\|_{\mathsf{F}} \le \frac{1}{\sqrt{d_k}} \left\| \phi_\ell\left( \tilde{\mathbf{\Lambda}} \right) \right\|_{\mathsf{F}} \le \frac{M_\ell}{\sqrt{d_k}}. \tag{33}$$

Therefore,

$$\text{④} = \left\| [\mathbf{X}]_{\mathcal{J}_k} (\mu_{\ell,k} \mathbf{I}) [\mathbf{X}]_{\mathcal{J}_k}^{\top} - \mathbf{P} [\mathbf{X}']_{\mathcal{J}_k} (\mu_{\ell,k} \mathbf{I}) [\mathbf{X}']_{\mathcal{J}_k}^{\top} \mathbf{P}^{\top} \right\|_{\mathsf{F}} \tag{34}$$

$$= |\mu_{\ell,k}| \left\| [\mathbf{X}]_{\mathcal{J}_k} [\mathbf{X}]_{\mathcal{J}_k}^{\top} - \mathbf{P} [\mathbf{X}']_{\mathcal{J}_k} [\mathbf{X}']_{\mathcal{J}_k}^{\top} \mathbf{P}^{\top} \right\|_{\mathsf{F}} \tag{35}$$

$$\le \frac{M_\ell}{\sqrt{d_k}} \left\| [\mathbf{X}]_{\mathcal{J}_k} [\mathbf{X}]_{\mathcal{J}_k}^{\top} - \mathbf{P} [\mathbf{X}']_{\mathcal{J}_k} [\mathbf{X}']_{\mathcal{J}_k}^{\top} \mathbf{P}^{\top} \right\|_{\mathsf{F}}. \tag{36}$$

Now, we consider two cases. Case 1: $k \ge 2$ or $\lambda_1 - \lambda_0 \ge \delta$. Define the matrices

$$\mathbf{Z}_k = [\mathbf{X}]_{\mathcal{J}_k} \quad \text{and} \quad \mathbf{Z}'_k = [\mathbf{X}']_{\mathcal{J}_k}. \tag{37}$$

There exists an orthogonal matrix $\mathbf{Q}_k \in \mathrm{O}(d_k)$ such that

$$\|\mathbf{Z}_k - \mathbf{P} \mathbf{Z}'_k \mathbf{Q}_k\|_{\mathsf{F}} \overset{(a)}{=} \left\| [\mathcal{X}_d(\mathbf{L})]_{\mathcal{J}_k} - [\mathcal{X}_d(\mathbf{P} \mathbf{L}' \mathbf{P}^{\top})]_{\mathcal{J}_k} \mathbf{Q}_k \right\|_{\mathsf{F}} \tag{38}$$

$$\overset{(b)}{\le} \frac{\sqrt{8} \|\mathbf{L} - \mathbf{P} \mathbf{L}' \mathbf{P}^{\top}\|_{\mathsf{F}}}{\min \left\{ \lambda_{j_k} - \lambda_{j_k-1}, \lambda_{j_{k+1}} - \lambda_{j_{k+1}-1} \right\}} \tag{39}$$

$$\overset{(c)}{\leq} \frac{\sqrt{8} \left\| \boldsymbol{L} - \boldsymbol{P}\boldsymbol{L}'\boldsymbol{P}^\top \right\|_{\mathsf{F}}}{\delta} \,, \tag{40}$$

where (a) holds by lemmas A.2 and A.3, (b) holds by proposition A.1,[2] and (c) holds because $j_k$ and $j_{k+1}$ are keypoint indices at which eigengaps are greater than or equal to $\delta$.

Case 2: $k = 1$ and $\lambda_1 - \lambda_0 < \delta$. Define the matrices

$$\boldsymbol{Z}_1 = \left[ \tfrac{1}{\sqrt{n}}\mathbf{1} \quad [\boldsymbol{X}]_{\mathcal{J}_1} \right] \text{ and } \boldsymbol{Z}_1' = \left[ \tfrac{1}{\sqrt{n}}\mathbf{1} \quad [\boldsymbol{X}']_{\mathcal{J}_1} \right]. \tag{41}$$

There exists an orthogonal matrix $\boldsymbol{Q}_1 \in \mathrm{O}(d_1 + 1)$ such that

$$\left\| \boldsymbol{Z}_1 - \boldsymbol{P}\boldsymbol{Z}_1'\boldsymbol{Q}_1 \right\|_{\mathsf{F}} \overset{(a)}{=} \left\| \mathfrak{X}_{\langle d_1+1\rangle}(\boldsymbol{L}) - \mathfrak{X}_{\langle d_1+1\rangle}\left(\boldsymbol{P}\boldsymbol{L}'\boldsymbol{P}^\top\right)\boldsymbol{Q}_1 \right\|_{\mathsf{F}} \tag{42}$$

$$\overset{(b)}{\leq} \frac{\sqrt{8}\left\| \boldsymbol{L} - \boldsymbol{P}\boldsymbol{L}'\boldsymbol{P}^\top \right\|_{\mathsf{F}}}{\lambda_{j_2} - \lambda_{j_2-1}} \tag{43}$$

$$\overset{(c)}{\leq} \frac{\sqrt{8}\left\| \boldsymbol{L} - \boldsymbol{P}\boldsymbol{L}'\boldsymbol{P}^\top \right\|_{\mathsf{F}}}{\delta} \,, \tag{44}$$

where (a) holds by lemmas A.2 and A.3, (b) holds by proposition A.1,[3] and (c) holds because $j_2$ is a keypoint index at which the eigengap is greater than or equal to $\delta$.

Hence, in both cases,

$$\left\| [\boldsymbol{X}]_{\mathcal{J}_k} [\boldsymbol{X}]_{\mathcal{J}_k}^\top - \boldsymbol{P}[\boldsymbol{X}']_{\mathcal{J}_k} [\boldsymbol{X}']_{\mathcal{J}_k}^\top \boldsymbol{P}^\top \right\|_{\mathsf{F}} \overset{(a)}{=} \left\| \boldsymbol{Z}_k\boldsymbol{Z}_k^\top - \boldsymbol{P}\boldsymbol{Z}_k'\boldsymbol{Z}_k'^\top\boldsymbol{P}^\top \right\|_{\mathsf{F}} \tag{45}$$

$$\overset{(b)}{=} \left\| \boldsymbol{Z}_k\boldsymbol{Z}_k^\top - \boldsymbol{P}\boldsymbol{Z}_k'\boldsymbol{Q}_k\boldsymbol{Q}_k^\top\boldsymbol{Z}_k'^\top\boldsymbol{P}^\top \right\|_{\mathsf{F}} \tag{46}$$

$$\overset{(c)}{\leq} \left\| \boldsymbol{Z}_k\boldsymbol{Z}_k^\top - \boldsymbol{Z}_k\boldsymbol{Q}_k^\top\boldsymbol{Z}_k'^\top\boldsymbol{P}^\top \right\|_{\mathsf{F}} + \left\| \boldsymbol{Z}_k\boldsymbol{Q}_k^\top\boldsymbol{Z}_k'^\top\boldsymbol{P}^\top - \boldsymbol{P}\boldsymbol{Z}_k'\boldsymbol{Q}_k\boldsymbol{Q}_k^\top\boldsymbol{Z}_k'^\top\boldsymbol{P}^\top \right\|_{\mathsf{F}} \tag{47}$$

$$\overset{(d)}{\leq} \left\| \boldsymbol{Z}_k \right\| \left\| \boldsymbol{Z}_k^\top - \boldsymbol{Q}_k^\top\boldsymbol{Z}_k'^\top\boldsymbol{P}^\top \right\|_{\mathsf{F}} + \left\| \boldsymbol{Z}_k - \boldsymbol{P}\boldsymbol{Z}_k'\boldsymbol{Q}_k \right\|_{\mathsf{F}} \left\| \boldsymbol{Q}_k \right\| \left\| \boldsymbol{Z}_k' \right\| \left\| \boldsymbol{P} \right\| \tag{48}$$

$$\overset{(e)}{=} \left\| \boldsymbol{Z}_k^\top - \boldsymbol{Q}_k^\top\boldsymbol{Z}_k'^\top\boldsymbol{P}^\top \right\|_{\mathsf{F}} + \left\| \boldsymbol{Z}_k - \boldsymbol{P}\boldsymbol{Z}_k'\boldsymbol{Q}_k \right\|_{\mathsf{F}} \tag{49}$$

$$\overset{(f)}{=} 2\left\| \boldsymbol{Z}_k - \boldsymbol{P}\boldsymbol{Z}_k'\boldsymbol{Q}_k \right\|_{\mathsf{F}} \tag{50}$$

$$\overset{(g)}{\leq} \frac{\sqrt{32}\left\| \boldsymbol{L} - \boldsymbol{P}\boldsymbol{L}'\boldsymbol{P}^\top \right\|_{\mathsf{F}}}{\delta} \,, \tag{51}$$

where (a) holds in Case 2 because

$$[\boldsymbol{X}]_{\mathcal{J}_k} [\boldsymbol{X}]_{\mathcal{J}_k}^\top - \boldsymbol{P}[\boldsymbol{X}']_{\mathcal{J}_k} [\boldsymbol{X}']_{\mathcal{J}_k}^\top \boldsymbol{P}^\top \tag{52}$$

$$= \frac{1}{n}\mathbf{1}\mathbf{1}^\top + [\boldsymbol{X}]_{\mathcal{J}_k} [\boldsymbol{X}]_{\mathcal{J}_k}^\top - \frac{1}{n}\mathbf{1}\mathbf{1}^\top - \boldsymbol{P}[\boldsymbol{X}']_{\mathcal{J}_k} [\boldsymbol{X}']_{\mathcal{J}_k}^\top \boldsymbol{P}^\top \tag{53}$$

$$= \frac{1}{n}\mathbf{1}\mathbf{1}^\top + [\boldsymbol{X}]_{\mathcal{J}_k} [\boldsymbol{X}]_{\mathcal{J}_k}^\top - \boldsymbol{P}\left( \frac{1}{n}\mathbf{1}\mathbf{1}^\top + [\boldsymbol{X}']_{\mathcal{J}_k} [\boldsymbol{X}']_{\mathcal{J}_k}^\top \right)\boldsymbol{P}^\top \tag{54}$$

$$= \left[ \tfrac{1}{\sqrt{n}}\mathbf{1} \quad [\boldsymbol{X}]_{\mathcal{J}_k} \right] \begin{bmatrix} \tfrac{1}{\sqrt{n}}\mathbf{1}^\top \\ [\boldsymbol{X}]_{\mathcal{J}_k}^\top \end{bmatrix} - \boldsymbol{P}\left[ \tfrac{1}{\sqrt{n}}\mathbf{1} \quad [\boldsymbol{X}']_{\mathcal{J}_k} \right] \begin{bmatrix} \tfrac{1}{\sqrt{n}}\mathbf{1}^\top \\ [\boldsymbol{X}']_{\mathcal{J}_k}^\top \end{bmatrix} \boldsymbol{P}^\top \tag{55}$$

$$= \boldsymbol{Z}_k\boldsymbol{Z}_k^\top - \boldsymbol{P}\boldsymbol{Z}_k'\boldsymbol{Z}_k'^\top\boldsymbol{P}^\top \,, \tag{56}$$

(b) holds because $\boldsymbol{Q}_k\boldsymbol{Q}_k^\top = \boldsymbol{I}$, (c) holds by the triangle inequality, (d) holds by lemma A.1, (e) holds because $\boldsymbol{Z}_k$ and $\boldsymbol{Z}_k'$ have orthonormal columns and $\boldsymbol{Q}_k$ and $\boldsymbol{P}$ are orthogonal, (f) holds because Frobenius norm is invariant to matrix transpose, and (g) holds by substituting in eqs. (40) and (44). Combining these results,

$$④ \leq \frac{\sqrt{32}\, M_\ell \left\| \boldsymbol{L} - \boldsymbol{P}\boldsymbol{L}'\boldsymbol{P}^\top \right\|_{\mathsf{F}}}{\sqrt{d_k}\, \delta} \,. \tag{57}$$

---

[2]We can apply proposition A.1 because $\mathfrak{X}_d$ extracts the same contiguous interval of eigenvalue indices for any matrix in $\mathrm{L}_d(n)$, and $\boldsymbol{P}\boldsymbol{L}'\boldsymbol{P}^\top \in \mathrm{L}_d(n)$ by lemmas A.2 and A.3.

[3]We can apply proposition A.1 because $\mathfrak{X}_{\langle d_1+1\rangle}$ extracts the same contiguous interval of eigenvalue indices for any matrix.

Next, we upper-bound ②:

$$② \overset{(a)}{\leq} \|\boldsymbol{X}'\|^2 \, \|\boldsymbol{\phi}_\ell(\boldsymbol{\Lambda}) - \boldsymbol{\phi}_\ell(\boldsymbol{\Lambda}')\|_{\mathsf{F}} \tag{58}$$

$$\overset{(b)}{=} \|\boldsymbol{\phi}_\ell(\boldsymbol{\Lambda}) - \boldsymbol{\phi}_\ell(\boldsymbol{\Lambda}')\|_{\mathsf{F}} \tag{59}$$

$$\overset{(c)}{\leq} K_\ell \, \|\boldsymbol{\Lambda} - \boldsymbol{\Lambda}'\|_{\mathsf{F}} \tag{60}$$

$$\overset{(d)}{=} K_\ell \sqrt{\sum_{j=1}^{d} \left(\lambda_j - \lambda_j'\right)^2} \tag{61}$$

$$\overset{(e)}{\leq} K_\ell \left\|\boldsymbol{L} - \boldsymbol{P}\boldsymbol{L}'\boldsymbol{P}^\top\right\|_{\mathsf{F}}, \tag{62}$$

where (a) holds by lemma A.1, (b) holds because $\boldsymbol{X}'$ has orthonormal columns, (c) holds by Lipschitz continuity of $\boldsymbol{\phi}_\ell$, the notation $\lambda_j'$ in (d) is the $j$th smallest positive eigenvalue of $\boldsymbol{L}'$, and (e) holds by proposition A.3 and lemma A.3.

Combining our results above,

$$\|\mathrm{SPE}_{n,d}(\boldsymbol{L}) - \boldsymbol{P}\,\mathrm{SPE}_{n,d}(\boldsymbol{L}')\|_{\mathsf{F}} \tag{63}$$

$$\overset{(a)}{\leq} J \sum_{\ell=1}^{m} \left\{ 2K_\ell \delta \sqrt{\sum_{k=1}^{p} d_k^3} + \sum_{k=1}^{p} \frac{4\sqrt{2}\,M_\ell \left\|\boldsymbol{L} - \boldsymbol{P}\boldsymbol{L}'\boldsymbol{P}^\top\right\|_{\mathsf{F}}}{\sqrt{d_k}\,\delta} + K_\ell \left\|\boldsymbol{L} - \boldsymbol{P}\boldsymbol{L}'\boldsymbol{P}^\top\right\|_{\mathsf{F}} \right\} \tag{64}$$

$$\overset{(b)}{\leq} J \sum_{\ell=1}^{m} \left\{ 2K_\ell d^{\frac{3}{2}} \delta + 4\sqrt{2}\,M_\ell d \frac{\left\|\boldsymbol{L} - \boldsymbol{P}\boldsymbol{L}'\boldsymbol{P}^\top\right\|_{\mathsf{F}}}{\delta} + K_\ell \left\|\boldsymbol{L} - \boldsymbol{P}\boldsymbol{L}'\boldsymbol{P}^\top\right\|_{\mathsf{F}} \right\} \tag{65}$$

$$\overset{(c)}{=} \alpha_1 d^{\frac{3}{2}} \delta + \alpha_2 d \frac{\left\|\boldsymbol{L} - \boldsymbol{P}\boldsymbol{L}'\boldsymbol{P}^\top\right\|_{\mathsf{F}}}{\delta} + \alpha_3 \left\|\boldsymbol{L} - \boldsymbol{P}\boldsymbol{L}'\boldsymbol{P}^\top\right\|_{\mathsf{F}} \tag{66}$$

$$\overset{(d)}{\leq} (\alpha_1 + \alpha_2)\,d^{\frac{5}{4}} \left\|\boldsymbol{L} - \boldsymbol{P}\boldsymbol{L}'\boldsymbol{P}^\top\right\|_{\mathsf{F}}^{\frac{1}{2}} + \left(\alpha_2 \frac{d}{\gamma} + \alpha_3\right) \left\|\boldsymbol{L} - \boldsymbol{P}\boldsymbol{L}'\boldsymbol{P}^\top\right\|_{\mathsf{F}} \tag{67}$$

as desired, where (a) holds by substituting in ①-④, (b) holds because $\sum_{k=1}^{p} d_k^3 \leq \left(\sum_{k=1}^{p} d_k\right)^3 = d^3$ and $\sum_{k=1}^{p} \frac{1}{\sqrt{d_k}} \leq p \leq d$, (c) holds by the definition of $\alpha_1$ through $\alpha_3$, and (d) holds because

$$\delta \leq d^{-\frac{1}{4}} \left\|\boldsymbol{L} - \boldsymbol{P}\boldsymbol{L}'\boldsymbol{P}^\top\right\|_{\mathsf{F}}^{\frac{1}{2}}, \tag{68}$$

$$\frac{\left\|\boldsymbol{L} - \boldsymbol{P}\boldsymbol{L}'\boldsymbol{P}^\top\right\|_{\mathsf{F}}}{\delta} \leq \frac{\left\|\boldsymbol{L} - \boldsymbol{P}\boldsymbol{L}'\boldsymbol{P}^\top\right\|_{\mathsf{F}}}{\gamma} + d^{\frac{1}{4}} \left\|\boldsymbol{L} - \boldsymbol{P}\boldsymbol{L}'\boldsymbol{P}^\top\right\|_{\mathsf{F}}^{\frac{1}{2}}. \tag{69}$$

$$\square$$

## A.2 PROOF OF PROPOSITION 3.1

**Proposition 3.1.** *Assume Assumption 3.1 hold, and assume a base GNN model* $\mathrm{GNN}(\boldsymbol{L}, \boldsymbol{X}) \in \mathbb{R}$ *that is $C$-Lipschitz continuous, i.e.,*

$$|\mathrm{GNN}(\boldsymbol{L}, \boldsymbol{X}) - \mathrm{GNN}(\boldsymbol{L}', \boldsymbol{X}')| \leq C \min_{\boldsymbol{P} \in \Pi(n)} \left(\left\|\boldsymbol{L} - \boldsymbol{P}\boldsymbol{L}'\boldsymbol{P}^\top\right\|_{\mathsf{F}} + \|\boldsymbol{X} - \boldsymbol{P}\boldsymbol{X}'\|_{\mathsf{F}}\right), \tag{4}$$

*for any Laplacians $\boldsymbol{L}, \boldsymbol{L}'$ and node features $\boldsymbol{X}, \boldsymbol{X}'$. Now let GNN take positional encodings as node features $\boldsymbol{X} = \mathrm{SPE}(\mathrm{EVD}(\boldsymbol{L}))$ and let the resulting prediction model be $h(\boldsymbol{L}) = \mathrm{GNN}(\boldsymbol{L}, \mathrm{SPE}(\mathrm{EVD}(\boldsymbol{L})))$. Then the domain generalization gap $\varepsilon_t(h) - \varepsilon_t(s)$ satisfies*

$$\varepsilon_t(h) - \varepsilon_s(h) \leq 2C(1 + \alpha_2 \tfrac{d}{\gamma} + \alpha_3) W(\mathbb{P}_{\mathcal{S}}, \mathbb{P}_{\mathcal{T}}) + 2C d^{5/4}(\alpha_1 + \alpha_2)\sqrt{W(\mathbb{P}_{\mathcal{S}}, \mathbb{P}_{\mathcal{T}})}, \tag{5}$$

*where $W(\mathbb{P}_{\mathcal{S}}, \mathbb{P}_{\mathcal{T}})$ is the 1-Wasserstein distance[4].*

---

[4]For graphs, $W(p_s, p_t) := \inf_{\pi \in \Pi(\mathbb{P}_{\mathcal{S}}, \mathbb{P}_{\mathcal{T}})} \int \min_{\boldsymbol{P} \in \Pi(n)} \left\|\boldsymbol{L} - \boldsymbol{P}\boldsymbol{L}'\boldsymbol{P}^\top\right\|_{\mathsf{F}} \pi(\boldsymbol{L}, \boldsymbol{L}') d\boldsymbol{L} d\boldsymbol{L}'$. Here $\Pi(\mathbb{P}_{\mathcal{S}}, \mathbb{P}_{\mathcal{T}})$ is the set of product distributions whose marginal distribution is $\mathbb{P}_{\mathcal{S}}$ and $\mathbb{P}_{\mathcal{T}}$ respectively.

*Proof.* The proof goes in two steps. The first step shows that a Lipschitz continuous base GNN with a Hölder continuity SPE yields an overall Hölder continuity predictive model. The second step shows that this Hölder continuous predictive model has a bounded generalization gap under domain shift.

**Step 1:** Suppose base GNN model $\text{GNN}(\boldsymbol{L}, \boldsymbol{X})$ is $C$-Lipschitz and SPE method $\text{SPE}(\boldsymbol{L})$ satifies Theorem 3.1. Let our predictive model be $h(\boldsymbol{L}) = \text{GNN}(\boldsymbol{L}, \text{SPE}(\text{EVD}(\boldsymbol{L}))) \in \mathbb{R}$. Then for any Laplacians $\boldsymbol{L}, \boldsymbol{L}' \in$ and any permutation $\boldsymbol{P} \in \Pi(n)$ we have

$$|h(\boldsymbol{L}) - h(\boldsymbol{L}')| = |\text{GNN}(\boldsymbol{L}, \text{SPE}(\text{EVD}(\boldsymbol{L}))) - \text{GNN}(\boldsymbol{L}', \text{SPE}(\text{EVD}(\boldsymbol{L}')))| \tag{70}$$

$$\overset{(a)}{\leq} C\left(\|\boldsymbol{L} - \boldsymbol{P}\boldsymbol{L}'\boldsymbol{P}\|_{\mathsf{F}} + \|\text{SPE}(\text{EVD}(\boldsymbol{L})) - \boldsymbol{P}\text{SPE}(\text{EVD}(\boldsymbol{L}'))\|_{\mathsf{F}}\right) \tag{71}$$

$$\overset{(b)}{\leq} C\left(1 + \alpha_2 \frac{d}{\gamma} + \alpha_3\right) \left\|\boldsymbol{L} - \boldsymbol{P}\boldsymbol{L}'\boldsymbol{P}^\top\right\|_{\mathsf{F}} + C\left(\alpha_1 + \alpha_2\right) d^{5/4} \left\|\boldsymbol{L} - \boldsymbol{P}\boldsymbol{L}'\boldsymbol{P}^\top\right\|_{\mathsf{F}}^{1/2}, \tag{72}$$

$$:= C_1 \left\|\boldsymbol{L} - \boldsymbol{P}\boldsymbol{L}'\boldsymbol{P}^\top\right\|_{\mathsf{F}} + C_2 \left\|\boldsymbol{L} - \boldsymbol{P}\boldsymbol{L}'\boldsymbol{P}^\top\right\|_{\mathsf{F}}^{1/2}, \tag{73}$$

where (a) holds by continuity assumption of base GNN, and (b) holds by the stability result Theorem 3.1.

**Step 2:** Suppose the ground-truth function $h^*$ lies in our hypothesis space (thus also satisfies eq. (73)). The absolute risk on source and target domain are defined $\varepsilon_s(h) = \mathbb{E}_{\boldsymbol{L}\sim\mathbb{P}_S}|h(\boldsymbol{L}) - h^*(\boldsymbol{L})|$ and $\varepsilon_t(h) = \mathbb{E}_{\boldsymbol{L}\sim\mathbb{P}_T}|h(\boldsymbol{L}) - h^*(\boldsymbol{L})|$ respectively. Note that function $f(\boldsymbol{L}) = |h(\boldsymbol{L}) - h^*(\boldsymbol{L})|$ is also Hölder continuous but with two times larger Hölder constant. This is because

$$|h(\boldsymbol{L}) - h^*(\boldsymbol{L})| \leq |h(\boldsymbol{L}) - h(\boldsymbol{L}')| + |h(\boldsymbol{L}') - h^*(\boldsymbol{L})| \quad \text{(triangle's inequality for arbitrary } \boldsymbol{L}') \tag{74}$$

$$\leq |h(\boldsymbol{L}) - h(\boldsymbol{L}')| + |h(\boldsymbol{L}') - h^*(\boldsymbol{L}')| + |h^*(\boldsymbol{L}) - h^*(\boldsymbol{L}')| \quad \text{(triangle's inequality)}, \tag{75}$$

and thus for arbitrary $\boldsymbol{L}, \boldsymbol{L}'$,

$$f(\boldsymbol{L}) - f(\boldsymbol{L}') = |h(\boldsymbol{L}) - h^*(\boldsymbol{L})| - |h(\boldsymbol{L}') - h^*(\boldsymbol{L}')| \leq |h(\boldsymbol{L}) - h(\boldsymbol{L}')| + |h^*(\boldsymbol{L}) - h^*(\boldsymbol{L}')| \tag{76}$$

$$\leq 2C_1 \left\|\boldsymbol{L} - \boldsymbol{P}\boldsymbol{L}'\boldsymbol{P}^\top\right\|_{\mathsf{F}} + 2C_2 \left\|\boldsymbol{L} - \boldsymbol{P}\boldsymbol{L}'\boldsymbol{P}^\top\right\|_{\mathsf{F}}^{1/2}. \tag{77}$$

We can show the same bound for $f(\boldsymbol{L}') - f(\boldsymbol{L})$. Thus $f$ is Hölder continuous with constants $2C_1, 2C_2$, and we denote such property by $\|f\|_H \leq (2C_1, 2C_2)$ for notation convenience.

An upper bound of generalization gap $\varepsilon_t(h) - \varepsilon_s(h)$ can be obtained:

$$\varepsilon_t(h) - \varepsilon_s(h) = \mathbb{E}_{\boldsymbol{L}\sim\mathbb{P}_T}|h(\boldsymbol{L}) - h^*(\boldsymbol{L})| - \mathbb{E}_{\boldsymbol{L}\sim\mathbb{P}_S}|h(\boldsymbol{L}) - h^*(\boldsymbol{L})| \tag{78}$$

$$\overset{(a)}{\leq} \sup_{\|f\|_H \leq (C_1, C_2)} \mathbb{E}_{\boldsymbol{L}\sim\mathbb{P}_T} f(\boldsymbol{L}) - \mathbb{E}_{\boldsymbol{L}\sim\mathbb{P}_S} f(\boldsymbol{L}) \tag{79}$$

$$= \sup_{\|f\|_H \leq (C_1, C_2)} \int f(\boldsymbol{L})(\mathbb{P}_T(\boldsymbol{L}) - \mathbb{P}_S(\boldsymbol{L}))\mathrm{d}\boldsymbol{L} \tag{80}$$

$$\overset{(b)}{=} \inf_{\pi \in \Pi(n)(\mathbb{P}_T, \mathbb{P}_S)} \sup_{\|f\|_H \leq (C_1, C_2)} \int (f(\boldsymbol{L}) - f(\boldsymbol{L}'))\pi(\boldsymbol{L}, \boldsymbol{L}')\mathrm{d}\boldsymbol{L}\mathrm{d}\boldsymbol{L}' \tag{81}$$

where (a) holds because $\||h(\boldsymbol{L}) - h^*(\boldsymbol{L})|\|_H \leq (C_1, C_2)$, and (b) holds because of the definition of product distribution

$$\Pi(\mathbb{P}_T, \mathbb{P}_S) = \left\{\pi : \int \pi(\boldsymbol{L}, \boldsymbol{L}')\mathrm{d}\boldsymbol{L}' = \mathbb{P}_T(\boldsymbol{L}) \wedge \int \pi(\boldsymbol{L}, \boldsymbol{L}')\mathrm{d}\boldsymbol{L} = \mathbb{P}_S(\boldsymbol{L})\right\}.$$

Notice the integral can be further upper bounded using Hölder continuity of $f$:

$$\sup_{\|f\|_H \leq (C_1, C_2)} \int (f(\boldsymbol{L}) - f(\boldsymbol{L}'))\pi(\boldsymbol{L}, \boldsymbol{L}')\mathrm{d}\boldsymbol{L}\mathrm{d}\boldsymbol{L}'$$

$$\leq \int \left(2C_1 \min_{\boldsymbol{P}\in\Pi(n)} \left\|\boldsymbol{L} - \boldsymbol{P}\boldsymbol{L}'\boldsymbol{P}^\top\right\|_{\mathsf{F}} + 2C_2 \min_{\boldsymbol{P}\in\Pi(n)} \left\|\boldsymbol{L} - \boldsymbol{P}\boldsymbol{L}'\boldsymbol{P}^\top\right\|_{\mathsf{F}}^{1/2}\right) \pi(\boldsymbol{L}, \boldsymbol{L}')\mathrm{d}\boldsymbol{L}\mathrm{d}\boldsymbol{L}'. \tag{82}$$

Let us define the Wasserstein distance of $\mathbb{P}_\mathcal{T}$ and $\mathbb{P}_\mathcal{S}$ be

$$W(\mathbb{P}_\mathcal{T}, \mathbb{P}_\mathcal{S}) = \inf_{\pi \in \Pi(\mathbb{P}_\mathcal{T}, \mathbb{P}_\mathcal{S})} \int \min_{\boldsymbol{P} \in \Pi(n)} \left\| \boldsymbol{L} - \boldsymbol{P} \boldsymbol{L}' \boldsymbol{P}^\top \right\|_\mathsf{F} \pi(\boldsymbol{L}, \boldsymbol{L}') \mathrm{d}\boldsymbol{L} \mathrm{d}\boldsymbol{L}'. \tag{83}$$

Then plugging eqs. (82, 83) into (81) yields the desired result

$$\varepsilon_t(h) - \varepsilon_s(h) \leq 2C_1 W(\mathbb{P}_\mathcal{T}, \mathbb{P}_\mathcal{S}) + 2C_2 \inf_{\pi \in \Pi(\mathbb{P}_\mathcal{T}, \mathbb{P}_\mathcal{S})} \int \min_{\boldsymbol{P} \in \Pi(n)} \left\| \boldsymbol{L} - \boldsymbol{P} \boldsymbol{L}' \boldsymbol{P}^\top \right\|_\mathsf{F}^{1/2} \pi(\boldsymbol{L}, \boldsymbol{L}') \mathrm{d}\boldsymbol{L} \mathrm{d}\boldsymbol{L}' \tag{84}$$

$$\overset{(a)}{\leq} 2C_1 W(\mathbb{P}_\mathcal{T}, \mathbb{P}_\mathcal{S}) + 2C_2 \inf_{\pi \in \Pi(\mathbb{P}_\mathcal{T}, \mathbb{P}_\mathcal{S})} \left( \int \min_{\boldsymbol{P} \in \Pi(n)} \left\| \boldsymbol{L} - \boldsymbol{P} \boldsymbol{L}' \boldsymbol{P}^\top \right\|_\mathsf{F} \pi(\boldsymbol{L}, \boldsymbol{L}') \mathrm{d}\boldsymbol{L} \mathrm{d}\boldsymbol{L}' \right)^{1/2} \tag{85}$$

$$= 2C_1 W(\mathbb{P}_\mathcal{T}, \mathbb{P}_\mathcal{S}) + 2C_2 W^{1/2}(\mathbb{P}_\mathcal{T}, \mathbb{P}_\mathcal{S}), \tag{86}$$

where (a) holds due to the concavity of sqrt root function. $\square$

## A.3   PROOF OF PROPOSITION 3.2

**Proposition 3.2** (Basis Universality). *SPE can universally approximate any continuous basis invariant function. That is, for any continuous $f$ for which $f(\boldsymbol{V}) = f(\boldsymbol{V}\boldsymbol{Q})$ for any eigenvalue $\boldsymbol{\lambda}$ and any $\boldsymbol{Q} \in O(\boldsymbol{\lambda})$, there exist continuous $\rho$ and $\phi$ such that $f(\boldsymbol{V}) = \rho(\boldsymbol{V} \mathrm{diag}(\phi(\boldsymbol{\lambda})) \boldsymbol{V}^\top)$.*

*Proof.* In the proof we are going to show basis universality by expressing BasisNet. Fix eigenvalues $\boldsymbol{\lambda} \in \mathbb{R}^d$. Let $\tilde{\boldsymbol{\lambda}} \in \mathbb{R}^L$ be a sorting of eigenvalues without repetition, i.e., $\tilde{\boldsymbol{\lambda}}_i = $ $i$-th smallest eigenvalues. Assume $m \geq d$. For eigenvectors $\boldsymbol{V}$, let $\mathcal{I}_\ell \subset [d]$ be indices of $\ell$-th eigensubspaces. Recall that BasisNet is of the following form:

$$\mathrm{BasisNet}(\boldsymbol{V}, \boldsymbol{\lambda}) = \rho^{(B)} \left( \phi_1^{(B)}(\boldsymbol{V}_\mathcal{I} \boldsymbol{V}_{\mathcal{I}_1}^\top), ..., \phi_L^{(B)}(\boldsymbol{V}_{\mathcal{I}_L} \boldsymbol{V}_{\mathcal{I}_L}^\top) \right), \tag{87}$$

where $L$ is number of eigensubspaces.

For SPE, let us construct the following $\phi_\ell$:

$$[\phi_\ell(\boldsymbol{x})]_i = \begin{cases} 1, & \text{if } x_i = \tilde{\lambda}_\ell, \\[2mm] 1 - \dfrac{x_i - \tilde{\lambda}_\ell}{\tilde{\lambda}_{\ell-1} - \tilde{\lambda}_\ell}, & \text{if } x_i \in (\tilde{\lambda}_l, \tilde{\lambda}_{l+1}), \\[2mm] \dfrac{x_i - \tilde{\lambda}_{\ell-1}}{x_i - \tilde{\lambda}_l}, & \text{if } x_i \in (\tilde{\lambda}_{\ell-1}, \tilde{\lambda}_\ell), \\[2mm] 0, & \text{otherwise.} \end{cases} \tag{88}$$

Note that this is both Lipschitz continuous with Lipschitz constant $1/\min_\ell(\tilde{\lambda}_{\ell+1} - \tilde{\lambda}_\ell)$, and permutation equivariant (since it is elementwise). Now we have $\boldsymbol{V} \mathrm{diag}\{\phi_\ell(\boldsymbol{\lambda})\} \boldsymbol{V}^\top = \boldsymbol{V}_{\mathcal{I}_\ell} \boldsymbol{V}_{\mathcal{I}_\ell}^\top$, since $\phi_\ell(\boldsymbol{\lambda})$ is either 1 (when $\lambda_i = \tilde{\lambda}_\ell$) or 0 otherwise. For $\ell > L$, we let $\phi_\ell = 0$ by default. Then simply let $\rho$ be:

$$\rho(\boldsymbol{A}_1, ..., \boldsymbol{A}_m) = \rho^{(S)} \left( \phi_1^{(S)}(\boldsymbol{A}_1), ..., \phi_m^{(S)}(\boldsymbol{A}_m) \right) = \rho^{(B)} \left( \phi_1^{(B)}(\boldsymbol{A}_1), ..., \phi_L^{(B)}(\boldsymbol{A}_L) \right). \tag{89}$$

Here $\phi_\ell^{(S)}(\boldsymbol{A}_\ell) = \phi_\ell^{(B)}(\boldsymbol{A}_\ell)$ for $\boldsymbol{A}_\ell \neq 0$ and WLOG $\phi_\ell^{(S)}(\boldsymbol{A}_\ell) = 0$ if $\boldsymbol{A}_\ell = 0$. And $\rho^{(S)}$ is a function that first ignores 0 matrices and mimic $\rho^{(B)}$. Therefore,

$$\begin{aligned} \mathrm{SPE}(\boldsymbol{V}, \boldsymbol{\lambda}) &= \rho(\boldsymbol{V} \mathrm{diag}\{\phi_1(\boldsymbol{\lambda})\} \boldsymbol{V}^\top, ..., \boldsymbol{V} \mathrm{diag}\{\phi_m(\boldsymbol{\lambda})\} \boldsymbol{V}^\top) \\ &= \rho^{(S)} \left( \phi_1^{(S)}(\boldsymbol{V}_{\mathcal{I}_1} \boldsymbol{V}_{\mathcal{I}_1}^\top), ..., \phi_m^{(S)}(\boldsymbol{V}_{\mathcal{I}_m} \boldsymbol{V}_{\mathcal{I}_m}^\top) \right) \\ &= \mathrm{BasisNet}(\boldsymbol{V}, \boldsymbol{\lambda}). \end{aligned} \tag{90}$$

Since BasisNet universally approximates all continuous basis invariant function, so can SPE. $\square$

## A.4 Proof of Proposition 3.4

**Proposition 3.4** (SPE can count cycles). *Assume Assumption 3.1 hold and let $\rho$ be 2-IGNs (Maron et al., 2019b). Then SPE can determine the number of 3, 4, 5 cycles of a graph.*

*Proof.* Note that from Lim et al. (2023), Theorem 3 we know that BasisNet can count 3, 4, 5 cycles. One way to let SPE count cycles is to approximate BasisNet first and round the approximate error, thanks to the discrete nature of cycle counting. The key observation is that the implementation of BasisNet to count cycles is a special case of SPE:

$$\#\text{cycles of each node} = \text{BasisNet}(\boldsymbol{V}, \boldsymbol{\lambda}) = \rho(\boldsymbol{V}\text{diag}\{\hat{\phi}_1(\boldsymbol{\lambda})\}\boldsymbol{V}^\top, ..., \boldsymbol{V}\text{diag}\{\hat{\phi}_m(\boldsymbol{\lambda})\}\boldsymbol{V}^\top), \quad (91)$$

where $[\hat{\phi}_\ell(\boldsymbol{\lambda})]_i = \mathbb{1}(\lambda_i \text{ is the } \ell\text{-th smallest eigenvalue})$ and $\rho$ is continuous. Unfortunately, these $\hat{\phi}_\ell$ are not continuous so SPE cannot express them under stability requirement. Instead, we can construct a continuous function $\phi_\ell$ to approximate discontinuous $\hat{\phi}_\ell$ with arbitrary precision $\varepsilon$, say,

$$\forall \boldsymbol{\lambda} \in [0,2]^d, \quad \left\|\hat{\phi}(\boldsymbol{\lambda}) - \phi(\boldsymbol{\lambda})\right\| < \varepsilon. \quad (92)$$

Then we can upper-bound

$$\left\|\boldsymbol{V}\text{diag}\{\hat{\phi}_\ell(\boldsymbol{\lambda})\}\boldsymbol{V}^\top - \boldsymbol{V}\text{diag}\{\phi_\ell(\boldsymbol{\lambda})\}\boldsymbol{V}^\top\right\|_{\mathsf{F}} \overset{(a)}{\leq} \|\boldsymbol{V}\|\left\|\boldsymbol{V}^\top\right\|\left\|\hat{\phi}_\ell(\boldsymbol{\lambda}) - \phi_\ell(\boldsymbol{\lambda})\right\| < \epsilon, \quad (93)$$

where (a) holds due to the Lemma A.1. Moreover, using the continuity of $\rho$ (defined in Assumption 3.1), we obtain

$$\|\text{BasisNet}(\boldsymbol{V}, \boldsymbol{\lambda}) - \text{SPE}(\boldsymbol{V}, \boldsymbol{\lambda})\|_{\mathsf{F}} \leq J \sum_{\ell=1}^m \left\|\boldsymbol{V}\hat{\phi}(\boldsymbol{\lambda})\boldsymbol{V}^\top - \boldsymbol{V}\phi_\ell(\boldsymbol{\lambda})\boldsymbol{V}^\top\right\|_{\mathsf{F}} < Jd\varepsilon. \quad (94)$$

Now, let $\varepsilon = Jd/2$, then we can upper-bound the maximal error of node-level counting:

$$\max_{i \in [n]} |\#\text{cycles of node } i - \text{SPE}(\boldsymbol{V}, \boldsymbol{\lambda})|^2 \leq \|\text{BasisNet}(\boldsymbol{V}, \boldsymbol{\lambda}) - \text{SPE}(\boldsymbol{V}, \boldsymbol{\lambda})\|_{\mathsf{F}}^2 < J^2 d^2 \varepsilon^2 = 1/4. \quad (95)$$

$$\implies \max_{i \in [n]} |\#\text{cycles of node } i - \text{SPE}(\boldsymbol{V}, \boldsymbol{\lambda})| < 1/2. \quad (96)$$

Then, by applying an MLP that universally approximates rounding function, we are done with the proof. $\qquad\square$

## A.5 Proof of Proposition 3.3

**Proposition 3.3.** *Suppose that $(\boldsymbol{V}, \boldsymbol{\lambda})$ and $(\boldsymbol{V}', \boldsymbol{\lambda}')$ are such that $\boldsymbol{V}\boldsymbol{Q} = \boldsymbol{V}'$ for some orthogonal matrix $\boldsymbol{Q} \in O(d)$ and $\boldsymbol{\lambda} \neq \boldsymbol{\lambda}'$. Then there exist 2-layer MLPs for each $\phi_\ell$ and a 2-layer MLP $\rho$, each with ReLU activations, such that $SPE(\boldsymbol{V}, \boldsymbol{\lambda}) \neq SPE(\boldsymbol{V}', \boldsymbol{\lambda}')$.*

*Proof.* Our proof does not require the use of the channel dimension—i.e., we take $m$ and $p$ to equal 1. The argument is split into two steps.

First we show that for the given $\boldsymbol{\lambda}, \boldsymbol{\lambda}'$ and any $\phi, \phi' \in \mathbb{R}^d$ there is a choice of two layer network $\phi(\boldsymbol{\lambda}) = \boldsymbol{W}_2\sigma(\boldsymbol{W}_1\boldsymbol{\lambda} + \boldsymbol{b}_1) + \boldsymbol{b}_2$ such that $\phi(\boldsymbol{\lambda}) = \phi$ and $\phi(\boldsymbol{\lambda}') = \phi'$. Our choices of $\boldsymbol{W}_1, \boldsymbol{W}_2$ will have dimensions $d \times d$, $\boldsymbol{b}_1, \boldsymbol{b}_2 \in \mathbb{R}^d$, and $\sigma$ denotes the ReLU activation function.

Second we choose $\phi, \phi' \in \mathbb{R}^d$ such that $\boldsymbol{V}\phi\boldsymbol{V}^\top$ has strictly positive entries, whilst $\boldsymbol{V}'\phi'\boldsymbol{V}'^\top = \boldsymbol{0}$ (the matrix of all zeros). The argument will conclude by choosing $a_1, a_2, b_1, b_2 \in \mathbb{R}$ such that the 2 layer network (on the real line) $\rho(x) = a_2 \cdot \sigma(a_1 x + b_1) + b_2$ (a 2 layer MLP on the real line, applied element-wise to matrices then summed over both $n$ dimensions) produces distinct embeddings for $(\boldsymbol{V}, \boldsymbol{\lambda})$ and $(\boldsymbol{V}', \boldsymbol{\lambda}')$.

We begin with step one.

**Step 1:** If $\phi = \phi'$ then we may simply take $W_1$ and $b_1$ to be the zero matrix and vector respectively. Then $\sigma(W_1\lambda + b_1) = \sigma(W_1\lambda' + b_1) = 0$. Then we may take $W_2 = I$ (identity) and $b_2 = \phi$, which guarantees that $\phi(\lambda) = \phi$ and $\phi(\lambda') = \phi'$.

So from now on assume that $\phi \neq \phi'$. Let $\lambda$ and $\lambda'$ differ in their $i$th entries $\lambda_i, \lambda_i'$, and assume without loss of generality that $\lambda_i < \lambda_i'$. Let $W_1 = [0, \ldots, 0, e_i, 0, \ldots, 0]$ the matrix of all zeros, except the $i$th column which is the $i$th standard basis vector. Then $W_1\lambda$ is the vector of all zeros, except for $i$ entry equaling $\lambda_i$ (similarly for $\lambda'$). Next take $b_1$ to be the vector of all zeros, except for $i$th entry $-(\lambda_i + \lambda_i')/2$, the midpoint between the differing eigenvalues. These choices make $z = \sigma(W_1\lambda + b_1) = 0$, and $z' = \sigma(W_1\lambda' + b_1)$ such that $z_j' = 0$ for $j \neq i$, and $z_i' = (\lambda_i' - \lambda_i)/2$. Next, taking $W_2 = [0, \ldots, 0, c_i, 0, \ldots, 0]$ where the $i$th column is $c_i = 2(\phi' - \phi)/(\lambda_i' - \lambda_i)$ ensures that

$$W_2 z = 0, \tag{97}$$

$$W_2 z' = \phi' - \phi. \tag{98}$$

Then we may simply take $b_2 = \phi$, producing the desired outputs:

$$\phi(\lambda) = W_2\sigma(W_1\lambda + b_1) + b_2 = \phi, \tag{99}$$

$$\phi(\lambda') = W_2\sigma(W_1\lambda' + b_1) + b_2 = \phi' \tag{100}$$

as claimed.

**Step 2:** Expanding the matrix multiplications into their sums we have:

$$[V\mathrm{diag}(\phi)V^\top]_{ij} = \sum_d \phi_d v_{id} v_{jd} \tag{101}$$

$$[V'\mathrm{diag}(\phi')V'^\top]_{ij} = \sum_d \phi_d' v_{id}' v_{jd}'. \tag{102}$$

Our first choice is to take $\phi' = 0$, ensuring that $V'\mathrm{diag}(\phi')V'^\top = 0$ (an $n \times n$ matrix of all zeros). Next, we aim to pick $\phi$ such that $[\,V\mathrm{diag}(\phi)V^\top]_{i^*j^*} > 0$ for some indices $i^*, j^*$. In fact, this is possible for an $i, j$ pair since each pair of eigenvectors is orthogonal, and non-zero, so for each $i, j$ there must be a $d^*$ such that $v_{id^*} v_{jd^*} > 0$, and we can simply take $\phi_d = 1$ if $d = d^*$ and $\phi_d = 0$ for $d \neq d^*$.

Thanks to the above choices, taking $a_1 = 1$ and $b_1 = 0$ ensures that

$$\sigma\big(a_1 \cdot V'\mathrm{diag}(\phi')V'^\top + b_1\big) = 0. \tag{103}$$

but that,

$$\big[\sigma\big(a_1 \cdot V\mathrm{diag}(\phi)V^\top + b_1\big)\big]_{ij} > 0 \tag{104}$$

for some $i, j$. Note that in both cases, the scalar operations are applied to matrices element-wise.

Finally, taking $a_2 = 1/\sum_{ij}[\sigma\big(a_1 \cdot V\mathrm{diag}(\phi)V^\top + b_1\big)]_{ij} > 0$ and $b_2 = 0$ produces embeddings

$$\mathrm{SPE}(V, \lambda) = 1 \neq 0 = \mathrm{SPE}(V', \lambda'). \tag{105}$$

$\square$

## A.6  AUXILIARY RESULTS

**Proposition A.1** (Davis-Kahan theorem (Yu et al., 2015, Theorem 2)). *Let $A, A' \in \mathrm{S}(n)$. Let $\lambda_1 \leq \cdots \leq \lambda_n$ be the eigenvalues of $A$, sorted in increasing order. Let the columns of $x, x' \in \mathrm{O}(n)$ contain the orthonormal eigenvectors of $A$ and $A'$, respectively, sorted in increasing order of their corresponding eigenvalues. Let $\mathcal{J} = [\![s, t]\!]$ be a contiguous interval of indices in $[n]$, and let $d = |\mathcal{J}|$ be the size of the interval. For notational convenience, let $\lambda_0 = -\infty$ and $\lambda_{n+1} = \infty$. Then there exists an orthogonal matrix $Q \in \mathrm{O}(d)$ such that*

$$\big\|[x]_{\mathcal{J}} - [x']_{\mathcal{J}} Q\big\|_\mathsf{F} \leq \frac{\sqrt{8}\min\big\{\sqrt{d}\,\|L - L'\|, \|L - L'\|_\mathsf{F}\big\}}{\min\{\lambda_s - \lambda_{s-1}, \lambda_{t+1} - \lambda_t\}}. \tag{106}$$

**Proposition A.2** (Weyl's inequality)**.** *Let* $\lambda_i : \mathrm{S}(n) \to \mathbb{R}$ *return the ith smallest eigenvalue of the given matrix. For all* $\boldsymbol{A}, \boldsymbol{A}' \in \mathrm{S}(n)$ *and all* $i \in [n]$, $|\lambda_i(\boldsymbol{A}) - \lambda_i(\boldsymbol{A}')| \le \|\boldsymbol{A} - \boldsymbol{A}'\|$.

*Proof.* By Horn & Johnson (2012, Corollary 4.3.15),

$$\lambda_i(\boldsymbol{A}') + \lambda_1(\boldsymbol{A} - \boldsymbol{A}') \le \lambda_i(\boldsymbol{A}) \le \lambda_i(\boldsymbol{A}') + \lambda_n(\boldsymbol{A} - \boldsymbol{A}') . \tag{107}$$

Therefore $\lambda_i(\boldsymbol{A}) - \lambda_i(\boldsymbol{A}') \in [\lambda_1(\boldsymbol{A} - \boldsymbol{A}'), \lambda_n(\boldsymbol{A} - \boldsymbol{A}')]$, and

$$|\lambda_i(\boldsymbol{A}) - \lambda_i(\boldsymbol{A}')| = \max\left\{\lambda_i(\boldsymbol{A}) - \lambda_i(\boldsymbol{A}'), \lambda_i(\boldsymbol{A}') - \lambda_i(\boldsymbol{A})\right\} \tag{108}$$

$$\le \max\left\{\lambda_n(\boldsymbol{A} - \boldsymbol{A}'), -\lambda_1(\boldsymbol{A} - \boldsymbol{A}')\right\} \tag{109}$$

$$= \max_{i \in [n]} |\lambda_i(\boldsymbol{A} - \boldsymbol{A}')| \tag{110}$$

$$= \sigma_{\max}(\boldsymbol{A} - \boldsymbol{A}') \tag{111}$$

$$= \|\boldsymbol{A} - \boldsymbol{A}'\| . \tag{112}$$

$\square$

**Proposition A.3** (Hoffman-Wielandt corollary (Stewart & Sun, 1990, Corollary IV.4.13))**.** *Let* $\lambda_i : \mathrm{S}(n) \to \mathbb{R}$ *return the ith smallest eigenvalue of the given matrix. For all* $\boldsymbol{A}, \boldsymbol{A}' \in \mathrm{S}(n)$,

$$\sqrt{\sum_{i=1}^{n} (\lambda_i(\boldsymbol{A}) - \lambda_i(\boldsymbol{A}'))^2} \le \|\boldsymbol{A} - \boldsymbol{A}'\|_{\mathsf{F}} . \tag{113}$$

**Lemma A.1.** *Let* $\{\boldsymbol{A}_k\}_{k=1}^{p}$ *be compatible matrices. For any* $\ell \in [p]$,

$$\left\|\prod_{k=1}^{p} \boldsymbol{A}_k\right\|_{\mathsf{F}} \le \left(\prod_{k=1}^{\ell-1} \|\boldsymbol{A}_k\|\right) \|\boldsymbol{A}_\ell\|_{\mathsf{F}} \left(\prod_{k=\ell+1}^{p} \|\boldsymbol{A}_k^\top\|\right) . \tag{114}$$

*Proof.* First, observe that for any matrices $\boldsymbol{A} \in \mathbb{R}^{m \times r}$ and $\boldsymbol{B} \in \mathbb{R}^{r \times n}$,

$$\|\boldsymbol{A}\boldsymbol{B}\|_{\mathsf{F}} = \sqrt{\sum_{j=1}^{n} \left\|[\boldsymbol{A}\boldsymbol{B}]_j\right\|^2} = \sqrt{\sum_{j=1}^{n} \|\boldsymbol{A}\boldsymbol{B}\mathbf{e}_j\|^2} \le \sqrt{\sum_{j=1}^{n} \|\boldsymbol{A}\|^2 \|\boldsymbol{B}\mathbf{e}_j\|^2} = \|\boldsymbol{A}\| \sqrt{\sum_{j=1}^{n} \left\|[\boldsymbol{B}]_j\right\|^2}$$
$$= \|\boldsymbol{A}\| \|\boldsymbol{B}\|_{\mathsf{F}} . \tag{115}$$

Therefore,

$$\left\|\prod_{k=1}^{p} \boldsymbol{A}_k\right\|_{\mathsf{F}} \overset{(a)}{\le} \left(\prod_{k=1}^{\ell-1} \|\boldsymbol{A}_k\|\right) \left\|\prod_{k=\ell}^{p} \boldsymbol{A}_k\right\|_{\mathsf{F}} \tag{116}$$

$$\overset{(b)}{=} \left(\prod_{k=1}^{\ell-1} \|\boldsymbol{A}_k\|\right) \left\|\prod_{k=0}^{p-\ell} \boldsymbol{A}_{p-k}^\top\right\|_{\mathsf{F}} \tag{117}$$

$$\overset{(c)}{\le} \left(\prod_{k=1}^{\ell-1} \|\boldsymbol{A}_k\|\right) \left(\prod_{k=0}^{p-\ell-1} \|\boldsymbol{A}_{p-k}^\top\|\right) \|\boldsymbol{A}_\ell^\top\|_{\mathsf{F}} \tag{118}$$

$$\overset{(d)}{=} \left(\prod_{k=1}^{\ell-1} \|\boldsymbol{A}_k\|\right) \|\boldsymbol{A}_\ell\|_{\mathsf{F}} \left(\prod_{k=\ell+1}^{p} \|\boldsymbol{A}_k^\top\|\right) , \tag{119}$$

where (a) and (c) hold by applying eq. (115) recursively, and (b) and (d) hold because Frobenius norm is invariant to matrix transpose. $\square$

**Lemma A.2** (Permutation equivariance of eigenvectors)**.** *Let* $\boldsymbol{A} \in \mathbb{R}^{n \times n}$ *and* $\boldsymbol{P} \in \mathrm{P}(n)$. *Then for any* $\boldsymbol{x} \in \mathbb{R}^n$, $\boldsymbol{P}\boldsymbol{x}$ *is an eigenvector of* $\boldsymbol{P}\boldsymbol{A}\boldsymbol{P}^\top$ *iff* $\boldsymbol{x}$ *is an eigenvector of* $\boldsymbol{A}$.

*Proof.*

$$\boldsymbol{Px} \text{ is an eigenvector of } \boldsymbol{PAP}^\top \xLeftrightarrow{(a)} \exists \lambda \in \mathbb{R}, \ \boldsymbol{PAP}^\top \boldsymbol{Px} = \lambda \boldsymbol{Px} \tag{120}$$

$$\xLeftrightarrow{(b)} \exists \lambda \in \mathbb{R}, \ \boldsymbol{PAx} = \lambda \boldsymbol{Px} \tag{121}$$

$$\xLeftrightarrow{(c)} \exists \lambda \in \mathbb{R}, \ \boldsymbol{PAx} = \boldsymbol{P}\lambda \boldsymbol{x} \tag{122}$$

$$\xLeftrightarrow{(d)} \exists \lambda \in \mathbb{R}, \ \boldsymbol{Ax} = \lambda \boldsymbol{x} \tag{123}$$

$$\xLeftrightarrow{(e)} \boldsymbol{x} \text{ is an eigenvector of } \boldsymbol{A} \,, \tag{124}$$

where (a) is the definition of eigenvector, (b) holds because permutation matrices are orthogonal, (c) holds by linearity of matrix-vector multiplication, (d) holds because permutation matrices are invertible, and (e) is the definition of eigenvector. $\qquad \square$

**Lemma A.3** (Permutation invariance of eigenvalues). *Let $\boldsymbol{A} \in \mathbb{R}^{n \times n}$ and $\boldsymbol{P} \in \mathrm{P}(n)$. Then $\lambda \in \mathbb{R}$ is an eigenvalue of $\boldsymbol{PAP}^\top$ iff $\lambda$ is an eigenvalue of $\boldsymbol{A}$.*

*Proof.*

$$\lambda \text{ is an eigenvalue of } \boldsymbol{PAP}^\top \xLeftrightarrow{(a)} \exists \mathbf{y} \neq \mathbf{0}, \ \boldsymbol{PAP}^\top \mathbf{y} = \lambda \mathbf{y} \tag{125}$$

$$\xLeftrightarrow{(b)} \exists \mathbf{y} \neq \mathbf{0}, \ \boldsymbol{PAP}^\top \mathbf{y} = \lambda \boldsymbol{PP}^\top \mathbf{y} \tag{126}$$

$$\xLeftrightarrow{(c)} \exists \mathbf{y} \neq \mathbf{0}, \ \boldsymbol{PAP}^\top \mathbf{y} = \boldsymbol{P}\lambda \boldsymbol{P}^\top \mathbf{y} \tag{127}$$

$$\xLeftrightarrow{(d)} \exists \boldsymbol{x} \neq \mathbf{0}, \ \boldsymbol{PAx} = \boldsymbol{P}\lambda \boldsymbol{x} \tag{128}$$

$$\xLeftrightarrow{(e)} \exists \boldsymbol{x} \neq \mathbf{0}, \ \boldsymbol{Ax} = \lambda \boldsymbol{x} \tag{129}$$

$$\xLeftrightarrow{(f)} \lambda \text{ is an eigenvalue of } \boldsymbol{A} \,, \tag{130}$$

where (a) is the definition of eigenvalue, (b) holds because permutation matrices are orthogonal, (c) holds by linearity of matrix-vector multiplication, (d)-(e) hold because permutation matrices are invertible, and (f) is the definition of eigenvalue. $\qquad \square$

# B  EXPERIMENTAL DETAILS AND ADDITIONAL RESULTS

## B.1  IMPLEMENTATION OF SPE

SPE includes parameterized permutation equivariant functions $\rho : \mathbb{R}^{n \times n \times m} \to \mathbb{R}^{n \times p}$ and $\phi_\ell : \mathbb{R}^d \to \mathbb{R}^d$.

For $\phi_\ell$, we treat input $\boldsymbol{\lambda}$ as $d$ vectors with input dimension 1 and use either elementwise-MLPs, i.e., $[\phi_\ell(\boldsymbol{\lambda})]_i = \mathrm{MLP}(\lambda_i)$, or Deepsets to process them. We also use piecewise cubic splines, which is a $\mathbb{R}$ to $\mathbb{R}$ piecewise function with cubic polynomials on each piece. Given number of pieces as a hyperparameter, the piece interval is determined by uniform chunking $[0, 2]$, the range of eigenvalues. The learnable parameters are the coefficients of cubic functions for each piece. To construct $\phi_\ell$, we simply let one piecewise cubic spline elementwisely act on each individual eigenvalues:

$$[\phi_\ell(\boldsymbol{\lambda})]_i = \mathrm{spline}(\lambda_i). \tag{131}$$

For $\rho$, in principle any permutation equivariant tensor neural networks can be applied. But in our experiments we adapt GIN as $\rho$. Here is how: for input $\boldsymbol{A} \in \mathbb{R}^{n \times n \times m}$, we first partition $\boldsymbol{A}$ along the second axis into $n$ many matrices $\boldsymbol{A}_i \in \mathbb{R}^{n \times m}$ (in code we do not actually need to partition them since parallel matrix multiplication does the work). Then we treat $\boldsymbol{A}_i$ as node features of the original graph and independently and identically apply a GIN to this graph with node features $\boldsymbol{A}_i$. This will produce node representations $\boldsymbol{Z}_i \in \mathbb{R}^{n \times p}$ from $\boldsymbol{A}_i$. Finally we let $\boldsymbol{Z} = \sum_{i=1}^n \boldsymbol{Z}_i$ be the final output of $\rho$. Note that this whole process makes $\rho$ permutation equivariant.

## B.2 IMPLEMENTATION OF BASELINES

For PEG, we follow the formula of their paper and augment edge features by

$$e_{i,j} \leftarrow e_{i,j} \cdot \text{MLP}(\|\boldsymbol{V}_{i,:} - \boldsymbol{V}_{j,:}\|). \tag{132}$$

For SignNet/BasisNet, we refer to their public code release at https://github.com/cptq/SignNet-BasisNet. Specifically, SignNet uses GINs as $\phi$ and BasisNet uses 2-IGN as $\phi$. Note that the original BasisNet does not support inductive learning, i.e., it cannot even apply to new graph structures. This is because it has separate weights for eigensubspaces with different dimension. Here we simply initialize an unlearned weights for eigensubspaces with unseen dimensions.

## B.3 OTHER TRAINING DETAILS

We use Adam optimizer with an initial learning rate 0.001 and 100 warm-up steps. We adopt a linear decay learning rate scheduler. Batch size is 128 for ZINC, Alchemy and substructures counting, 64 for DrugOOD.

## B.4 CONTROLLING LIPSCHITZ CONSTANT OF MLPs

Here we state how we control the Lipschitz constant of MLPs in Section 5.3. Technically, by Lipschitz constant we actually mean an upper bound for best Lipschitz constant (the minimal Lipschitz constant for function). Note that for a composititon of Lipschitz functions ($f_1 \circ f_2 \circ ... \circ f_k$) with individual Lipschitz constants $C_1, C_2, ..., C_k$, the product $C_1 \cdot C_2 \cdot ... \cdot C_k$ is a valid Lipschitz constant. A MLP consists of linear layers and ReLU activation. The Lipschitz constants of linear layers are simply the operator norm of weight matrices, while ReLU is 1-Lipschitz. So we can easily get the Lipschitz constant of MLPs by multiplying the operator norm of weight matrices. As a result, we can control the Lipschitz constant to be $C$ by first normalizing weight matrices to be unit norm and then multiply a constant $C^{1/k}$ between layers assuming there are $k$ layers.

## B.5 GENERALIZATION GAP ON ZINC

We show the training loss and the generalization gap (test loss - training loss) on ZINC dataset as shown below. These loss are all evaluated at the epoch with minimal validation loss. We can see that though SignNet and BasisNet achieve a pretty low training MAE (high expressive power), their generalization gap is larger than other baselines (poor stability) and thus the final test MAE is not the best. For baseline GNN and PEG, they are pretty stable with small generalization gap, but the poor expressive power make them hard to fit the dataset well (training loss is high). In contrast, SPE has not only a lowest training MAE (high expressive power) but also a small generalization gap (good stability). That is why it can obtain the best test performance among all the models.

Table 3: Test/training MAE and generalization gap results (4 random seeds) on ZINC. Bold-face black, blue and pink are used to denote the **first**, second and third best method for each column (each method only counts once as there are multiple configurations).

| Dataset | PE method | #PEs | #param | Test MAE | Training MAE | General. Gap |
|---------|-----------|------|--------|----------|--------------|--------------|
| ZINC | No PE | N/A | 575k | $0.1772_{\pm 0.0040}$ | $0.1509_{\pm 0.0086}$ | $\mathbf{0.0263_{\pm 0.0113}}$ |
| | PEG | 8 | 512k | $0.1444_{\pm 0.0076}$ | $0.1088_{\pm 0.0066}$ | $0.0382_{\pm 0.0100}$ |
| | PEG | Full | 512k | $0.1878_{\pm 0.0127}$ | $0.1486_{\pm 0.0191}$ | $0.0342_{\pm 0.0206}$ |
| | SignNet | 8 | 631k | $0.1034_{\pm 0.0056}$ | $0.0418_{\pm 0.0101}$ | $0.0602_{\pm 0.0112}$ |
| | SignNet | Full | 662k | $0.0853_{\pm 0.0026}$ | $0.0349_{\pm 0.0078}$ | $0.0502_{\pm 0.0103}$ |
| | BasisNet | 8 | 442k | $0.1554_{\pm 0.0068}$ | $0.0513_{\pm 0.0053}$ | $0.1042_{\pm 0.0063}$ |
| | BasisNet | Full | 513k | $0.1555_{\pm 0.0124}$ | $0.0684_{\pm 0.0202}$ | $0.0989_{\pm 0.0258}$ |
| | SPE | 8 | 635k | $0.0736_{\pm 0.0007}$ | $\mathbf{0.0324_{\pm 0.0058}}$ | $0.0413_{\pm 0.0057}$ |
| | SPE | Full | 650k | $\mathbf{0.0693_{\pm 0.0040}}$ | $0.0334_{\pm 0.0054}$ | $0.0359_{\pm 0.0087}$ |

### B.6 RUNNING TIME EVALUATION

We evaluate the running time of SPE and other baselines on ZINC and DrugOOD. The results represents the training/inference time on the whole training dataset (ZINC or DrugOOD) over 5 trials. We can see that the speed SPE is overall comparable to SignNet, and is much faster than BasisNet. This is possibly because BasisNet has to deal with the irregular and length-varying input $[V_1V_1^\top, V_2V_2^\top, ...]$, which is hard for parallel computation in a batch, while SPE simply needs to deal with the more uniform $V \operatorname{diag}(\phi(\lambda))V^\top$.

Table 4: Training and inference time (average over 5 trials) on ZINC and DrugOOD, evaluated on the whole training dataset. GPU is Quadro RTX 6000

| Dataset | PE method | #PEs | Train Time (s) | Inference Time (s) |
|---|---|---|---|---|
| ZINC | No PE | N/A | $3.319_{\pm0.400}$ | $2.852_{\pm0.310}$ |
| | PEG | 8 | $3.785_{\pm0.424}$ | $3.590_{\pm0.341}$ |
| | PEG | Full | $3.639_{\pm0.387}$ | $3.518_{\pm0.318}$ |
| | SignNet | 8 | $8.724_{\pm0.686}$ | $3.546_{\pm0.366}$ |
| | SignNet | Full | $23.157_{\pm1.932}$ | $7.883_{\pm0.374}$ |
| | BasisNet | 8 | $49.923_{\pm6.391}$ | $18.295_{\pm0.569}$ |
| | BasisNet | Full | $66.176_{\pm4.015}$ | $27.546_{\pm0.622}$ |
| | SPE | 8 | $10.888_{\pm0.416}$ | $5.336_{\pm0.738}$ |
| | SPE | Full | $11.576_{\pm0.472}$ | $5.406_{\pm0.499}$ |
| DrugOOD | No PE | N/A | $13.560_{\pm0.657}$ | $4.260_{\pm0.140}$ |
| | PEG | 32 | $14.212_{\pm1.084}$ | $4.780_{\pm0.351}$ |
| | SignNet | 32 | $30.705_{\pm0.723}$ | $12.844_{\pm0.307}$ |
| | BasisNet | 32 | $199.364_{\pm2.807}$ | $86.529_{\pm1.693}$ |
| | SPE | 32 | $37.577_{\pm0.833}$ | $21.706_{\pm0.850}$ |

To see how complexity grows with graph size, we construct Erdos–Renyi random graphs for different graph sizes, ranging from 10 to 320. For each graph size, we construct 1,000 such random graphs with fixed node degree 2.5. Then we train and test each methods on these 1,000 graph for 10 epochs to estimate the time complexity. For fairness, each model has 60k parameters. By default we use batch size 50, and if it is out-of-memory (OOM), we use batch size 5 then. It batch size 5 still leads to OOM, we will denote OOM in the results. Below we report the average training/inference time per epoch.

| Graph size | GINE | PEG | SignNet | SPE | BasisNet |
|---|---|---|---|---|---|
| 10 | $0.505_{\pm0.277}$ | $0.574_{\pm0.321}$ | $1.750_{\pm0.332}$ | $1.357_{\pm0.321}$ | $5.185_{\pm0.407}$ |
| 20 | $0.535_{\pm0.275}$ | $0.631_{\pm0.336}$ | $1.648_{\pm0.325}$ | $1.323_{\pm0.309}$ | $3.418_{\pm0.323}$ |
| 40 | $0.561_{\pm0.297}$ | $0.644_{\pm0.332}$ | $2.293_{\pm0.362}$ | $1.507_{\pm0.352}$ | $6.019_{\pm0.305}$ |
| 80 | $0.609_{\pm0.294}$ | $0.750_{\pm0.338}$ | $5.810_{\pm0.335}$ | $4.030_{\pm0.369}$ | $40.848_{\pm0.359}$ |
| 160 | $1.056_{\pm0.271}$ | $1.410_{\pm0.356}$ | $21.153_{\pm0.275}$ | $55.256_{\pm0.757}$ | OOM |
| 320 | $2.714_{\pm0.411}$ | $4.403_{\pm0.425}$ | $83.833_{\pm0.168}$ | OOM | OOM |

Table 5: Average training time (s) on random graph dataset over 10 epochs.

| Graph size | GINE | PEG | SignNet | SPE | BasisNet |
|---|---|---|---|---|---|
| 10 | $0.125_{\pm0.001}$ | $0.154_{\pm0.001}$ | $0.465_{\pm0.002}$ | $0.385_{\pm0.002}$ | $2.326_{\pm0.248}$ |
| 20 | $0.163_{\pm0.001}$ | $0.204_{\pm0.001}$ | $0.434_{\pm0.002}$ | $0.299_{\pm0.001}$ | $1.659_{\pm0.006}$ |
| 40 | $0.174_{\pm0.001}$ | $0.212_{\pm0.002}$ | $0.986_{\pm0.002}$ | $0.543_{\pm0.002}$ | $3.167_{\pm0.101}$ |
| 80 | $0.213_{\pm0.003}$ | $0.307_{\pm0.004}$ | $2.976_{\pm0.017}$ | $2.093_{\pm0.007}$ | $22.054_{\pm0.017}$ |
| 160 | $0.512_{\pm0.035}$ | $0.778_{\pm0.032}$ | $11.886_{\pm0.134}$ | $38.747_{\pm0.247}$ | OOM |
| 320 | $1.500_{\pm0.098}$ | $2.841_{\pm0.113}$ | $48.179_{\pm0.264}$ | OOM | OOM |

Table 6: Average inference/test time (s) on random graph dataset over 10 epochs.

### B.7 ABLATION STUDY

One key component of SPE is to leverage eigenvalues using $\phi_\ell(\boldsymbol{\lambda})$. Here We try removing the use of eigenvalues, i.e., set $\phi_\ell(\boldsymbol{\lambda}) = 1$ to see the difference. Mathematically, this will result in $\text{SPE}(\boldsymbol{V}, \boldsymbol{\lambda}) = \rho([\boldsymbol{V}\boldsymbol{V}^\top]_{\ell=1}^m)$. This is pretty similar to PEG and we loss expressive power from this over-stable operation $\boldsymbol{V}\boldsymbol{V}^\top$. As shown in the table below, removing eigenvalue information leads to a dramatic drop of performance on ZINC-subset. Therefore, the processing of eigenvalues is an effective and necessary design in our method.

| Method | Test MAE |
|---|---|
| SPE ($\phi_\ell$ =MLPs) | $0.0693_{\pm 0.0040}$ |
| SPE ($\phi_\ell = 0$) | $0.1230_{\pm 0.0323}$ |

Table 7: Abalation study for SPE on ZINC (subset).

### B.8 MORE RESULTS ON TUDATASETS

We further conduct experiments on TUDatasets. For each task, we randomly split dataset into training, validation and test by 8:1:1. We uniformly use batch size 128 and train 250 epoch. Architectures and hyperparameters follows the same ones as on ZINC. We report the test accuracy at the epoch with highest validation accuracy over 5 random seeds. See Table 8 for results.

| | GINE | EPG | SignNet | BasisNet | SPE |
|---|---|---|---|---|---|
| PROTEINS | $71.07_{\pm 4.03}$ | $70.71_{\pm 7.20}$ | $73.21_{\pm 2.67}$ | $63.57_{\pm 2.42}$ | $74.82_{\pm 4.72}$ |
| ENZYMES | $51.33_{\pm 8.96}$ | $57.00_{\pm 2.50}$ | $45.00_{\pm 8.45}$ | $32.00_{\pm 4.08}$ | $52.34_{\pm 6.24}$ |
| PTC_MR | $54.86_{\pm 4.04}$ | $58.29_{\pm 8.85}$ | $54.86_{\pm 12.67}$ | $57.14_{\pm 11.67}$ | $58.29_{\pm 8.08}$ |
| MUTUG | $85.00_{\pm 6.29}$ | $84.00_{\pm 7.50}$ | $85.00_{\pm 14.36}$ | $79.00_{\pm 8.54}$ | $89.00_{\pm 4.08}$ |

Table 8: Test accuracy over 5 random seeds on TUDatasets.

## C WHY PREVIOUS POSITIONAL ENCODINGS ARE UNSTABLE?

### C.1 ALL SIGN-INVARIANT METHODS ARE UNSTABLE

One line of work (Dwivedi & Bresson, 2021; Kreuzer et al., 2021; Lim et al., 2023) is to consider the sign ambiguity of each individual eigenvectors and aim to make positional encodings invariant to sign flipping. The underlying assumption is that eigenvalues are all distinct so eigenvectors are equivalent up to a sign transformation. However, we are going to show that all these sign-invariant methods are **unstable**, regardless of the eigenvalues being distinct or not.

Firstly, suppose eigenvalues are distinct. Lemma 3.4 in Wang et al. (2022a) states that

**Lemma C.1.** *For any positive semi-definite matrix $B \in \mathbb{R}^{N \times N}$ without multiple eigenvalues, set positional encoding $PE(B)$ as the eigenvectors given by the smallest $p$ eigenvalues sorted as $0 = \lambda_1 < \lambda_2 < ... < \lambda_p(< \lambda_{p+1})$ of $B$. For any suffciently small $\epsilon > 0$, there exists a perturbation $\Delta B$, $\|B\|_F \leq \epsilon$ such that*

$$\min_{S \in SN(p)} \|PE(B) - PE(B + \Delta B)\| \geq 0.99 \max_{1 \geq i \leq p} |\lambda_{i+1} - \lambda_i|^{-1} \|\Delta B\|_F + o(\epsilon), \quad (133)$$

*where $SN(p) = \{Q \in \mathbb{R}^{p \times p} : Q_{i,i} = \pm 1, Q_{i,j} = 0, \text{for } i \neq j\}$ is the sign flipping operations.*

This Lemma shows that when there are two closed eigenvalues, a small perturbation to graph may still yield a huge change of eigenvectors that cannot be compensated by sign flipping. Therefore, these sign-invariant methods are highly unstable on graphs with distinct but closed eigenvalues.

On the other hand, if eigenvalues have repeated values, then the same graph may produce different eigenvectors that are associated by basis transformations. Simply invariant to sign flipping cannot

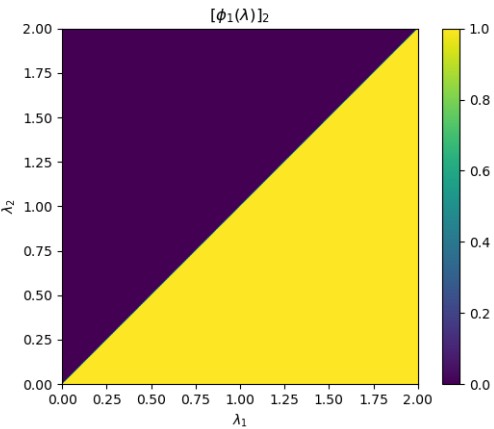

Figure 4: A illustration of $[\phi_1(\boldsymbol{\lambda})]_2$ for $\boldsymbol{\lambda} = (\lambda_1, \lambda_2)$. Clearly $\phi_1$ is discontinuous.

handle this basis ambiguity. As a result, these sign-invariant methods will produce different positional encodings for the same input graph. That means there is no stability gurantee for them at all.

### C.2 BASISNET IS UNSTABLE

Another line of work (e.g, BasisNet (Lim et al., 2023)) further consider basis invariance of eigenvectors by separately dealing with each eigensubspaces instead of each individual eigenvectors. The idea is to first partition eigenvectors $V \in \mathbb{R}^{n \times d}$ into their corresponding eigensubspace $(V_1, V_2, ...)$ according to eigenvalues, where $V_k \in \mathbb{R}^{n \times d_k}$ is the eigenvectors in $k$-th eigensubspace of dimension $d_k$. Then neural networks $\phi_{d_k} : \mathbb{R}^{n \times n} \to \mathbb{R}^{n \times p}$ is applied to each $V_k V_k^\top$ and the output will be $\rho(\phi_{d_1}(V_1 V_1^\top), \phi_{d_2}(V_2 V_2^\top), ...)$ where $\rho : \mathbb{R}^{n \times (d \cdot p)} \to \mathbb{R}^{n \times p}$ is a MLP. Intuitively, this method is unstable because a perturbation of graph can change the dimension of eigensubspace and thus dramatically change the input $(V_1, V_2, ...)$. As an example, let us say we have three eigenvectors ($d = 3$), and denote the three columns of $V$ as $u_1, u_2, u_3$. We construct two graphs: the original graph $A$ has $\lambda_1 = \lambda_2 < \lambda_3$ while the perturbed graph $A'$ has $\lambda_1' < \lambda_2' < \lambda_3'$. Two graphs share the same eigenvectors. Note that the difference between $A$ and $A'$ can be arbitrarily small to make $\lambda_2'$ a little bit different from $\lambda_2$. BasisNet will produce the following embeddings:

$$\text{BasisNet}(A) = \rho(\phi_2(u_1 u_1^\top + u_2 u_2^\top), \phi_1(u_3 u_3^\top))$$

$$\text{BasisNet}(A') = \rho(\phi_1(u_1 u_1^\top), \phi_1(u_2 u_2^\top), \phi_1(u_3 u_3^\top)).$$

Clearly as the input to $\rho$ are completly different, there is no way to ensure stability even if $\rho$ and $\phi$ are continuous.

### C.3 WHY STABILITY THEOREM 3.1 CANNOT BE APPLIED FOR PREVIOUS METHODS

One may wonder why we cannot prove the stability of previous hard-partition methods following the same argument in Theorem 3.1. The reason is that they do not satisfy Assumption 3.1 or the requirement that $\phi$ is equivariant, both of which are the key to prove stability result in Theorem 3.1.

To see this, let us first consider sign-invariant methods (e.g., SignNet). Using the notation of SPE, it is equivalent to use a $\phi_\ell(\lambda)$ to separate the $\ell$-th eigenvectors, thus whose $k$-th entry is

$$[\phi_\ell(\lambda)]_k = \delta_{k,\ell}. \tag{134}$$

This $\phi$ function does not rely on $\lambda$ and thus is Lipschitz continuous. However, it is not **permutation equivariant** to $\lambda$ vector. So it violates "$\phi$ is always permutation equivariant" as stated in the definition of SPE, and thus we cannot apply Theorem 3.1 to sign-invariant methods (and they are indeed

unstable as shown before).

On the other hand, let us consider basis-invariant methods using hard partition of eigensubspaces (e.g., BasisNet). In this case, the $k$-th entry of the hard partition $\phi_\ell(\lambda)$ is

$$[\phi_\ell(\lambda)]_k = \begin{cases} 1, & \text{if } \lambda_k \text{ is } \ell\text{-th smallest eigenvalue} \\ 0, & \text{otherwise} \end{cases}$$

This $\phi$ function is actually discontinuous. As an example, we may consider $\lambda = (\lambda_1, \lambda_2)$ and plot the figure of $[\phi_1(\lambda)]_2$ below. Clearly there is a sharp transition from 0 to 1 when $\lambda_2$ approches $\lambda_1$. Therefore, $\phi_\ell(\lambda)$ is more like a **step function** instead of a constant function. They are **discontinuous**, and thus are not Lipschitz continuous. So Assumption 3.1 does not hold and Theorem 3.1 cannot be applied for such hard partition functions.

