# OpenReview forum: "On the Stability of Expressive Positional Encodings for Graphs"
_ICLR.cc/2024/Conference — ICLR 2024 poster_

### Official Review · Reviewer_mePj · 2023-10-30

**Soundness:** 3 good
**Presentation:** 3 good
**Contribution:** 3 good
**Rating:** 6
**Confidence:** 4

**Summary:**

This paper studies the stability of eigenvector-based positional encodings while previous methods mainly focus on the sign- and basis-invariant properties. The authors claim that the instability of the previous method is caused by the hard partition of eigenvectors and the ignorance of eigenvalues. To address this challenge, this paper proposes SPE, which leverages the eigenvalues to re-weight the eigenvectors in a soft partition way. SPE is provably stable and shows great expressive power. Experiments on various tasks validate the superiority of the proposed method over baselines.

**Strengths:**

1. The proposed SPE is provably stable, which means that it can generalize to unseen graphs. I think this strong inductive learning ability is crucial for graph representation learning. The theoretical contribution is great.

2. The proposed SPE shows great expressive power, which not only universally approximates previous basis invariant functions but also can distinguish the cycles in graphs. Experimental results validate the effectiveness of the proposed method.

3. In addition to previous methods that conduct experiments in the basic molecular property prediction tasks, this paper also considers a more challenging out-of-distribution (OOD) task for evaluation.

**Weaknesses:**

1. The complexity of the proposed SPE is much larger than previous positional encoding methods because it needs to reconstruct graph structures, i.e., $\boldsymbol{V} \operatorname{diag}\left(\phi(\boldsymbol{\lambda})\right) \boldsymbol{V}^{\top}$, whose complexity is $\mathcal{O}(KN^{2})$. In contrast, the Transformer-based methods, e.g., BasisNet, only have the complexity of $\mathcal{O}(NK^{2})$, where $K \ll N$.

2. In the molecular property prediction task, SPE has more parameters than baselines. It would be better if the authors could align the number of parameters across different methods. Additionally, in the OOD tasks, the improvement of SPE over baselines is marginal.

**Questions:**

Here are some concepts that I am not sure I fully understand. Please correct me if there are any misunderstandings.

1. In equation (1), what does $\mathbb{R}^{n \times d} \times \mathbb{R}^d \rightarrow \mathbb{R}^{n \times p}$ mean? I understand that $\mathbb{R}^{n \times d} \rightarrow \mathbb{R}^{n \times p}$ represents the function applied to the position features of each node. What does $\mathbb{R}^d$ indicate? Operation on eigenvalues?

2. What is the difference between a hard partition and a soft partition?  I do not see a clear definition. Does hard partition indicate a fixed number of eigenvectors and does soft partition mean it can handle a variable number of eigenvectors?

3. Is it possible to replace the element-wise MLPs of $\phi$ with polynomial functions? In this situation, I think the complexity can be significantly reduced and the expressiveness can be preserved since polynomials are also non-linear.

---

> ### Author Response · Authors · 2023-11-18
> **Response to reviewer mePj**
>
> We thank the reviewer for the constructive comments.
>
> > **W1**: The complexity of the proposed SPE is much larger than previous positional encoding methods because it needs to reconstruct graph structures, i.e., $V\phi V^{\top}$, whose complexity is $N\times N\times K$. In contrast, the Transformer-based methods, e.g., BasisNet, only have the complexity of $NK^2$, where $K<<N$.
>
> Note that the complexity of BasisNet is also $N\times N\times K$, because it also needs to compute the matrix $V_kV^{\top}_k\in\mathbb{R}^{N\times N\times d_k}$ in $k$-th eigensubspace for $k=1, 2,...$. Also in general a transformer-based method should have complexity $O(N^2)$ because it requires to compute pair-wise correlation.
>
> Empirically, we also find SPE's running time is much faster than BasisNet. Kindly check our unified response ``*running time evaluation*'' above.
>
> > **W2**: In the molecular property prediction task, SPE has more parameters than baselines. It would be better if the authors could align the number of parameters across different methods. Additionally, in the OOD tasks, the improvement of SPE over baselines is marginal.
>
> Thank you for the suggestion. We will align the number of parameters in the revised manuscript.
>
> For OOD task, the core idea is not to achieve state-of-the-art performance but to show how previous unstable methods suffer from the **risk of instability** on OOD data. In this sense, standard GNN model is actually a strong baseline because it is very stable.
>
> > **Q1**: In equation (1), what does $\mathbb{R}^{n\times d}\times\mathbb{R}^d\to\mathbb{R}^{n\times p}$ mean?
>
> Yes, it means a general positional encoding method can take both eigenvalues and eigenvectors as input.
>
> > **Q2**: What is the difference between a hard partition and a soft partition? I do not see a clear definition. Does hard partition indicate a fixed number of eigenvectors and does soft partition mean it can handle a variable number of eigenvectors?
>
> Hard partition literally needs to do partition of eigenspace, i.e., splitting eigenvectors $V$ into different eigensubspace $(V_1, V_2,....)$ and computing $(V_1V_1^{\top}, V_2V_2^{\top}, ....)$. While ``soft partition'' actually **does not do partition** but instead is just a terminology to describe the smooth combination over all eigenvectors, $V\text{diag}(\phi(\lambda))V^{\top}$.
>
> > **Q3**: Is it possible to replace the element-wise MLPs of  with polynomial functions? In this situation, I think the complexity can be significantly reduced and the expressiveness can be preserved since polynomials are also non-linear.
>
> We think MLPs are not be the bottleneck of computation complexity. This is because the input size is simply the number of eigenvectors to use, which is usually fixed and does not scale with graph size. On the other hand, MLPs are usually believed to be more expressive than polynomials. So overall MLPs are preferable for us.

---

> > ### Comment · Reviewer_mePj · 2023-11-21
> > **Response to the rebuttal**
> >
> > Thanks for the clarifications. Now I have no doubt about the efficiency of SPE.

---

### Official Review · Reviewer_qQbF · 2023-10-30

**Soundness:** 3 good
**Presentation:** 4 excellent
**Contribution:** 3 good
**Rating:** 5
**Confidence:** 4

**Summary:**

This paper proposes a new approach for generating positional encodings which are stable and can universally approximate basis invariant functions. To compute those encodings, the method first decomposes the Laplacian matrix, it then applies different permutation equivariant functions to the eigenvalues, and uses the output of those layers to produce matrices of dimension $n \times n$ which are then fed to another permutation equivariant network (e.g., a GNN). The proposed method is evaluated on molecular property prediction datasets and also on a dataset with domain shifts where it outperforms other positional encoding methods in most cases.

**Strengths:**

- The stability of graph learning algorithms is a topic that has not been explored that much yet and deserves more attention from the community. The results presented in this paper contribute to this direction.

- In my view, the paper has some value since several individuals from the graph learning community would be interested in knowing its findings. Practitioners would also be interested in utilizing the proposed encodings since in many settings, existing GNN models fail to generalize to unseen domains.

- The proposed model achieves low values of MAE on the ZINC and Alchemy datasets and outperforms the baselines. This might be related to the model's ability to identify and count cycles of different lengths.

**Weaknesses:**

- I feel that the paper lacks some explanations. It is not clear which modules of the proposed method contribute to it being stable. If no $\phi_\ell$ layers are added, wouldn't $K_\ell$ be equal to 1? In my understanding, this wouldn't hurt stability. Also, it seems to me that as $m$ increases the bound becomes looser and looser. If that's the case, why do we need multiple such permutation equivariant layers?

- One of the main weaknesses of this paper is the proposed model's complexity. Function $\rho$ takes a tensor of dimension $n \times n \times m$ as input. This might not be problematic in case the model is trained on molecules since molecules are small graphs. But in case of other types of graphs such as those extracted from social networks which are significantly larger, this can lead to memory issues.

- The proposed approach is much more complex that a standard GNN model, but in most cases it provides minor improvements over a model that does not use positional encodings. For instance, the improvement on Alchemy is minor, and also on DrugOOD, SPE provides minor improvements in the Assay and Scaffold domains and no improvements in the Size domain.

- No running times of the different models are reported in the paper.

- The proposed model seems to advance the state of the art in the field of positional encodings for graphs, however, it is not clear whether it also advances the state of the art in the graph learning community. I would suggest the authors compare the proposed approach against some recently proposed GNN models, and not only against methods that produce positional encodings.

Typos:\
p.6: "hold and Let" -> "hold and let"\
p.7: "we take to $\rho$ to be" -> "we take $\rho$ to be"\
p.8: "which hypothesizes is because" -> "which is because"

**Questions:**

In Figure 2, how did you compute the Lipschitz constant of MLPs? We can compute the Lipschitz constant for models that consist of a single layer, but exact computation of the Lipschitz constant of MLPs
is NP-hard [1].

[1] Virmaux, Aladin; Scaman, Kevin. Lipschitz regularity of deep neural networks: analysis and efficient estimation. Advances in Neural Information Processing Systems, 2018.

---

> ### Author Response · Authors · 2023-11-18
> **Response to reviewer qQbF**
>
> We thank the reviewer for the thoughful review. The followings are our response.
>
> > **W1**: I feel that the paper lacks some explanations. It is not clear which modules of the proposed method contribute to it being stable. If no $\phi_i$ layers are added, wouldn't  $K_i$ be equal to 1? In my understanding, this wouldn't hurt stability.
>
> The stability stems from the operation $V\text{diag}(\phi(\lambda)) V^{\top}$. The key intuition is to combine multiple eigensubspaces with smooth composition, avoiding the splitting of individual eigensubspaces (which is what previous methods did).
>
> If you simply let $\phi(\lambda)$ be $\lambda$, it is true that this model is also stable. However, its **expressive power** will be damaged. For instance, if we let $\phi(\lambda)=\lambda$ and use full eigenvectors, then $V\text{diag}(\phi(\lambda))V^{\top}$ simply becomes Laplacian $L$ (and equivalently adjacency matrix $A$). In this case SPE is just acting like a normal graph neural networks on $A$, bringing no aditional expressive power. Similarly, larger $m$ (number of $\phi$) may damage stability but could bring more expressivity. So in practice we let $m$ be a hyperparameter for the stability-expressivity trade-off.
>
> > **W2**: One of the main weaknesses of this paper is the proposed model's complexity.
>
> We agree with the reviewer that complexity of current implementation of SPE could be a problem for large graphs. This problem also appears in BasisNet, whose complexity is also $O(N^2)$. Empirically, our unified response ``*running time evaluation*'' above shows the running time of SPE and other methods. We can see that SPE's running time is comparable to SignNet and is much faster than BasisNet.
>
> One alternative way for reducing complexity could be to sparse the $V\text{diag}(\phi) V^{\top}$ by only computing and storing its $(i,j)$-entry where $(i,j)$ is an edge. Then these positional encodings are treated as edge features for downstream tasks. The complexity will reduce from $O(n^2)$ to $O(|E|)$ where $|E|$ is the number of edges. In practice, the sparsing of $V\text{diag}(\phi) V^{\top}$ can be efficiently done by firstly assigning $V_{i, :}$ to $i$-th node features and using an one-layer message passing neural network to update and store edge features $[V\text{diag}(\phi) V^{\top}]\_{i,j}=\sum_{k}[\phi(\lambda)]\_k V\_{i,k}V\_{j,k}$.
>
> > **W3**: `The proposed approach is much more complex that a standard GNN model, but in most cases it provides minor improvements over a model that does not use positional encodings.'
>
> It is true that on Alchemy all PE methods provide minor improvements. But still SPE is the best. We conjecture that Laplacian encodings may not be a very good inductive bias on this dataset. For DrugOOD, the core idea is not to achieve state-of-the-artperformance but to show how previous unstable methods suffer from the **risk of instability** on OOD data. Standard GNN model is actually a strong baseline because it is very stable.
>
> > **W4**: No running times of the different models are reported in the paper.
>
> Thank you for raising this point. We add running time of all methods on ZINC and DrugOOD in unified response ``*running time evaluation*'' above. Overall we can see that for our current implementation SPE has comparable running time to SignNet and much faster than BasisNet.
>
> > **W5**: The proposed model seems to advance the state of the art in the field of positional encodings for graphs, however, it is not clear whether it also advances the state of the art in the graph learning community. I would suggest the authors compare the proposed approach against some recently proposed GNN models, and not only against methods that produce positional encodings.
>
> Thank you for the valuable sugeestion. We plan to add a few recently proposed GNN models for comparison during the rebuttal period. Again, graph positional encodings are important for graph learning community since (1) it plays a fundamental role for constructing powerful graph transformers [1, 2]; (2) it can be plugged into any GNN models as feature augmentation [3] and improve their expressive power.
>
>
> [1] Vijay Prakash Dwivedi and Xavier Bresson. A generalization of transformer networks to graphs. AAAI Workshop on Deep Learning on Graphs: Methods and Applications, 2021.
>
> [2] Devin Kreuzer, Dominique Beaini, Will Hamilton, Vincent Lé etourneau, and Prudencio Tossou. Rethinking graph transformers with spectral attention. Advances in Neural Information Processing Systems, 34, 2021.
>
> [3] Vijay Prakash Dwivedi, Anh Tuan Luu, Thomas Laurent, Yoshua Bengio, and Xavier Bresson. Graph neural networks with learnable structural and positional representations. In International Conference on Learning Representations, 2022.

---

> ### Author Response · Authors · 2023-11-18
> **Response to reviewer qQbF (2)**
>
> > **Q1**: In Figure 2, how did you compute the Lipschitz constant of MLPs? We can compute the Lipschitz constant for models that consist of a single layer, but exact computation of the Lipschitz constant of MLPs is NP-hard
>
> Thank you for pointing this out. Technically, the paper mentioned refers to the best (or minimal) Lipschitz constant. Here by Lipschitz constant we actually mean an upper bound for best Lipschitz constant. Note that we can upper-bound the best Lipschitz constant of MLPs by multiplying the operator norm of weight matrices. We clarify this in the Appendix Section B.4 in the revised manuscript.

---

> > ### Comment · Reviewer_qQbF · 2023-11-21
> > **Response to Authors**
> >
> > I would like to thank the authors for their response. I appreciate the authors' comments and additional experiments. Actually, SPE's running time seems not to be prohibitive. However, the considered datasets contain only small graphs. I would like to see how the running time grows as a function of the size of the input graphs. The authors could create some synthetic dataset for this purpose.
> >
> > I agree with the authors that positional encodings are really important mainly because they are very general and can be plugged into any model. But do we actually need all these positional encodings? As I mentioned in the review, SPE provides minor improvements on the considered datasets. Could the authors construct some synthetic dataset where the SPE (because of its stability properties) significantly outperforms the models that use different encodings?
> >
> > I am not fully convinced by the authors' response to my first comment. The authors claim that we might need a larger $m$ (number of $\phi$) such that the model is more expressive, but this may damage stability. The main selling point of the paper is the stability of the produced representations, but by adding several layers, the model might become stable. I think this deserves further discussion and some experimental results on that would strengthen the paper.

---

> ### Author Response · Authors · 2023-11-22
> **Follow-up response to reviewer qQbF**
>
> We thank the reviewer for the new response.
>
> At the beginning, we would like to address your concern about stability property, which is the most important contribution of our work.
> >  The authors claim that we might need a larger $m$ (number of $\phi$) such that the model is more expressive, but this may damage stability. The main selling point of the paper is the stability of the produced representations, but by adding several layers, the model might become stable.
>
> Note that the one of main message we want to convey is **``stability guarantee is important''**, instead of ``more stability is better''. This is because a model class with stability guarantee has generalization guarantee as well. Its stability level (which relates to generalization) and expressive power naturally forms a trade-off. The Lipschitz constant of $\phi$, number of $\phi$, etc., serve as factors to control the trade-off and can be tuned to achieve empirical good results in practice. So it **does not hurt** to let $m$ be a hyper-parameter and it does not conflict with our stability argument.
>
> > I think this deserves further discussion and some experimental results on that would strengthen the paper.
>
> We agree on this point. Actually what we showed in Figure 2 is exactly the effect of stability level on model's expressive power and generalization, although we control the stability of $\phi$ by its Lipschitz constant instead of the number of $\phi$ to use.
>
> > I would like to see how the running time grows as a function of the size of the input graphs.
>
> We think this is a good suggestion and we are working on it right now.
>
> > Could the authors construct some synthetic dataset where the SPE (because of its stability properties) significantly outperforms the models that use different encodings?
>
> From the theory (see lemma 3.4 in ``*why previous methods are unstable'*'above), we know that by properly constructing graphs with small eigengap, previous unstable positional encoding methods should suffer from huge stability issue. Unfortunately, we may not be able to finish this experiment due to limited time of rebuttal period.

---

> ### Author Response · Authors · 2023-11-23
> **More running time results**
>
> Dear reviwer qQbF,
>
> Please kindly check Table 5 and 6 in Appendix B.6 for new running time results on synthetic datasets with different graph size. Basically, we can see that SPE has a similar running time as SignNet when graph size is less than 160. In contrast, BasisNet causes out-of-memory (OOM) problem even we use batch size 5 for graph size 160. When graph size increases to 320, both SPE and BasisNet will lead to OOM due to square complexity, and SignNet's running time is very high as well.

---

### Official Review · Reviewer_TVwL · 2023-11-01

**Soundness:** 3 good
**Presentation:** 3 good
**Contribution:** 3 good
**Rating:** 6
**Confidence:** 3

**Summary:**

This paper introduces Stable and Expressive Positional Encodings (SPE), an architecture that mainly addresses the challenges of instability in using Laplacian eigenvectors as positional encodings for graph neural networks. The key insight to overcome instability is to avoid `hard partitions' of eigen-subspaces, and instead, use soft partitions via Lipshitz continuous functions over the spectrum. The stability of SPE is proved and validated via out-of-distribution generalization experiments.  Universal expressiveness is also proved, mainly based on another work, i.e., BasisNet.

**Strengths:**

- S1. I like the design of the experiment in that the authors validate the stability of SPEs from an aspect of out-of-distribution generalization.

- S2. The authors target at robustness/instability and generalization of PEs, which is novel in the literature.

- S3. The paper is overall well written.

**Weaknesses:**

>  W1. The instability of prior method (i.e., the so-called hard partition method) is not proved.

 The authors point out under Eq.2 that **hard partition** is induced when $\[\phi_{\ell}(\boldsymbol{\lambda})\]_j=\mathbb{1}$(other places in $\phi(\cdot)$ are zeros),

and then $\boldsymbol{V}\text{diag}(\phi_{\ell}(\boldsymbol{\lambda}))\boldsymbol{V}^{T}$ is the $\ell$-th subspace.

The problem is that, if we set $\\{ \phi_i \\}_{i=1}^{m}$ that induce the hard partitions,

then they are **constant functions** and meet the $K_{\ell}$-Lipschitz continuous assumption in Assumption 3.1, which is then used to prove the stability of SPE.

Therefore, the question is,  **is the prior method (i.e., the counterpart that uses hard partitions) really unstable?** It seems that hard-partitioned SPE can be proved to be stable via exactly the same proof of Theorem 3.1.

> W2. Equivalence for $\\{\phi_i\\}_{i=1}^{m}$ .

The authors restrict $\\{\phi_i\\}_{i=1}^{m}$ to be permutation equivariant, whose input is the Laplacian spectrum. Here, the authors are asking for equivalence under the reordering of eigenmaps/eigenvalues, instead of the reordering of graph nodes.

> W3. On the universal expressiveness.

The proof of this SPE's universality relies on being reduced to BasisNet. Therefore, two problems arise:

- The experiment regarding expressiveness, i.e., the graph substructure counting, does not include BasisNet.
- According to Lim et al. (2023), the instance of BasisNet, Unconstrained-BasisNet, universally approximates any continuous basis invariant function. In Unconstrained-BasisNet,  IGN-2 (Maron et al., 2018) is the core part to achieve such expressiveness. However, in implementation, the authors set $\rho$ to be one identical GIN (Xu et al., 2019),  which would surely limit the expressiveness.

> W4. Lack of description of baseline models.

For the same reason as in W3, in the experimental part, specific choices of baseline instances, i.e.,  $\rho$ and $\phi$ of BasisNet, should be described more clearly.

**Questions:**

Please check W1, W2, and W3. Below is an additional question:

Q1: Would the learned  $\\{\phi_i\\}_{i=1}^{m}$ be close to each other? This would lead to similar position encodings.

---

> ### Author Response · Authors · 2023-11-18
> **Response to Reviewer TVwL**
>
> We thank the reviewer for the comprehensive and constructive review.
>
> > **W1**: The instability of prior method (i.e., the so-called hard partition method) is not proved.
>
> Thank you for pointing this out. Kindly check our unified response ``*Why previous methods are unstable?*'' above for detailed proof.
>
> Regarding your argument that hard paritition is stable by Theorem 3.1, it is not valid because $\phi$ is actually **discontinuous**. To see this, let us consider basis-invariant methods using hard partition of eigensubspaces (e.g., BasisNet). In this case, the $k$-th entry of the hard partition $\phi_{\ell}(\lambda)$ is $$[\phi\_{\ell}(\lambda)]\_k=1,  \text{if $\lambda_k$ is $\ell$-th smallest eigenvalue},\quad [\phi\_{\ell}(\lambda)]\_k =0, \text{otherwise}$$
> This $\phi$ function  is more like a **step function** instead of a constant function. It is **discontinuous**, and thus is not Lipschitz continuous. So Assumption 3.1 does not hold and Theorem 3.1 cannot be applied for such hard partition functions. You may kindly check Appendix C in the revised manuscript for details and examples.
>
> > **W2**: The authors restrict $\phi$ to be permutation equivariant, whose input is the Laplacian spectrum. Here, the authors are asking for equivalence under the reordering of eigenmaps/eigenvalues, instead of the reordering of graph nodes.
>
> The overall model is indeed permutation **equivariant to node reordering**. This is because the permutation equivariance of $\phi$ (w.r.t eigenvalues) gurantees the permutation equivariance of the $V\phi(\lambda)V^{\top}$ to node indices reordering. To formally see this, suppose we have eigendecomposition for $L=V\text{diag}(\lambda)V^{\top}$. Applying permutation matrix $P$ to $L$ may produce an new decomposition in form of $PLP^{\top}=PVQ\text{diag}(\lambda)Q^{\top}V^{\top}P^{\top}$ with certain $Q$, where $Q\in O(\lambda)$ is an block-diagonal matrix where each block is a rotation matrix (i.e., $O(\lambda)=\\{\oplus_iQ_i: Q_i\in O(d_i), d_i\text{ is the dim of $i$-th eigensubspace}\\}$). That means the new eigenvectors after permutation become $PVQ$. If $\phi$ is permutation equivaraint, then it means $Q\text{diag}(\phi(\lambda))Q^{\top}=QQ^{\top}\text{diag}(\phi(\lambda))=\text{diag}(\phi(\lambda))$. Here the first equality holds since permutation equivariant $\phi$ produces same values for entries of the same eigensubspace and thus we can switch the order of multiplication with block-diagonal $Q$.  Therefore the new positional encoding $PVQ\text{diag}(\phi(\lambda))Q^{\top}V^{\top}P^{\top}=PV\text{diag}(\phi(\lambda))V^{\top}P^{\top}$, that is, the permutation version of the old positional encoding.
>
> > **W3**: The proof of this SPE's universality relies on being reduced to BasisNet. Therefore, two problems arise: 1. The experiment regarding expressiveness, i.e., the graph substructure counting, does not include BasisNet. 2. According to Lim et al. (2023), the instance of BasisNet, Unconstrained-BasisNet, universally approximates any continuous basis invariant function. In Unconstrained-BasisNet, IGN-2 (Maron et al., 2018) is the core part to achieve such expressiveness. However, in implementation, the authors set to be one identical GIN (Xu et al., 2019), which would surely limit the expressiveness.
>
> 1. Please kindly check the new counting results of BasisNet in Figure 3 in the refined manuscript.
> 2. We agree that $\rho$ has to be sufficiently powerful to achieve universality. But in practice we find GIN can already get promising results. This may be because the proposed architecture $V\text{diag}(\phi(\lambda))V^{\top}$ is already expressive and thus a complicated $\rho$ may not be that necessary for the current tasks.
>
> > **W4**: For the same reason as in W3, in the experimental part, specific choices of baseline instances, i.e., $\phi$ and  $\rho$ of BasisNet, should be described more clearly.
>
> Thank you for pointing this out. We follow the same setting in BasisNet paper and use 2-IGNs for $\rho$ and MLPs for $\phi$. These details are made more clear in the Appendix B.2 in our revised manuscript.
>
> > **Q1**: Would the learned $\phi_i$ be close to each other? This would lead to similar position encodings.
>
> We don't observe such phenomenon in practice, but if it is the case it just means one channel for such $\phi$ is sufficient for positional encoding on this specific task.

---

> > ### Comment · Reviewer_TVwL · 2023-11-19
> > **Still on the "discontinuity" of hard-partioning \phi  (W1)**
> >
> > Thanks to the author for the reply. However, I don't think I articulated **W1** clearly enough that **you completely missed my point**. Given the limited time remaining for author-reviewer interaction, I'd like to restate **W1** first (before I can go through ``Why previous methods are unstable?'' carefully).
> >
> > Your response is:
> > > **RW1** ... **This function is more like a step function instead of a constant function**. It is discontinuous, and thus is not Lipschitz continuous. So Assumption 3.1 does not hold and Theorem 3.1 cannot be applied for such hard partition functions. You may kindly check Appendix C in the revised manuscript for details and examples.
> >
> > I would like to draw the author's attention that   $\phi_{\ell}$, **as you have written in P4, is multivariate**, i.e. the input and output are both $d$-dimensional. Therefore, I can set $\phi_{\ell}$ to be **constant**, which is in the form of $[0,0,\cdots, 1, 0, 0]$. Such an **constant function** is of course continuous and satisfies Assumption 3.1, which is then used to prove the stability of SPE.

---

> ### Author Response · Authors · 2023-11-19
> **Phi also needs to be permutation equivariant as stated in the page 5**
>
> We thank the reviewer for raising this really good question and we are sorry for making this confusion. The reviewer says setting $\phi_{\ell}(\lambda)$ to be a **constant vector** also satisfies Assumption 3.1. This is definitely **true**. However, this does not necessarily make it stable, because we also stated that ``**$\phi_{\ell}$ are always permutation equivariant neural networks**"" in Page 5, after equation (2). Theorem 3.1 is based the equivariant condition of $\phi_{\ell}$. In constrat, a constant vector of form $[0, 0, ..., 1, 0, 0]$  is clearly NOT permutation equivariant, so Theorem 3.1 cannot apply. The only equivariant constant vector is $c\cdot [1, 1, 1, ..., 1, 1]$ (c is some constant) and it kind of reduces to PEG and is stable indeed.
>
> In fact, if $\phi_{\ell}$ is NOT permutation equivariant, then the overall model is NOT permutation equivariant as well. The overall permutation equivariance is a necessary condition for stability (Remark 3.1), this is why we always constrain $\phi_{\ell}$ to be equivariant when discussing SPE. For this, please see our response above to your question on W2.
>
> Specifically, to understand why setting $[0, 0, ..., 1, 0, 0]$ (say, $k$-th entry is 1) is unstable, we can consider there are multiple eigenvalues, say $\lambda_k=\lambda_{k+1}$, and the corresponding eigenspace $S=\text{Span}(v_{k},v_{k+1})$ where $v_k, v_{k+1}\in \mathbb{R}^n$ give the basis the eigenspace. When $\phi=[0,0,...,1,0,0]$, even if the graph does not change, the positional encoding $V\phi V^T$ becomes $v v^T$ with any $v\in S$ and $||v||_2=1$, which is non-unique and thus causes instability.
>
> Again, thank you for let us realize this is not clear to the readers. We will make this equivariance condition more clear in the refined manuscript. Also, you can kindly check Appendix C in the refined manuscript, which explains why stability Theorem 3.1 cannot be applied to all of the previous methods.

---

> ### Comment · Reviewer_TVwL · 2023-11-20
> **Response: on Phi**
>
> Thanks for the quick follow-up. Now I can accept the property and role of $\phi$ :
>
> - $\phi_{\ell}$ are permutation equivariant, such that they yield the same values on indices corresponding to the same eigenvalues, which then leads to the equivalence relation $Q\text{diag}(\phi(\lambda))Q^{\top}=QQ^{\top}\text{diag}(\phi(\lambda))=\text{diag}(\phi(\lambda))$. I think it is better to say that the permutation equivariance of  $\phi_{\ell}$ contributes to basis invariance. The equivariance of $\phi$ digests $Q$, which is brought by the uncertainty of basis $V$ -- In BasisNet, $Q$ is ``digested" by $V V^{\top}$, instead.
>
> -  $\phi_{\ell}$ are continuous, combing equivariance, this makes SPE stable.
>
> I will raise my score.

---

> ### Comment · Reviewer_TVwL · 2023-11-20
> **Response: on W3&W4**
>
> In the initial stages of the rebuttal, my primary focus centered around two key points:
>
> - W1&W2: Addressing the role of $\phi$.
>
> - W3&W4: Discussing the relationship between this study and BasisNet.
>
> While progress has been made on the first point, my attention remains directed toward remaining issues on the second aspect.
>
> The utilization of a single identical GIN repeated $n$ times for setting $\rho$ raises concerns among reviewers. This concern isn't solely due to the potential increase in time complexity, but also stems from the fact that such a model structure is notably uncommon. Could the use of stacked GINs (although limits expressiveness theoretically) be a contributing factor enabling your model to outperform BasisNet?

---

> > ### Author Response · Authors · 2023-11-23
> > **Further response to W3&W4**
> >
> > We adimit that the empirical choice of $\rho$ is not fully explored, as we mainly focused on the theoretical property of stability. Initially we adopted GINs since it is the most straightforward permutation equivariant networks. Due to time limit of discussion period, we may not be able to completely study the empirical impact of choosing different $\rho$.

---

### Official Review · Reviewer_fM7Q · 2023-11-02

**Soundness:** 3 good
**Presentation:** 2 fair
**Contribution:** 2 fair
**Rating:** 5
**Confidence:** 3

**Summary:**

This paper proposes stable and expressive Laplacian positional encodings (SPE) by performing a soft and learnable partition of eigensubspaces. The encoding guarantees that small perturbations to the input Laplacian induce a small change to the final positional encodings. The empirical results suggest a trade-off between stability (correlated with better generalization) and expressive power.

**Strengths:**

- The idea to address the stability of LapPE is novel, with SPE being a universal basis invariant architecture.
- The motivation is well established, the difference to other related works is concise, the propositions are described well and the strength of SPE as a universal basis invariant architecture is presented thoroughly.
- The experiments show the improvement in generalisation for SPE and its improved capabilities in recognising substructures, which are interesting outcomes of the architecture.

**Weaknesses:**

- The novelty of the method itself is partially limited as the idea to use a weighted correlation over the eigenvectors closely resembles the correlation used in BasisNet.
- The experiments are limited. The performance of SPE in Table 1 is sub-par, and details of the experimental results in Figure 2 are unclear and the hyperparameters seem not to be reported, which makes it hard to reproduce the experiments.
- The point regarding the trade-off between expressivity and generalisation is unclear. Is there a formal explanation which we can quantify?
- Perhaps additional experiments could be useful, e.g.,:
  - An experiment comparing the generalisation gap of LapPE/BasisNet/SPE.
  - Evaluating the performance of LapPE/BasisNet/SPE on LRGB or TUDatasets.

**Questions:**

Please refer to my review.

---

> ### Author Response · Authors · 2023-11-18
> **Response to Reviewer fM7Q**
>
> We first thank the reviewer for the valuable comments. Here is our response.
>
> > **W1**: The novelty of the method itself is partially limited as the idea to use a weighted correlation over the eigenvectors closely resembles the correlation used in BasisNet.
>
> Note that the key difference between our method and BasisNet is not just adding a weight: BasisNet computes the correlation $V_kV_k^{\top}$ between eigenvectors $V_k$ **within** the **$k$-th eigensubspace** for different $k$; in contrast, SPE computes the correlation $V\phi V^{\top}$ between **all the eigenvectors $V$ across different eigensubspaces**. This allows us to combine multiple eigensubspaces with smooth composition. The main argument is that this operation equipped with a continuous and equivariant $\phi$ is the key to achieve **stability**.
>
> > **W2**: The experiments are limited. The performance of SPE in Table 1 is sub-par.
>
> The SPE's result on ZINC is 0.0693, which greatly improves the baseline such as SignNet (0.0853) and BasisNet (1.5555). In fact, this result also outperforms many other recently proposed positional encodings methods [1, 2, 3] and even highly expressive GNN models [4, 5, 6].
>
> For Alchemy dataset, SPE is still the best, though all positional encodings bring a little improvement. We conjecture that Laplacian encodings may not be a good inductive bias for this dataset.
>
> Also thanks for pointing the hyperparameter things out. Some of the hyperameters were reported in the Appendix, Section B and we add more details of them in the revised manuscript.
>
> > **W3**: The point regarding the trade-off between expressivity and generalisation is unclear. Is there a formal explanation which we can quantify?
>
> A classic interpretation is the **variance bias trade-off**. Let $\hat{\epsilon}_n$ be the training loss, $\epsilon$ be the test loss, $\mathcal{H}$ be the hypothesis class and $\hat{h}_n=\arg\min\_{h\in \mathcal{H}}\hat{\epsilon}\_n(h)$ be the trained model. If we increase the size of the hypothesis class (i.e., make the model more complicated and expressive, which corresponds to enlarge the Lipschitz constants of $\phi$ in Figure 2), intuitively
> - training loss $\hat{\epsilon}_n(\hat{h}_n)$ is going to decrease, because we are optimizing over a larger hypothesis class. Empirically this corresponds to the overall decreasing trend of training loss as shown in the left-most plots in Figure 2.
> - generalization gap $\epsilon(\hat{h}_n)-\hat{\epsilon}_n(\hat{h}_n)$ is going to increase, because from Proposition 3.1 this term is upper bounded by the complexity of the model and the divergence between training and test data distribution. Empirically this corresponds to the overall increasing trend of generalization gap shown in the right-most plots in Figure 2.
>
> > **W4**: Perhaps additional experiments could be useful, e.g.,: 1. An experiment comparing the generalisation gap of LapPE/BasisNet/SPE. 2. Evaluating the performance of LapPE/BasisNet/SPE on LRGB or TUDatasets.
>
> Thank you for the good suggestions.
>
> 1. Now we add the training loss and the generalization gap (test loss - training loss) on ZINC dataset as shown in Table 3, Appendix B.5 in the refined manuscript. We can see that though SignNet and BasisNet achieve a pretty low training MAE (high expressive power), their generalization gap is larger than other baselines (poor stability) and thus the final test MAE is not the best. For baseline GNN and PEG, they are pretty stable with small generalization gap, but the poor expressive power make them hard to fit the dataset well (training loss is high). In contrast, SPE has not only a lowest training MAE (high expressive power) but also a small generalization gap (good stability). That is why it can obtain the best test performance among all the models.
>
> 2. We are working on the TUDatasets you suggest and the results will be released before the end of rebuttal. LRGB consists of more and larger molecular graphs and is hard to be covered due to limited time of rebuttal period.
>
> [1] Dwivedi, Vijay Prakash, et al. "Graph neural networks with learnable structural and positional representations." arXiv preprint arXiv:2110.07875 (2021).
>
> [2] Eliasof, Moshe, et al. "Graph Positional Encoding via Random Feature Propagation." arXiv preprint arXiv:2303.02918 (2023).
>
> [3] Rampášek, Ladislav, et al. "Recipe for a general, powerful, scalable graph transformer." Advances in Neural Information Processing Systems 35 (2022): 14501-14515.
>
> [4] Bodnar, Cristian, et al. "Weisfeiler and lehman go cellular: Cw networks." Advances in Neural Information Processing Systems 34 (2021): 2625-2640.
>
> [5] Frasca, Fabrizio, et al. "Understanding and extending subgraph gnns by rethinking their symmetries." Advances in Neural Information Processing Systems 35 (2022): 31376-31390.
>
> [6] Zhang, Bohang, et al. "A complete expressiveness hierarchy for subgraph gnns via subgraph weisfeiler-lehman tests." arXiv preprint arXiv:2302.07090 (2023).

---

> > ### Author Response · Authors · 2023-11-22
> > **New results on TUDataset**
> >
> > We just get the new results on TUDataset as you suggested. Please kindly refer to Appendix B.8, Table 6 for results. Overall we can see that SPE achieves the best results on 3 out of 4 different datasets.

---

### Official Review · Reviewer_Fdup · 2023-11-03

**Soundness:** 4 excellent
**Presentation:** 3 good
**Contribution:** 2 fair
**Rating:** 8
**Confidence:** 4

**Summary:**

This works further deepens results on basis-invariant GNNs building on work of e.g. Wang et al (ICLR 2022) and Lim&Robinson et al (ICLR 2023). The authors propose a simple generalization of basis-net, with strong theoretical guarantees (Hölder-smoothness / stability and basis-invariance). They achieve strong performance in standard molecular benchmarks, out-of-distribution benchmarks and cycle count tasks.

**Strengths:**

Extremely well written and easy to follow. Figure 1 is nice!

The proposed model SPE (Eq. 2) is a simple generalization extension of basis-net. It is very interesting to see that such a straightforward modification yields such strong (and non-trivial) theoretical results (Theorem 1, etc.).

Empirical performance is very convincing, e.g. for Zinc and the OOD tests.

------ during rebuttal ------
as reviewers addressed my concerns and in fact added clarifications on runtime and more importantly on the unstability of previous methods I raised my score and now clearly vote for acceptance.

**Weaknesses:**

The achieved theoretical and empirical results, while very interesting, seem somewhat incremental compared to Wang et al (ICLR 2022) and Lim&Robinson et al (ICLR 2023). The authors should discuss the differences more clearly. In particular:
* The reason for Hölder continuity ($c\neq1$) is not fully clear. E.g. does PEG / standard basis-net already already satisfy your stability criterion Def 3.1 and / or Assumption 3.1.? If yes, what is the conceptual / theoretical benefit of your proposed architecture. If not, could you please provide an argument why the assumptions fail for PEG / basis-net.
* Is $c\neq 1$ crucial for any proof / guarantee / assumption?
* Are there cases where SPE satisfies the stability assumption of Wang et al (ICLR 2022), i.e., with $c=1$?

Please see also the questions below.

Minor:
* Remark 3.1 Should probably be attributed to Wang et al (ICLR 2022), as they have the same statement for $c=1$.

I am happy to raise my score, if the authors properly address my concerns and questions.

**Questions:**

Please provide runtimes for your proposed method. Preferably for pre-processing and overall runtime. This would help to put the achieved results into context with basis-net, etc.

While the OOD generalization bound is very interesting, can you also state a standard PAC-style generalization bound (same distribution for train and test)?

Do you have a counter-example where SPE cannot count $k$-cycles for $k\leq 6$?

The $n\times n\times m$ might be somewhat excessive for certain datasets. Would it be possible to exchange $V\\phi(\cdot) V^T$ to $V^T\phi(\cdot) V^T$ to get the much smaller $d\times d\times m$ instead? If not what would this more compact model correspond to?

If I am notmistaken SPE and basis-net should have the same expressivity and thus at least theoretically be both equally capable of counting cycles etc. Can you provide some intuition why SPE performs significantly better than basis-net in this task (Figure 3)?

---

> ### Author Response · Authors · 2023-11-18
> **Response to Reviewer Fdup**
>
> We thank the reviewer for the insightful question. Below is our response.
> >  **W1**: does PEG / standard basis-net already satisfy your stability criterion Def 3.1 and / or Assumption 3.1.?
>
> BasisNet does not have stability guarantee Def 3.1 because it does not satisfy Assumption 3.1. In short that is because we can see it as using discontinuous $\phi$ functions, which violates Assumption 3.1. Kindly check our unified response  ``*Why previous methods are unstable?* '' above.
>
> PEG is indeed stable, but its expressive power is extremely limited. This is becasue PEG cannot distinguish eigenvectors $V\in\mathbb{R}^{n\times d}$ and eigenvectors $VQ\in\mathbb{R}^{n\times d}$, where $Q\in O(d)$ is an arbitrary $d$-dim rotation matrix. This point is also summarized in our Introduction section `` But, to achieve stability, it (PEG) completely ignores the distinctness of each eigensubspaces
> and processes the merged eigenspaces homogeneously. Consequently, it loses expressive power and
> has, e.g., a subpar performance on molecular graph regression tasks''. And this also got verified by our experiments on ZINC, Table 1, which shows that the performance of PEG is limited. Additionally, we show that it is due to the high training loss of PEG in Table 3, Appendix B.5 in the revised manuscript. This verifies the poor expressive power of PEG.
>
> > **W2&3**: Is $c\neq 1$ crucial for any proof / guarantee / assumption? Are there cases where SPE satisfies the stability assumption of Wang et al (ICLR 2022), i.e., with $c=1$?
>
> $c\neq 1$ serves for describing the stability property of SPE, which is the general Holder continuity instead of Lipschitz continuity. For the second question, we think it is an open question. At this point we are not sure whether SPE can have a Lipschitz bound instead of a Holder bound. But fortunately even $c\neq 1$ is sufficient to provide a OOD generalization guarantee.
>
> > **Q1**: Please provide runtimes for your proposed method. Preferably for pre-processing and overall runtime. This would help to put the achieved results into context with basis-net, etc.
>
> Thank you for pointing this out. We add the average running time of training and inference, which can be found in our unified response ``*running time evaluation*'' above. The pre-processing is the standard Laplacian decomposition same as others.
>
> > **Q2**: While the OOD generalization bound is very interesting, can you also state a standard PAC-style generalization bound (same distribution for train and test)?
>
> The PAC generalization bound is quite different from OOD bound. OOD bound states how the population risk may change for the same prediction model but two different population distributions, while the PAC theory describes the difference of population risk between empirical risk minimizer and the population risk minimizer.
>
> > **Q3**: Do you have a counter-example where SPE cannot count $k$-cycles for $k\le 6$?
>
> As implied by Proposition 3.4, SPE can count all $k$-cycles with $k\le 6$. So there is no such counter-example.
>
> > **Q4**: The  $n\times n\times m$ might be somewhat excessive for certain datasets. Would it be possible to exchange $V\phi(\cdot)V^{\top}$to $V^{\top}\phi(\cdot)V^{\top}$ to get the much smaller $d\times d\times m$ instead? If not what would this more compact model correspond to?
>
> We cannot naively do $V^{\top}\phi V$ as the dimension does not match. One alternative way could be to sparse the $V\phi V^{\top}$ by only computing and storing its $(i,j)$-entry where $(i,j)$ is an edge. Then these positional encodings are treated as edge features for downstream tasks. The complexity will reduce from $O(n^2)$ to $O(|E|)$ where $|E|$ is the number of edges. In practice, the sparsing of $V\phi V^{\top}$ can be efficiently done by firstly assigning $V_{i, :}$ to $i$-th node features and using an one-layer message passing neural network to update and store edge features $[V\phi V^{\top}]\_{i,j}=\sum_{k}[\phi(\lambda)]_{k} V\_{i,k}V\_{j,k}$.
>
> > **Q5**: Can you provide some intuition why SPE performs significantly better than basis-net in this task (Figure 3)?
>
> This is because BasisNet is not a stable method and thus generalizes poorly. Actually both SPE and BasisNet can fit pretty well on training dataset (training loss < 0.05). However, BasisNet suffers from overfitting with a large generalization gap (test error - training error). In contrast, the generalization gap of SPE is almost zero. This phenomenon is also observed on ZINC, as shown in Table 3, Appendix B.5 in the refined manuscript.

---

> > ### Comment · Reviewer_Fdup · 2023-11-18
> > **Quick follow up**
> >
> > Thank you for your reply. Two quick follow-up comments.
> >
> > * The $\leq 6$ was a typo, I've meant to ask whether you know counter-examples where SPE cannot count cycles of size $\geq 7$.
> > * Of course PAC and OOD are different. I was just curious whether your assumptions (stability, Hölder-smooth, etc.) can also be used to derive PAC bounds.

---

> > > ### Author Response · Authors · 2023-11-19
> > > **Response to follow up**
> > >
> > > Thank you for the follow-up comments.
> > >
> > > - we did not examine whether such counter-example exists. One possible way is to first check if the classic 8-cycle counter-example, 4x4 rook’s graph and Shrikhande graph [1], can be distinguished by SPE.
> > > - Holder constant can be used to denote model capacity and thus is related to PAC bound. But currently we do not know whether there is a quantitative relation between them.
> > >
> > > [1] Arvind V, Fuhlbrück F, Köbler J, et al. On weisfeiler-leman invariance: Subgraph counts and related graph properties[J]. Journal of Computer and System Sciences, 2020, 113: 42-59

---

> > > > ### Comment · Reviewer_Fdup · 2023-11-19
> > > >
> > > > Thank you for addressing my concerns. In particular, thanks for the clarifications about the unstability of previous methods and the runtime aspects. I raise my score.

---

### Author Response · Authors · 2023-11-18
**Rebuttal Summary**

We thank the reviewers for the time to read our manuscript and their constructive suggestions. All reviewers agree on the importance of stable positional encodings and appreciate our theoretical contributions of analysing and addressing the stability problem. In particular, Reviewer qQbF and mePj believe our work can greatly benefits the graph learning community since stability means better generalization to unseen graphs but it is a topic unexplored too much yet. Reviewer fM7Q think the way we address stability is novel. On the other hand, reviewers Fdup and TVwL feel uncertain about if previous methods are truly unstable and provide insightful questions. Reviewers Fdup, qQbF and mePj have concerns regarding the complexity of the proposed method. We are going to address these two main concerns in unified responses below. We also respond to each reviewer separately and carefully, and expect to get more feedback. Correspondingly, we take the suggestions from reviewers and make changes in the **refined manuscript in blue font**.

---

### Author Response · Authors · 2023-11-18
**Why previous methods are unstable?**

To address the concern from reviewers Fdup and TVwL, we are going to show why previous positional encodings are not stable. The following analysis can also be found at Appendix C in the refined manuscript.

## All sign-invariant methods are unstable
One line of work [1, 2, 3] is to consider the sign ambiguity of each individual eigenvectors and aim to make positional encodings invariant to sign flipping. The underlying assumption is that eigenvalues are all distinct so eigenvectors are equivalent up to a sign transformation. However, we are going to show that all these sign-invariant methods are **unstable**, regardless of the eigenvalues being distinct or not.

Firstly, suppose eigenvalues are distinct. Lemma 3.4 in [4] states that

**Lemma 3.4.**
For any positive semi-definite matrix $B\in\mathbb{R}^{N\times N}$ without multiple eigenvalues, set positional encoding $\text{PE}(B)$ as the eigenvectors given by the smallest $p$ eigenvalues sorted as $0=\lambda_1<\lambda_2<...<\lambda_p(<\lambda_{p+1})$ of $B$. For any suffciently small $\epsilon>0$, there exists a perturbation $\Delta B$, $\lVert B\rVert_{F} \le \epsilon$ such that
$$
    \min_{S\in \text{SN}(p)}\lVert{\text{PE}(B)-\text{PE}(B+\Delta B)\rVert_{F}\ge 0.99\max_{1\ge i\le p}|\lambda_{i+1}-\lambda_i|^{-1}\lVert\Delta B}\rVert_{F}+o(\epsilon),
$$
where $\text{SN}(p)=\{Q\in\mathbb{R}^{p\times p}:Q_{i,i}=\pm 1, Q_{i,j}=0, \text{for } i\neq j\}$ is the sign flipping operations.

This Lemma shows that when there are two closed eigenvalues, a small perturbation to graph may still yield a huge change of eigenvectors that cannot be compensated by sign flipping. Therefore, these sign-invariant methods are highly unstable on graphs with distinct but closed eigenvalues.

On the other hand, if eigenvalues have repeated values, then the same graph may produce different eigenvectors that are associated by basis transformations. Simply invariant to sign flipping cannot handle this basis ambiguity. As a result, these sign-invariant methods will produce different positional encodings for the same input graph. That means there is no stability gurantee for them at all.




## BasisNet is unstable

Another line of work (e.g, BasisNet [3]) further consider basis invariance of eigenvectors by separately dealing with each eigensubspaces instead of each individual eigenvectors.
The idea is to first partition eigenvectors $V\in\mathbb{R}^{n\times d}$ into their corresponding eigensubspace $(V_1, V_2, ...)$ according to eigenvalues, where $V_k\in\mathbb{R}^{n\times d_k}$ is the eigenvectors in $k$-th eigensubspace of dimension $d_k$. Then neural networks $\phi_{d_k}:\mathbb{R}^{n\times n}\to\mathbb{R}^{n\times p}$ is applied to each $V_kV_k^{\top}$ and the output will be $\rho(\phi_{d_1}(V_1V_1^{\top}), \phi_{d_2}(V_2V_2^{\top}), ...)$ where $\rho:\mathbb{R}^{n\times (d\cdot p)}\to\mathbb{R}^{n\times p}$ is a MLP. Intuitively, this method is unstable because a perturbation of graph can change the dimension of eigensubspace and thus dramatically change the input $(V_1, V_2,...)$. As an example, let us say we have three eigenvectors ($d=3$), and denote the three columns of $V$ as $u_1, u_2, u_3$. We construct two graphs: the original graph $A$ has $\lambda_1=\lambda_2<\lambda_3$ while the perturbed graph $A^{\prime}$ has $\lambda_{1}^{\prime}<\lambda_{2}^{\prime}<\lambda_{3}^{\prime}$. Two graphs share the same eigenvectors. Note that the difference between $A$ and $A^{\prime}$ can be arbitrarily small to make $\lambda_2^{\prime}$ a little bit different from $\lambda_2$. BasisNet will produce the following embeddings:
$$\text{BasisNet}(A)=\rho(\phi_{2}(u_1u_1^{\top}+u_2u_2^{\top}), \phi_1(u_3u_3^{\top}))$$
$$\text{BasisNet}(A^{\prime})=\rho(\phi_1(u_1u_1^{\top}), \phi_1(u_2u_2^{\top}), \phi_1(u_3u_3^{\top})).$$
Clearly as the input to $\rho$ are completly different, there is no way to ensure stability even if $\rho$ and $\phi$ are continuous.



[1] Vijay Prakash Dwivedi and Xavier Bresson. A generalization of transformer networks to graphs. AAAI Workshop on Deep Learning on Graphs: Methods and Applications, 2021.

[2] Devin Kreuzer, Dominique Beaini, Will Hamilton, Vincent  Létourneau, and Prudencio Tossou. Rethinking graph transformers with spectral attention. Advances in Neural Information Processing Systems, 34, 2021.

[3] Derek Lim, Joshua David Robinson, Lingxiao Zhao, Tess Smidt, Suvrit Sra, Haggai Maron, and Stefanie Jegelka. Sign and basis invariant networks for spectral graph representation learning. In The Eleventh International Conference on Learning Representations, 2023.

[4] Haorui Wang, Haoteng Yin, Muhan Zhang, and Pan Li. Equivariant and stable positional encoding for more powerful graph neural networks. In International Conference on Learning Representations, 2022a.

---

### Author Response · Authors · 2023-11-18
**Running time evaluation**

To address the complexity concern, we empirically evaluate the running time of SPE and other baselines on ZINC and DrugOOD. The GPU we used is Quadro RTX 6000. All methods follow the same settings as in previous experiments on ZINC and DrugOOD. We uniformly use batch size 128 for ZINC and batch size 64 for DrugOOD. The results represent the training/inference time on the whole training dataset (ZINC or DrugOOD) over 5 runs. We can see that on ZINC (average graph size 23.2) the speed SPE is overall comparable to SignNet, and is much faster than BasisNet. On DrugOOD, the graph size becomes larger (average graph size 34.6), so SPE becomes a little bit slower than SignNet. Still, SPE is much faster than BasisNet. This is possibly because BasisNet has to deal with the irregular and length-varying input $[V_1V_1^{\top}, V_2V_2^{\top}, ...]$, which is hard for parallel computation in a batch, while SPE simply needs to deal with the more uniform $V\text{diag}(\phi(\lambda))V^{\top}$.

### **ZINC**
| PE Method | #PEs | Training Time (s)    | Inference Time (s)   |
|-----------|------|----------------------|----------------------|
| No PE     | N/A  | $3.319_{\pm 0.400}$  | $2.852_{\pm 0.310}$  |
| PEG       | 8    | $3.785_{\pm 0.424}$  | $3.590_{\pm 0.341}$  |
| PEG       | full | $3.639_{\pm 0.387}$  | $3.518_{\pm 0.318}$  |
| SignNet   | 8    | $8.724_{\pm 0.686}$  | $3.546_{\pm 0.366}$  |
| SignNet   | full | $23.157_{\pm 1.932}$ | $7.883_{\pm 0.374}$  |
| BasisNet  | 8    | $49.923_{\pm 6.391}$ | $18.295_{\pm 0.569}$ |
| BasisNet  | full | $66.176_{\pm 4.015}$ | $27.546_{\pm 0.622}$ |
| SPE       | 8    | $10.888_{\pm 0.416}$ | $5.336_{\pm 0.738}$  |
| SPE       | full | $11.576_{\pm 0.472}$ | $5.406_{\pm 0.499}$  |

### **DrugOOD-assay**
| PE Method | #PEs | Training Time (s)     | Inference Time (s)   |
|-----------|------|-----------------------|----------------------|
| No PE     | N/A  | $13.560_{\pm0.657}$   | $4.260_{\pm 0.140}$  |
| PEG       | 32   | $14.212_{\pm 1.084}$  | $4.780_{\pm 0.351}$  |
| SignNet   | 32   | $30.705_{\pm 0.723}$  | $12.844_{\pm 0.307}$ |
| BasisNet  | 32   | $199.364_{\pm 2.807}$ | $86.529_{\pm 1.693}$ |
| SPE       | 32   | $37.577_{\pm 0.833}$  | $21.706_{\pm 0.850}$ |

---

### Meta-Review · Area_Chair_Vg81 · 2023-12-06

**Metareview:**

This manuscript presents a novel approach to addressing stability in graph positional encodings. SPE offers both theoretical and practical benefits. The paper is well-articulated, with a clear exposition. The theoretical analysis of SPE is interesting (proven stability and expressive power). The empirical results, particularly in molecular property prediction and OOD generalization, are convincing.

The concerns raised by some reviewers regarding the novelty and complexity of the method are sound. Nevertheless, I believe they are quite minors regarding paper's overall contribution. The authors have adequately addressed most of the important issues raised by the reviewers in their responses and manuscript revisions.

**Justification For Why Not Higher Score:**

Somewhat incremental compared to existing methods, and the complexity of the SPE architecture could limit its applicability in certain scenarios.

**Justification For Why Not Lower Score:**

Strong theoretical foundation and convincing empirical results, clearly warrant acceptance.

---

### Decision · Program_Chairs · 2024-01-16

Accept (poster)